# MACHINE LEARNING FOR ELLIPTIC PDES: FAST RATE GENERALIZATION BOUND, NEURAL SCALING LAW AND MINIMAX OPTIMALITY

**Yiping Lu**
Institute for Computational
& Mathematical Engineering
Stanford University
Stanford, CA 94305, USA
`yplu@stanford.edu`

**Haoxuan Chen**
Department of Computing
and Mathematical Sciences,
Caltech
Pasadena, CA 91125, USA
`haoxuan@caltech.edu`

**Jianfeng Lu**
Mathematics Department
Duke University
Durham, NC 27708-0320
`jianfeng@math.duke.edu`

**Lexing Ying**
Department of Mathematics
Stanford University
Stanford, CA 94305, USA
`lexing@stanford.edu`

**Jose Blanchet**
Department of Management Science & Engineering
Stanford University
Stanford, CA 94305, USA
`jose.blanchet@stanford.edu`

## ABSTRACT

In this paper, we study the statistical limits of deep learning techniques for solving elliptic partial differential equations (PDEs) from random samples using the Deep Ritz Method (DRM) and Physics-Informed Neural Networks (PINNs). To simplify the problem, we focus on a prototype elliptic PDE: the Schrödinger equation on a hypercube with zero Dirichlet boundary condition, which is applied in quantum-mechanical systems. We establish upper and lower bounds for both methods, which improve upon concurrently developed upper bounds for this problem via a fast rate generalization bound. We discover that the current Deep Ritz Method is sub-optimal and propose a modified version of it. We also prove that PINN and the modified version of DRM can achieve minimax optimal bounds over Sobolev spaces. Empirically, following recent work which has shown that the deep model accuracy will improve with growing training sets according to a power law, we supply computational experiments to show similar-behavior of dimension dependent power law for deep PDE solvers.

## 1 INTRODUCTION

Partial differential equations (PDEs) play a prominent role in many disciplines of science and engineering. The recent deep learning breakthrough and the rapid development of sensors, computational power, and data storage in the past decade has drawn attention to numerically solving PDEs via machine learning methods (Long et al., 2018; 2019; Raissi et al., 2019; Han et al., 2018; Sirignano & Spiliopoulos, 2018; Khoo et al., 2017), especially in high dimensions where conventional methods become impractical. The set of applications that motivate this interest is wide-ranging, including computational physics (Han et al., 2018; Long et al., 2018; Raissi et al., 2019), inverse problem (Zhang et al., 2018; Gilton et al., 2019; Fan & Ying, 2020) and quantitative finance (Heaton et al., 2017; Germain et al., 2021). The numerical methods generated by the use of deep learning techniques are mesh-less methods, see the discussion in (Xu, 2020). A natural deep learning technique in the

problems that are based on a standard feed-forward type of architecture takes advantage (when available) of a variational formulation, whose solution coincides with the solution of the PDE of interest. Despite the success and popularity of adopting neural networks for solving high-dimensional PDEs, the following question still remains poorly answered.

*For a given PDE and a data-driven approximation architecture, how large the sample size and how complex the model are needed to reach a prescribed performance level?*

In this paper, we aim to establish the numerical analysis of such deep learning based PDE solvers. Inspired by recent works which showed that the empirical performance of a model is remarkably predictable via a power law of the data number, known as the neural scaling law (Kaplan et al., 2020; Hestness et al., 2017; Sharma & Kaplan, 2020), we aim to explore the neural scaling law for deep PDE solvers and compare its performance to Fourier approximation.

Among the various approaches of using deep learning methods for solving PDEs, in this work, we focus on the Deep Ritz method (DRM) (E & Yu, 2018; Khoo et al., 2017) and the Physics-Informed Neural Networks (PINN) approach (Sirignano & Spiliopoulos, 2018; Raissi et al., 2019), both of which are based on minimizing neural network parameters according to some loss functional related to the PDEs. To provide theoretical guarantees for DRM and PINN, following (Lu et al., 2021b; Duan et al., 2021; Bai et al., 2021), we decompose the error into approximation error (Yarotsky, 2017; Suzuki, 2018; Shen et al., 2021) and generalization error (Bartlett et al., 2005; Xu & Zeevi, 2020; Farrell et al., 2021; Schmidt-Hieber et al., 2020; Suzuki, 2018). However, instead of the $O(1/\sqrt{n})$ ($n$ is the number of data sampled) slow rate generalization bounds established in prior work (Lu et al., 2021b; Shen et al., 2021; Xu, 2020; Shin et al., 2020), we utilize the strongly convex structure of the DRM and PINN objectives and provide an $O(1/n)$ fast rate generalization bound (Bartlett et al., 2005; Xu & Zeevi, 2020) that leads us to a non-parametric estimation bound. Our theory also suggests an optimal selection of network size with respect to the number of sampled data. Moreover, to illustrate the optimality of our upper bound, we also establish an information-theoretic lower bound which matches our upper bound for PINN and a modified version of DRM.

We also test our theory by numerical experiments. Recent works (Hestness et al., 2017; Kaplan et al., 2020; Rosenfeld et al., 2019; Mikami et al., 2021) studying a variety of deep learning algorithms all find the same polynomial scaling relation between the testing error and the number of data. As the number of training data $n$ increases, the population loss $\mathcal{L}$ of well-trained and well-tuned models scales with $n$ as a power-law $\mathcal{L} \propto \frac{1}{n^\alpha}$ for some $\alpha$. (Sharma & Kaplan, 2020) also scans over a large range of $\alpha$ and problem dimension $d$ and finds an approximately $\alpha \propto \frac{1}{d}$ scaling law. In Section 4, we conduct numerical experiments to show that this phenomenon still appears for deep PDE solvers and this neural scaling law tests more idiosyncratic features of the theory.

## 1.1 RELATED WORKS

**Neural Scaling Law**   The starting point of our work is the recent observation across speech, vision and text (Hestness et al., 2017; Kaplan et al., 2020; Rosenfeld et al., 2019; Rosenfeld, 2021) that the empirical performance of a model satisfies a power law scales as a power-law with model size and dataset size. (Sharma & Kaplan, 2020) further finds out that the power of the scaling law depends on the intrinsic dimension of the dataset. Theoretical works (Schmidt-Hieber et al., 2020; Suzuki, 2018; Suzuki & Nitanda, 2019; Chen et al., 2019b; Imaizumi & Fukumizu, 2020; Farrell et al., 2021; Jiao et al., 2021c) explore the optimal power law under the non-parametric curve estimation setting via a plug-in neural network. Our work extends this line of research to solving PDEs.

**Deep Network Based PDE Solver.**   Solving high dimensional partial differential equations (PDEs) has been a long-standing challenge due to the curse of dimensionality. At the same time, deep learning has shown superior flexibility and adaptivity in approximating high dimensional functions, which leads to state-of-the-art performances in a wide range of tasks ranging from computer vision to natural language processing. Recent years, pioneer works (Han et al., 2018; Raissi et al., 2019; Long et al., 2018; Sirignano & Spiliopoulos, 2018; Khoo et al., 2017) try to utilize the deep neural networks to solve different types of PDEs and achieve impressive results in many tasks (Lu et al., 2021a; Li et al., 2020). Based on the natural idea of representing solutions of PDEs by (deep) neural networks, different loss functions for solving PDEs are proposed. (Han et al., 2018; 2020) utilize the

Feynman-Kac formulation which turns solving PDE to a stochastic control problem and the weak adversarial network (Zang et al., 2020) solves the weak formulations of PDEs via an adversarial network. In this paper, we focus on the convergence rate of the Deep Ritz Method (DRM) (E & Yu, 2018; Khoo et al., 2017) and Physics-Informed neural network (PINN) (Raissi et al., 2019; Sirignano & Spiliopoulos, 2018). DRM (E & Yu, 2018; Khoo et al., 2017) utilizes the variational structure of the PDE, which is similar to the Ritz-Galerkin method in classical numerical analysis of PDEs, and trains a neural network to minimize the variational objective. PINN (Raissi et al., 2019; Sirignano & Spiliopoulos, 2018) trains a neural network directly to minimize the residual of the PDE, i.e., using the strong form of the PDE.

**Theoretical Guarantees For Machine Learning Based PDE Solvers.** Theoretical convergence results for deep learning based PDE solvers raises wide interest recently. Specifically, (Lu et al., 2021b; Grohs & Herrmann, 2020; Marwah et al., 2021; Wojtowytsch et al., 2020; Xu, 2020; Shin et al., 2020; Bai et al., 2021) investigate the regularity of PDEs approximated by neural network and (Lu et al., 2021b; Luo & Yang, 2020) further provide a generalization analysis. (Nickl et al., 2020) introduces a prior over the solution of the PDE and considers an equivalent white noise model (Brown & Low, 1996). (Nickl et al., 2020) provides the rate of convergence of the posterior. Our paper does not need to introduce the prior on the target function and provides a non-asymptotic guarantee for finite number of data. At the same time, (Nickl et al., 2020) can only be applied to linear PDEs while our proof technique can be extended to nonlinear ones. All these papers also fail to answer the question that how to determine the network size corresponding to the sampled data number to achieve a desired statistical convergence rate. (Hütter & Rigollet, 2019; Manole et al., 2021) consider the similar problem for the optimal transport problem, *i.e.* Monge-ampere equation. Nevertheless, the variational problem we considered is different from (Hütter & Rigollet, 2019; Manole et al., 2021) and leads to technical difference. The most related works to ours are two **concurrent** papers (Duan et al., 2021; Jiao et al., 2021a;b). However, our upper bound is faster than (Duan et al., 2021; Jiao et al., 2021a;b). In this paper, we also show that generalization analysis in (Lu et al., 2021b; Duan et al., 2021; Luo & Yang, 2020) are loose due to the lack of a localization technique (De Boor & De Boor, 1978; Bartlett et al., 2005; Koltchinskii, 2011; Xu, 2020). With observation of the strong convexity of the loss function, we follow the fast rate results for ERM (Schmidt-Hieber et al., 2020; Xu & Zeevi, 2020; Farrell et al., 2021) and provide a near optimal bound for both DRM and PINN.

## 1.2 CONTRIBUTION

In short, we summarize our contribution as follows

- In this paper, we first considered the statistical limit of learning a PDE solution from sampled observations. The lower bound shows a non-standard exponent different from non-parametric estimation of a function.

- Instead of the $O(1/\sqrt{n})$ slow rate generalization bounds in (Lu et al., 2021b; Duan et al., 2021; Jiao et al., 2021a;c), we utilized the strongly convex nature of the variational form and provided a fast rate generalization bound via the localization methods (Van De Geer, 1987; Bartlett et al., 2005; Koltchinskii, 2011; Srebro et al., 2010; Xu & Zeevi, 2020). We discovered that the current Deep Ritz Methods is sub-optimal and propose a modified version of it. We showed that PINN and the modified version of DRM can achieve nearly min-max optimal convergence rate. Our result is listed in Table 1.

- We tested the recently discovered neural scaling law (Hestness et al., 2017; Kaplan et al., 2020; Rosenfeld et al., 2019; Hashimoto, 2021) for deep PDE solvers numerically. The empirical results verified our theory.

## 2 SET-UP

We consider the static Schrödinger equation with zero Dirichlet boundary condition on the domain $\Omega$, which we assume to be the unit hypercube in $\mathbb{R}^d$. In order to precisely introduce the problem, we recall some standard notions. We consider our domain as $\Omega = [0, 1]^d$ and use $L^2(\Omega)$ to denote the space of square integrable functions on $\Omega$ with respect to the Lebesgue measure. We let $L^\infty(\Omega)$ be

| | Upper Bounds | | | Lower Bound |
|---|---|---|---|---|
| Objective Function | Neural Network | Previous Bound | Fourier Basis | |
| Deep Ritz | $n^{-\frac{2s-2}{d+2s-2}}\log n$ | $n^{-\frac{2s-2}{d+4s-4}}\log n$ (Duan et al., 2021) | $n^{-\frac{2s-2}{d+2s-2}}$ | $n^{-\frac{2s-2}{d+2s-4}}$ |
| Modified Deep Ritz | $n^{-\frac{2s-2}{d+2s-2}}\log n$ | / | $n^{-\frac{2s-2}{d+2s-4}}$ | $n^{-\frac{2s-2}{d+2s-4}}$ |
| PINN | $n^{-\frac{2s-4}{d+2s-4}}\log n$ | $n^{-\frac{2s-4}{d+4s-8}}\log n$ (Jiao et al., 2021a) | $n^{-\frac{2s-4}{d+2s-4}}$ | $n^{-\frac{2s-4}{d+2s-4}}$ |

Table 1: Upper bounds and lower bounds we achieve in this paper and previous work. The upper bound colored in red indicates that the convergence rate matches the min-max lower bound.

the space of essentially bounded (with respect to the Lebesgue measure) functions on $\Omega$ and $C(\partial\Omega)$ denotes the space of continuous functions on $\partial\Omega$.

Let $f \in L^2(\Omega), V \in L^\infty(\Omega)$, and $, g \in L^\infty(\Omega)$. Our focus is on the analysis of Deep-Learning-based numerical methods to solve the elliptic PDE:

$$\begin{aligned} -\Delta u + Vu &= f &&\text{in } \Omega, \\ u &= g &&\text{on } \partial\Omega. \end{aligned} \tag{2.1}$$

## 2.1 LOSS FUNCTIONS FOR SOLVING PDEs AND INDUCED EVALUATION METRIC

In this paper, we mainly focus on analyzing Deep Ritz Methods (DRM) and Physics-Informed Neural Network (PINN). In this subsection, we first introduce the objective function and algorithm of the two methods.

**Deep Ritz Methods**  (E & Yu, 2018; Sirignano & Spiliopoulos, 2018) Recall that the equation 2.1 is equivalent to following variational form

$$u^* = \arg\min_{H_0^1(\Omega)} \boldsymbol{E}^{\text{DRM}}(u) := \arg\min_{H_0^1(\Omega)} \frac{1}{2}\int_\Omega \|\nabla u\|^2 + V|u|^2 \, dx - \int_\Omega fu dx, \tag{2.2}$$

where $u$ is minimized over $H_0^1(\Omega)$ with boundary condition given by $g$ on $\partial\Omega$.

This variational form provides the basis for the DRM type method for solving the static Schrödinger equation based on neural network ansatz. More specifically, the energy functional given in equation 2.2 is viewed as the population risk function to train an optimal estimator approximation of the solution to the PDE within a parameterized hypothesis function class $\boldsymbol{F} \subset H_0^1(\Omega)$. In this paper, we also rely on the strong convexity of the DRM objective with respect to the $H^1$ norm.

**Proposition 2.1.** *For DRM, we further assume $0 < V_{\min} \le V(x) \le V_{\max}$, then we have*

$$\frac{2}{\max\{1, V_{\max}\}}\left(\boldsymbol{E}^{DRM}(u) - \boldsymbol{E}^{DRM}(u^*)\right) \le \|u - u^*\|_{H^1}^2 \le \frac{2}{\max\{1, V_{\min}\}}\left(\boldsymbol{E}^{DRM}(u) - \boldsymbol{E}^{DRM}(u^*)\right)$$

*holds for all $u \in H_0^1(\Omega)$*

**Physics-Informed Neural Network**  (Raissi et al., 2019; Sirignano & Spiliopoulos, 2018). PINN solves 2.1 via minimizing the following objective function

$$u^* = \arg\min_{H_0^1(\Omega)} \boldsymbol{E}^{\text{PINN}}(u) := \arg\min_{H_0^1(\Omega)} \int_\Omega |\Delta u(x) - V(x)u(x) + f(x)|^2 dx.$$

The objective function $\boldsymbol{E}^{\text{PINN}}$ can also be viewed as the population risk function and we can train an optimal estimator approximation of the solution to the PDE within a parameterized hypothesis function class $\boldsymbol{F} \subset H_0^1(\Omega)$. In this paper, we also rely on the strong convexity of the PINN objective with respect to the $H^2$ norm, for which we need some additional assumptions on the potential.

**Proposition 2.2.** *For PINN, we further assume* $V \in L^{\infty}(\Omega)$ *with* $0 < C_{\min} < V^2 - \Delta V, 0 < C_{\min} < V(x) \leq V_{\max}$ *and* $-\Delta V(x) \leq V_{\max}$, *then we have for all* $u \in H_0^1(\Omega)$

$$\frac{1}{2\left(1 + V_{\max} + V_{\max}^2\right)}\left(\boldsymbol{E}^{PINN}(u) - \boldsymbol{E}^{PINN}(u^*)\right) \leq \|u - u^*\|_{H^2}^2$$

$$\leq \frac{2}{\max\{1, C_{\min}\}}\left(\boldsymbol{E}^{PINN}(u) - \boldsymbol{E}^{PINN}(u^*)\right).$$

## 2.2 ESTIMATOR SETTING

**Empirical Loss Minimization** In order to access the $d$-dimensional integrals, DRM (E & Yu, 2018; Khoo et al., 2017) and PINN (Raissi et al., 2019; Sirignano & Spiliopoulos, 2018) employ a Monte-Carlo method for computing the high dimensional integrals, which leads to the so-called *empirical risk minimization* training for neural networks. To define the empirical loss, let $\{X_j\}_{j=1}^n$ be an i.i.d. sequence of random variables distributed according to the uniform distribution $\boldsymbol{P}_{\Omega}$ in domain $\Omega$. We also have access to noisy observations $f_j = f(X_j) + \xi_j$ $(1 \leq j \leq n)$ of the right hand side of the PDE (2.1), where $\xi_j$ are i.i.d. bounded random variables with zero mean and independent of $X_j$. Define the empirical losses $\boldsymbol{E}_n$ by setting

$$\boldsymbol{E}_n^{\text{DRM}}(u) = \frac{1}{n}\sum_{j=1}^n\left[|\Omega| \cdot \left(\frac{1}{2}\|\nabla u(X_j)\|^2 + \frac{1}{2}V(X_j)|u(X_j)|^2 - f_j u(X_j)\right)\right], \tag{2.3}$$

$$\boldsymbol{E}_n^{\text{PINN}}(u) = \frac{1}{n}\sum_{j=1}^n\left[|\Omega| \cdot \left(\Delta u(X_j) - V(X_j)u(X_j) + f_j\right)^2\right], \tag{2.4}$$

where $|\Omega|$ represents the Lebesgue measure of the domain.

Once given an empirical loss $\boldsymbol{E}_n$, we apply the empirical loss minimization to seek the estimation $u_n$, i.e. $u_n = \arg\min_{u \in \boldsymbol{F}} \boldsymbol{E}_n(u)$, where $\boldsymbol{F}$ is the parametrized hypothesis function space we consider. Some examples can be reproducing kernel Hilbert space (Chen et al., 2021b) and tensor training format (Richter et al., 2021; Chen et al., 2021a). In this paper, we consider sparse neural network and truncated Fourier basis, which can achieve min-max optimal estimation rate for the non-parametric function estimation (Tsybakov, 2008; Schmidt-Hieber et al., 2020; Farrell et al., 2021; Suzuki, 2018; Chen et al., 2019b; Jiao et al., 2021c; Nitanda & Suzuki, 2020).

**Sparse Neural Network Function Space** In this paper, the hypothesis function space $\boldsymbol{F}$ is expressed by the neural network function space following (Schmidt-Hieber et al., 2020; Suzuki, 2018; Farrell et al., 2021). Let us denote the ReLU[3] activation by $\eta_3(x) = \max\{x^3, 0\}$ $(x \in \mathbb{R})$, which is used in (E & Yu, 2018). For a vector $x$, $\eta(x)$ is operated in an element-wise manner. Define the space of all neural networks with height $L$, width $W$, sparsity constraint $S$ and norm constraint $B$ as

$$\Phi(L, W, S, B) := \{(\mathcal{W}^{(L)}\eta_3(\cdot) + b^{(L)}) \circ \cdots (\mathcal{W}^{(2)}\eta_3(\cdot) + b^{(2)}) \circ (\mathcal{W}^{(1)}x + b^{(1)}) \mid$$
$$\mathcal{W}^{(L)} \in \mathbb{R}^{1 \times W}, b^{(L)} \in \mathbb{R}, \mathcal{W}^{(1)} \in \mathbb{R}^{W \times d}, b^{(1)} \in \mathbb{R}^W, \mathcal{W}^{(l)} \in \mathbb{R}^{W \times W}, b^{(l)} \in \mathbb{R}^W (1 < l < L),$$
$$\sum_{l=1}^L(\|\mathcal{W}^{(l)}\|_0 + \|b^{(l)}\|_0) \leq S, \max_l \|\mathcal{W}^{(l)}\|_{\infty,\infty} \vee \|b^{(l)}\|_{\infty} \leq B\}, \tag{2.5}$$

where $\circ$ denotes the function composition, $\|\cdot\|_0$ is the $\ell_0$-norm of the matrix (the number of non-zero elements of the matrix) and $\|\cdot\|_{\infty,\infty}$ is the $\ell_{\infty}$-norm of the matrix (maximum of the absolute values of the elements).

**Truncated Fourier Basis Estimator** We also consider the Truncated Fourier basis as our estimator. Denote the domain we are interested in by $\Omega \subseteq [0,1]^d$. For any $z \in \mathbb{N}^d$, we consider the corresponding Fourier basis function $\phi_z(x) := e^{2\pi i \langle z, x \rangle}$ $(x \in \Omega)$. Any function $f \in L^2(\Omega)$ can be represented as a weighted sum of the Fourier basis $f(x) := \sum_{z \in \mathbb{N}^d} f_z \phi_z(x)$, where $f_z := \int_{\Omega} f(x)\overline{\phi_z(x)}dx$ $(\forall z \in \mathbb{N}^d)$ is the Fourier coefficient. This inspires us to use the Fourier basis whose index lies in a truncated set $Z_{\xi} = \{z \in \mathcal{Z} \mid \|z\|_{\infty} \leq \xi\}$ to represent the function class $\boldsymbol{F}$ as $\boldsymbol{F}_{\xi} = \{\sum_{\|z\|_{\infty} \leq \xi} a_z \phi_z \mid a_z \in \mathbb{R}, \|z\|_{\infty} \leq \xi\}$.

## 3 LOWER BOUND

In this section, we aim to consider the statistical limit of learning the solution of a PDE. As discussed in Propositions 2.1 and 2.2, we directly consider the $H^1$ norm for DRM and $H^2$ norm for PINN as the evaluation metric. The lower bounds are shown as follows.

**Theorem 3.1** (Lower bound). *We denote $u^*(f)$ to be the solution of the PDE 2.1 and we can access randomly sampled data $\{X_i, f_i\}_{i=1,\cdots,n}$ as described in Section 2.2. We further assume $u^*(f) \in H^s$ for a given $s \in \mathbb{Z}^+$. Then we have the following lower bounds.*

**DRM Lower Bound.** *For all estimators $\psi : (\mathbb{R}^d)^{\otimes n} \times \mathbb{R}^{\otimes n} \to H^s(\Omega)$, we have*

$$\inf_\psi \sup_{u^* \in H^s(\Omega)} \mathbb{E}\|\psi(\{X_i, f_i\}_{i=1,\cdots,n}) - u^*(f)\|_{H^1}^2 \gtrsim n^{-\frac{2s-2}{d+2s-4}}. \tag{3.1}$$

**PINN Lower Bound.** *For all estimators $\psi : (\mathbb{R}^d)^{\otimes n} \times \mathbb{R}^{\otimes n} \to H^s(\Omega)$, we have*

$$\inf_\psi \sup_{u^* \in H^s(\Omega)} \mathbb{E}\|\psi(\{X_i, f_i\}_{i=1,\cdots,n}) - u^*(f)\|_{H^2}^2 \gtrsim n^{-\frac{2s-4}{d+2s-4}}. \tag{3.2}$$

Given that $n^{-\frac{2(\beta-k)}{d+2\beta}}$ is the minimax rate of estimating the $k$-th derivative of a $\beta$-smooth density in $L^2$ (Liu & Wang, 2012; Prakasa Rao, 1996; Müller & Gasser, 1979), the lower bound obtained here is the rate of estimating the right hand side function $f$ in terms of the $H^{-1}$ norm. Given the $H^{-1}$ norm error estimate on $f$, we can achieve an estimate of $u$, which provides an alternative way to understand our upper bound. (See discussion in Appendix E.) The lower bound is non-standard, for the $2s - 2$ in the numerator is different from the $2s - 4$ in the denominator.

## 4 UPPER BOUND

To theoretically understand the empirical success of Physics-Informed Neural Networks and the Deep Ritz solver, in this section, we aim to prove that the excess risk $\Delta \boldsymbol{E}_n := \boldsymbol{E}(u_n) - \boldsymbol{E}(u^*)$ of a well-trained neural networks on the PINN/DRM loss function will follow precise power-law scaling relations with the size of the training dataset. Similar to (Xu, 2020; Lu et al., 2021b; Duan et al., 2021; Jiao et al., 2021a;b), we decompose the excess risk into approximation error and generalization error. Different from the concurrent bound (Duan et al., 2021; Jiao et al., 2021a), we provide a fast rate $O(1/n)$ by utilizing the strong convexity of the objective function established in Section 2.1 and achieve a faster and near optimal upper bound. We show that the generalization error can be bounded by the fixed point (*i.e.* the solution of $\phi(r) = r$) of the local Rademacher complexity

$$\psi(r) = R_n(\{\mathcal{I}(u) \mid \|u - u^*\|_A^2 \le r\}),$$

where $R_n$ is the Rademacher complexity, $\mathcal{I}(u) = \Delta u + Vu, \| \cdot \|_A = \| \cdot \|_{H^2}$ for PINN and $\mathcal{I}(u) = \|\nabla u\|^2 + Vu, \| \cdot \|_A = \| \cdot \|_{H^1}$ for DRM. We put detailed definition and analysis in Appendix B.4. We first provide a meta theorem to decompose the error into approximation and a fast rate generalization error. Then we plug in the approximation and generalization error calculated in Appendix B.3 and Appendix B.2 and finally achieve the following upper bounds. We also put a more detailed proof sketch in Appendix A.2.

**Physics Informed Neural Network.**

**Theorem 4.1.** *(Informal Upper Bound of PINN with Deep Neural Network Estimator) With proper assumptions, consider the sparse Deep Neural Network function space $\Phi(L, W, S, B)$ with parameters $L = O(1)$, $W = O(n^{\frac{d}{d+2s-4}})$, $S = O(n^{\frac{d}{d+2s-4}})$, $B = O(n^{\frac{d}{d+2s-4}})$, then the Physics Informed estimator $\hat{u}_{PINN}^{DNN} = \min_{u \in \Phi(L,W,S,B)} \boldsymbol{E}_n^{PINN}(u)$ satisfies the following upper bound with high probability*

$$\|\hat{u}_{PINN}^{DNN} - u^*\|_{H^2}^2 \lesssim n^{-\frac{2s-4}{d+2s-4}} \log n.$$

**Theorem 4.2.** *(Informal Upper Bound of PINN with Truncated Fourier Series Estimator) With proper assumptions, consider the Physics Informed Neural Network objective with a plug-in Fourier Series estimator $\hat{u}_{PINN}^{Fourier} = \min_{u \in \boldsymbol{F}_\xi(\Omega)} \boldsymbol{E}_n^{PINN}(u)$ with $\xi = \Theta(n^{\frac{1}{d+2s-4}})$, then with high probability we have*

$$\|\hat{u}_{PINN}^{Fourier} - u^*\|_{H^2}^2 \lesssim n^{-\frac{2s-4}{d+2s-4}}.$$

**Deep Ritz Methods.**

**Theorem 4.3.** *(Informal Upper Bound of DRM with Deep Neural Network Estimator) With proper assumptions, consider the sparse Deep Neural Network function space $\Phi(L, W, S, B)$ with parameters $L = O(1)$, $W = O(n^{\frac{d}{d+2s-2}})$, $S = O(n^{\frac{d}{d+2s-2}})$, $B = O(n^{\frac{d}{d+2s-2}})$, then the Deep ritz estimator $\hat{u}_{DRM}^{DNN} = \min_{u \in \Phi(L,W,S,B)} \boldsymbol{E}_n^{DRM}(u)$ satisfies the following upper bound with high probability*

$$\|\hat{u}_{DRM}^{DNN} - u^*\|_{H^1}^2 \lesssim n^{-\frac{2s-2}{d+2s-2}} \log n.$$

**Theorem 4.4.** *(Informal Upper Bound of DRM with Truncated Fourier Series Estimator) With proper assumptions, consider the Deep Ritz objective with a plug in Fourier Series estimator $\hat{u}_{DRM}^{Fourier} = \min_{u \in \boldsymbol{F}_\xi(\Omega)} \boldsymbol{E}_n^{DRM}(u)$ with $\xi = \Theta(n^{\frac{1}{d+2s-2}})$, then with high probability we have*

$$\|\hat{u}_{DRM}^{Fourier} - u^*\|_{H^1}^2 \lesssim n^{-\frac{2s-2}{d+2s-2}}.$$

**Remark.**

- There is a common belief that Machine learning based PDE solvers can break the curse of dimensionality (E & Yu, 2018; Grohs et al., 2018; Lanthaler et al., 2021). However, we obtained an $n^{-\frac{2s-2}{2s-4+d}}$ convergence rate, which can become super slow in high dimension. Our analysis showed that it is essential to constrain the function space to break the curse of dimensionality. (Lu et al., 2021b) considered the DRM in Barron spaces. (Ongie et al., 2019) showed that functions in the Barron space enjoy a smoothness $s$ at the same magnitude as $d$, which will also lead to convergence rate independent of the dimension using our upper bound. Neural network can also approximate mixed sparse grid spaces (Montanelli & Du, 2019; Suzuki, 2018) and functions on manifold (Nitanda & Suzuki, 2020; Chen et al., 2019b) without curse of dimensionality. Combined with these approximation bounds, we can also achieve a bound that breaks the curse of dimensionality using Theorem B.12 and B.9. In this paper, we aim to consider the statistical power of the loss function in common function spaces and leave the curse of dimensionality as a separate topic.

- Our bound is faster than the concurrent bound (Duan et al., 2021; Jiao et al., 2021a) for we provided a fast rate $O(1/n)$ by utilizing the strong convexity of the objective function and improved the convergence rate from $n^{-\frac{2s-2}{d+4s-4}}$ to $n^{-\frac{2s-2}{d+2s-2}}$ for Deep Ritz and from $n^{-\frac{2s-4}{d+4s-8}}$ to $n^{-\frac{2s-4}{d+2s-4}}$ for PINN. Compared to the lower bound provided in Section 3, our bounds for PINN is near-optimal while the upper bound for DRM is sub-optimal. We believe our bound is tight and put the discussion in Appendix E. We'll propose a modified version of DRM to match the upper and lower bound in the next section.

- For upper bound of DRM, due to a technical issue, we assumed that the observation we access is clean, *i.e* $f_i = f(X_i)$. We conjecture that add noising on observation will not effect the rate and leave this to future work.

## 5 MODIFIED DEEP RITZ METHODS

Comparing the lower bound in Section 3 and the upper bound in Section 4, we find out that the Physics-Informed Neural Network achieves min-max optimality while the Deep Ritz Method does not. In this section, we propose a modified version of Deep Ritz which can be statistically optimal.

As discussed in Appendix E, the reason behind the suboptimality of DRM comes from the high complexity introduced via the uniform concentration bound of the gradient term in the variational form. At the same time, we further observed that the $\int \|\nabla u\|^2 dx$ does not require any query from the right hand side function $f$, which means that we can easily make another splitted sample to approximate the $\int \|\nabla u\|^2 dx$ term more precisely.

$$\boldsymbol{E}_{N,n}^{\text{MDRM}}(u) = \frac{1}{N} \sum_{j=1}^N \left[ |\Omega| \cdot \frac{1}{2} \|\nabla u(X_j')\|^2 \right] + \frac{1}{n} \sum_{j=1}^n \left[ |\Omega| \cdot \left( \frac{1}{2} V(X_j) |u(X_j)|^2 - f_j u(X_j) \right) \right]$$

$$(5.1)$$

Once we sampled more data for approximating $\int |\nabla u|^2 dx$, we can achieve an near optimal bound for the Truncated Fourier Estimator when $\frac{N}{n} \gtrsim n^{\frac{2}{d+2s-4}}$.

**Theorem 5.1.** *(Informal Upper Bound of DRM with Truncated Fourier Series Estimator)With proper assumptions, consider the Deep Ritz objective with a plug in Fourier Series estimator $\hat{u}_{MDRM}^{Fourier} = \min_{u \in \boldsymbol{F}_\xi(\Omega)} \boldsymbol{E}_{N,n}^{MDRM}(u)$ with $\xi = \Theta(n^{\frac{1}{d+2s-4}})$ and $\frac{N}{n} \gtrsim n^{\frac{2}{d+2s-4}}$, then we have*

$$\|\hat{u}_{MDRM}^{Fourier} - u^*\|_{H^1}^2 \lesssim n^{-\frac{2s-2}{d+2s-4}}.$$

**Remark.** We still cannot achieve optimal rate for neural network even with modified DRM methods. The reason is because the number of neurons is not a good complexity measure for the gradient of the function. Thus, the bound for $\psi(r) = R_n(\{\mathcal{I}(u) \mid \|u - u^*\|_{H^1}^2 \leq r\})$ is not enough for achieving optimal convergence rate. However, following (Schmidt-Hieber et al., 2020; Suzuki, 2018; Imaizumi & Fukumizu, 2020; Chen et al., 2019b; Farrell et al., 2021) to use deep networks for estimating functions, we optimize the best neural network with constrained sparsity in our paper. Here we conjecture that there exists a computable complexity measure that can make DRM statistically optimal and leave finding the right complexity of the neural network's gradient to be future work.

## 6 Experiments

In this section, we conduct several numerical experiments to verify our theory. We follow the neural network and hyper-parameter setting in (Chen et al., 2020). Due to the page limit, we only put the experiments for Deep Ritz Methods here.

### 6.1 The Modified Deep Ritz Methods

In this section, we conduct experiments which substantiate our theoretical results for modified Deep Ritz methods. For simplicity, we take $V(x) = 1$ in our experiment. We conduct experiment in 2-dimension and select the solution of the PDE as $u^* = \sum_z \|z\|^{-s}\phi_z(x) \in H^s$. We show the log-log plot of the $H^1$ loss against the number of sampled data for $s = 4$ in Figure 1. We use an OLS estimator to fit the log-log plot and put the estimated slope and corresponding $R^2$ score in Figure 1. As our theory predicts, the modified Deep Ritz Method converges faster than the original one. All the derivation of the two estimators is listed in Appendix E.

### 6.2 Dimension Dependent Scaling Law.

We conduct experiments to illustrate that the population loss of well-trained and well-tuned Deep Ritz method will scale with the $d$-dimensional training data number $N$ as a power-law $\mathcal{L} \propto \frac{1}{N^\alpha}$. We also scan over a range of $d$ and $\alpha$ and verify an approximately $\alpha \propto \frac{1}{d}$ scaling law as our theory suggests. We use the same test function in Section 6.1 as the solution of our PDE. For simplicity, we take $V(x) = 1$ in our experiment. We train the deep Ritz method on 20, 80, 320, 1280, 10240 sampled data points for 5,6,7,8,9,10 dimensional problems and we plot our results on the log-log scale. Results are shown in Figure 2. We discover the $L \propto n^{\frac{1}{d+2}}$ scaling law in practical situations.

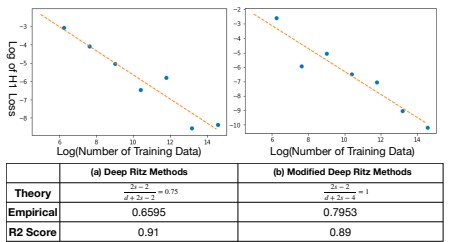

| | (a) Deep Ritz Methods | (b) Modified Deep Ritz Methods |
|---|---|---|
| Theory | $\frac{2s-2}{d+2s-2} = 0.75$ | $\frac{2s-2}{d+2s-4} = 1$ |
| Empirical | 0.6595 | 0.7953 |
| R2 Score | 0.91 | 0.89 |

Figure 1: The log-log plot and estimated convergence slope for Modified DRM and DRM using Fourier basis, showing the median error over 5 replicates.

### 6.3 Adaptation To The Simpler Functions.

(Sharma & Kaplan, 2020) showed that the neural scaling law will adapt to the structure that the target function enjoys. This adaptivity enables the neural network to break the cure of the dimensionality for simple functions in high dimension. (Suzuki & Nitanda, 2019; Chen et al., 2019a) also observed this theoretically. For solving PDEs, we also observed this adaptivity in practice. Here we tested the following two hypothesis

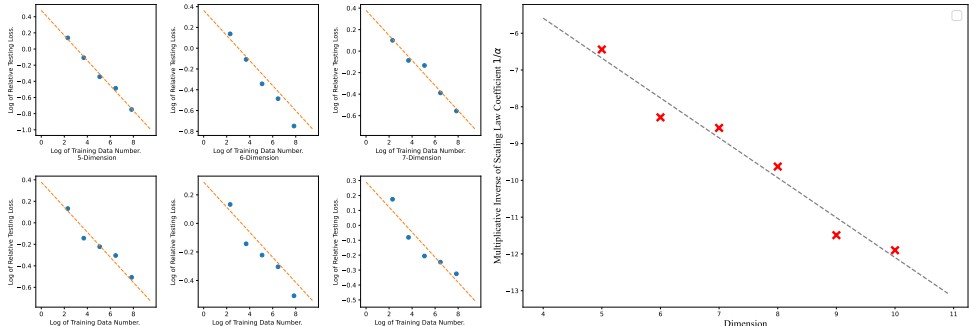

Figure 2: We verify the dimension dependent scaling law empirically. The multiplicative inverse of the scaling law coefficient is highly linear with the dimension $d$, showing the mean error over 2 replicates.

- **Random Neural Network Teacher.** Following (Sharma & Kaplan, 2020), we also tested random neural network using He initialization (He et al., 2015) as the ground turth solution $u^*$. (De Palma et al., 2018) showed that random deep neural networks are biased towards simple functions and in practice we observed a scaling law at the parametric rate. Specifically, we obtained a linear estimate with slope $\alpha = -0.50679429$ and a $R^2$ score $= 0.96$ in the log-log plot. See Figure 3(a).

- **Simple Polynomials.** Neural network can approximate simple polynomials exponentially fast (Wang et al., 2018). Thus, we select the ground truth solution to be the following simple polynomial in 10 dimensional spaces $u^*(x) = x_1 x_2 + \cdots + x_9 x_{10}$. In this example, we obtained a linear estimate with slope $\alpha = -0.49755418$ and a $R^2$ score $= 0.99$ in the log-log plot. See Figure 3(b).

## 7 CONCLUSION AND DISCUSSION

**Conclusion** In this paper, We considered the statistical min-max optimality of solving a PDE from random samples. We improved the previous bounds (Xu, 2020; Lu et al., 2021b; Duan et al., 2021; Jiao et al., 2021a) by providing the first fast rate generalization bound for learning PDE solutions via the strongly convex nature of the two objective functions. We achieved the optimal rate via the PINN and a modified Deep Ritz method. We verified our theory via numerical experiments and explored the dimension dependent scaling laws of Deep PDE solvers.

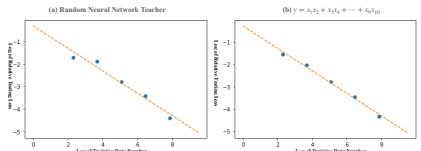

Figure 3: Neural networks have the ability to adapt to simple functions and achieve convergence without curse of dimensionality, showing the median error over 5 replicates.

**Discussion and Future Work** Here we discuss several drawbacks of our theory

- We restricted our target function and estimators in $W^{1,\infty}$ instead of $H^1$ due to boundedness assumption made in the local Rademacher complexity arguments. However, typical functional used in physics is always unbounded, such as the Newtonian potential $\frac{1}{\|x-y\|^{d-2}}$, which limits the application of our theory.

- This paper did not discuss any optimization aspect of the deep PDE solvers and always assumed that global optimum can be achieved. However, it is important to investigate whether the optimization error (Suzuki & Akiyama, 2020; Chizat, 2021) will finally dominate.

- Instead of solving a single PDE, recent works (Long et al., 2018; 2019; Li et al., 2020; Lanthaler et al., 2021; Bhattacharya et al., 2020; Fan & Ying, 2020; Feliu-Faba et al., 2020) considered the so-called "operator learning", which aims to learn a family of PDE/inverse

problems using a single network. It is interesting to investigate the generalization bound and neural scaling law there.

- We find out that the sparsity of the neural network is not a good complexity measure of neural network's gradient. We conjecture that there exists an oracle complexity measure, whose approximation and generalization bounds can lead Modified DRM to achieve the optimal convergence rate.

## ACKNOWLEDGMENTS

Yiping Lu is supported by the Stanford Interdisciplinary Graduate Fellowship (SIGF). Jianfeng Lu is supported in part by National Science Foundation via grants DMS-2012286 and CCF1934964. Lexing Ying is supported by National Science Foundation under award DMS-2011699. Jose Blanchet is supported in part by the Air Force Office of Scientific Research under award number FA9550-20-1-0397 and NSF grants 1915967, 1820942, 1838576. Yiping Lu also thanks Taiji Suzuki, Atsushi Nitanda, Yifan Chen, Junbin Huang, Wenlong Ji, Greg Yang, Yufan Chen, Zong Shang, Denny Wu, Jikai Hou, Jun Hu, Fang Yao, Bin Dong and George Em Karniadakis for helpful comments and feedback.

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

# Appendix

## A   APPENDIX ORGANIZATION AND PROOF SKETCH

### A.1   NOTATIONS

In this section, we provide all the notations we need in the proof. Let $\Omega \subset \mathbb{R}^d$ be some open set. We denote $C(\Omega)$ the space of continuous functions on $\Omega$ and $C^k(\Omega)$ the space of all functions that are $k$ times continuously differentiable on $\Omega$ ($\forall\, k \in \mathbb{Z}^+$). For any $n \in \mathbb{N}_0$ ($\mathbb{N}_0 := \mathbb{Z}^+ \cup \{0\}$ is the set of all non-negative integers) and $1 \le p \le \infty$, we define the Sobolev space $W^{n,p}(\Omega)$ by

$$W^{n,p}(\Omega) := \{f \in L^p(\Omega) : D^\alpha f \in L^p(\Omega),\ \forall \alpha \in \mathbb{N}_0^d \text{ with } |\alpha| \le n\}.$$

In particular, when $p = 2$, we define $H^n(\Omega) := W^{n,2}(\Omega)$ for any $n \in \mathbb{N}_0$. Moreover, for any $f \in W^{n,p}(\Omega)$ with $1 \le p < \infty$, we define the Sobolev norm by:

$$\|f\|_{W^{n,p}(\Omega)} := \Big( \sum_{0 \le |\alpha| \le n} \|D^\alpha f\|_{L^p(\Omega)}^p \Big)^{\frac{1}{p}}.$$

In particular, when $p = \infty$, we have:

$$\|f\|_{W^{n,\infty}(\Omega)} := \max_{0 \le |\alpha| \le n} \|D^\alpha f\|_{L^\infty(\Omega)}.$$

Consider the Fourier expansion $f := \sum_{z \in \mathbb{N}^d} f_z \phi_z(x)$ of the function $f \in W^{n,p}(\Omega)$. We can equivalently express the Sobolev norm as:

$$\|f\|_{W^{n,p}(\Omega)} = \Big( \sum_z \|z\|^{np} |f_z|^p \Big)^{1/p},$$

where $f_z = \int_\Omega f(x)\overline{\phi_z(x)}dx = \int_\Omega f(x)e^{-2\pi i \langle z, x\rangle}dx$ ($x \in \Omega$) is the $z$−th Fourier coefficient of $f$. Moreover, we use $W_0^{1,p}(\Omega)$ to denote the closure of $C_c^1(\Omega)$ in $W^{1,p}(\Omega)$. In particular, when $p = 2$, we define $H_0^1(\Omega) := W_0^{1,2}(\Omega)$.

Furthermore, we use $\|\cdot\|$ to present the vector 2 norm and, given a data sample $\{X_i\}_{i=1}^n \subset \Omega$, $\|\cdot\|_{n,p} = (\mathbb{E}_n \cdot^p)^{1/p}$ denote the empirical $p$ norm, where $\mathbb{E}_n : L^2(\Omega) \to \mathbb{R}$ is the corresponding empirical average operator defined as $\mathbb{E}_n f := \frac{1}{n}\sum_{i=1}^n f(X_i)$, $\forall\, f \in L^2(\Omega)$. Given two quantities $X$ and $Y$, we write $X \lesssim Y$ when the inequality $X \le CY$ holds, where $C$ is some constant. For two functions $f$ and $g$ mapping from $\mathbb{R}^+$ to $\mathbb{R}$, we write $f = O(g)$ when there exist two constants $C'$ and $x_0$ independent of $f$ and $g$, such that the inequality $f(x) \le C'g(x)$ holds for any $x \ge x_0$. We use $X \simeq Y$ to denote $X \lesssim Y$ and $Y \lesssim X$.

### A.2   APPENDIX ORGANIZATION AND PROOF SKETCH

In this section, we list the organization of the appendix and put a more detailed proof sketch of our main results. We put all the proof of upper bounds in Appendix B and all the proof of lower bounds in Appendix D. All the proof of the results about modified Deep Ritz method is given in Appendix C. The proof of lower bounds is based on standard Fano method. In this section, we focus on the proof sketch of the fast rate upper bound.

**Error Decomposition.** We first decompose the excess risk $\Delta E_n := E(u_n) - E(u^*)$ of a well-trained neural network on the PINN/DRM loss function into approximation error and generalization error, similar to (Xu, 2020; Lu et al., 2021b; Duan et al., 2021; Jiao et al., 2021a;b). The regularity results used in the decomposition are proved in Appendix B.1. Explicitly, for any $u_{\boldsymbol{F}} \in \boldsymbol{F}(\Omega)$, we can decompose the excess risk as

$$
\begin{aligned}
\Delta \boldsymbol{E}^{(n)}(\hat{u}) &= \boldsymbol{E}(\hat{u}) - \boldsymbol{E}(u^\star) \\
&= \big[\boldsymbol{E}(\hat{u}) - \boldsymbol{E}_n(\hat{u})\big] + \big[\boldsymbol{E}_n(\hat{u}) - \boldsymbol{E}_n(u_{\boldsymbol{F}})\big] + \big[\boldsymbol{E}_n(u_{\boldsymbol{F}}) - \boldsymbol{E}(u_{\boldsymbol{F}})\big] + \big[\boldsymbol{E}(u_{\boldsymbol{F}}) - \boldsymbol{E}(u^\star)\big] \\
&\le \underbrace{\big[\boldsymbol{E}(\hat{u}) - \boldsymbol{E}_n(\hat{u})\big] + \big[\boldsymbol{E}_n(u_{\boldsymbol{F}}) - \boldsymbol{E}(u_{\boldsymbol{F}})\big]}_{\text{Generalization Error}} + \underbrace{\big[\boldsymbol{E}(u_{\boldsymbol{F}}) - \boldsymbol{E}(u^\star)\big]}_{\text{Approximation Error}},
\end{aligned}
$$

$$(A.1)$$

where the expectation is on uniformly sampled data, $\boldsymbol{F}(\Omega)$ is the space of parametrized estimators we used like truncated Fourier series or sparse neural networks, $\hat{u}$ is the minimizer of the empirical loss $\boldsymbol{E}_n$ in $\boldsymbol{F}(\Omega)$ and $u^*$ is the minimizer of the population loss $\boldsymbol{E}$ (i.e, ground truth solution). The inequality in the third line follows from the fact that $\hat{u}$ is the minimizer of the empirical loss $\boldsymbol{E}_n$ in the space $\boldsymbol{F}(\Omega)$, which implies $\boldsymbol{E}_n(\hat{u}) \leq \boldsymbol{E}_n(u_{\boldsymbol{F}})$. We call the first term generalization error as it's measuring the difference between $\boldsymbol{E}_n$ and $\boldsymbol{E}$. We call the second term approximation error as it seeks for a parametrized estimator $u_{\boldsymbol{F}}$ that approximates the ground truth solution $u^*$ well in $\boldsymbol{F}(\Omega)$. The upper bounds on generalization and approximation error that we achieved in this paper are listed in Table 2.

Let $n$ denote the number of sampled datapoints. For the generalization error, different from the concurrent upper bound $O(\frac{1}{\sqrt{n}})$ (Duan et al., 2021; Jiao et al., 2021a), we provide a faster and near optimal upper bound $O(\frac{1}{n})$ by utilizing the strong convexity of the objective function established in Appendix B.1. Via using the Peeling Lemma (Lemma B.4, for completeness, we also provide a proof), we show that the generalization error can be bounded by the fixed point of the local Rademacher complexity

$$\phi(r) = R_n(\{\mathcal{I}(u) \mid \|u - u^*\|_A^2 \leq r\}),$$

where $R_n$ is the Rademacher complexity, $\mathcal{I}(u) = \Delta u + Vu, \|\cdot\|_A = \|\cdot\|_{H^2}$ for PINN and $\mathcal{I}(u) = \|\nabla u\|^2 + Vu, \|\cdot\|_A = \|\cdot\|_{H^1}$ for DRM. Once we show that $\phi(r)$ is of magnitude $O(\sqrt{\frac{r}{n}})$, we can achieve the $O(\frac{1}{n})$ convergence rate via solving the fix point equation $\phi(r) = O(\sqrt{\frac{r}{n}}) = r \Rightarrow r = O(\frac{1}{n})$. Using the solution of the fixed point equation of the local Rademacher complexity to bound the generalization error is a standard result in empirical process Bartlett et al. (2005); Srebro et al. (2010); Koltchinskii (2011); Xu & Zeevi (2020); Farrell et al. (2021). The difference is that we used the $H^1/H^2$ norm to define the localized set, while the previous papers used the $\ell_2$ distance. The way to obtain the fast rate generalization bound is using the Peeling Lemma.

We present the error decomposition results as a meta theorem, which is shown in Theorem B.12 for PINN, Theorem B.9 for DRM and Theorem C.1 for MDRM, respectively. To make the final rate depend on the data number only, we need bounds of the approximation error in Appendix B.3 and bounds of the local Rademacher complexity in Appendix B.2.

**Approximation Error.** The proof of the approximation results of truncated Fourier series is easy and intuitive. For completeness, we provide it in Appendix B.3.1. The proof of the approximation results of neural networks follows from the fact that a B-spline approximation can be formulated as a ReLU3 neural network efficiently. Our proof basically follows Duan et al. (2021); Jiao et al. (2021a), while the only difference is the activation function. Our proof is also very similar to (Yarotsky, 2017; Suzuki, 2018), but the depth of our network is of constant magnitude instead of $O(\frac{1}{\log \epsilon})$ magnitude, where $\epsilon$ denotes the desired approximation error. Such improvement of depth results from the fact that ReLU3 activations can approximate B-splines more easily than the ReLU activations, which is useful in our generalization analysis. Although the proof of the approximation results of neural networks in the Sobolev space is standard, we still list it in Appendix B.3.2.

**Generalization Error.** As we discussed above, the generalization error can be bounded by the fix point of the local Rademacher complexity, *i.e.* the solution of $\phi(r) = r$. Once we have a $O(\sqrt{\frac{r}{n}})$ bound of $\phi(r)$, we can achieve the $O(\frac{1}{n})$ fast rate generalization bound we want. It remains to upper bound the the local Rademacher complexity $\phi(r)$.

For the upper bound on the local Rademacher complexity of truncated Fourier series estimators, our proof technique is similar to that of the kernel estimators, whose Rademacher complexity can be bounded by the trace of the Gram matrix (*i.e.* the effective number of basis). One interesting thing we showed is that the final upper bound of the Rademacher complexity localized by $H^1$ norm is $\sqrt{\frac{\xi^{d-2}r}{n}}$. The term $\xi^{d-2}$ in the numerator is smaller than $\xi^d$, which is the exact number of Fourier basis. This improvement results from the $H^1$ norm localization. The detailed proof is given in Lemma B.6, Lemma B.7 and Lemma B.8.

For the upper bound on the local Rademacher complexity bound for neural network, we follow Schmidt-Hieber et al. (2020); Suzuki (2018); Farrell et al. (2021) to use a Dudley integral theorem and a covering number argument. The covering number arguments are shown in Theorem B.4, Theorem

B.5 and Theorem B.6. The final local Rademacher complexity bounds are given in Lemma B.17 and Lemma B.18. The difference is that the complexity of gradient of ReLU3 activation function makes the covering number depend exponentially on the neural network's depth. However, the improvement of neural network's depth to constant magnitude mentioned above in the approximation results saves this problem. One drawback of our proof is that the $H^1$ norm localization wouldn't improve the bound for Rademacher complexity and leads to sub-optimal upper bounds. We hypothesize that our bound is tight for sparse neural network and put seeking a right complexity measure of neural network for solving PDEs as a future work.

| Objective Function | Estimator | Approximation | Generalization | Complexity Measure |
|---|---|---|---|---|
| PINNs | Neural Network | $N^{-\frac{2s-4}{d}}$ | $\frac{N}{n}$ | $N$: Number of parameters |
| | Fourier Seriers | $\xi^{-2(s-2)}$ | $\frac{\xi^d}{n}$ | $\xi$:maximum frequency |
| DRM | Neural Network | $N^{-\frac{2s-2}{d}}$ | $\frac{N}{n}$ | $N$: Number of parameters |
| | Fourier Seriers | $\xi^{-2(s-1)}$ | $\frac{\xi^d}{n}$ | $\xi$:maximum frequency |
| MDRM | Neural Network | $N^{-\frac{2s-2}{d}}$ | $\frac{N}{n}$ | $N$: Number of parameters |
| | Fourier Seriers | $\xi^{-2(s-1)}$ | $\frac{\xi^{d-2}}{n}$ | $\xi$:maximum frequency |

Table 2: Approximation and generalization results we achieved in this paper.

# B PROOF OF THE UPPER BOUNDS

## B.1 REGULARITY RESULT FOR THE PDE MODEL.

**Regularity Results of the DRM Objective Function**

**Theorem B.1.** *We consider the static Schrödinger equation on the unit hypercube on $\mathbb{R}^d$ with the zero Direchlet boundary condition:*

$$-\Delta u + Vu = f \text{ on } \Omega,$$
$$u = 0 \text{ on } \partial\Omega. \tag{B.1}$$

*where $f \in L^2(\Omega)$ and $V \in L^\infty(\Omega)$ with $0 < V_{\min} \leq V(x) \leq V_{\max} > 0$. There exists a unique weak solution $u_S^*$ to the equivalent variational problem (Evans, 1998):*

$$u_S^* = \underset{u \in H_0^1(\Omega)}{\arg\min} \boldsymbol{E}^{DRM}(u) := \underset{u \in H_0^1(\Omega)}{\arg\min} \left\{ \frac{1}{2} \int_\Omega \left[ \|\nabla u\|^2 + V|u|^2 \right] dx - \int_\Omega fu dx \right\}. \tag{B.2}$$

*Then for any $u \in H^1(\Omega)$, we have:*

$$\frac{\min(1, V_{\min})}{2} \|u - u_S^*\|_{H^1(\Omega)}^2 \leq \boldsymbol{E}^{DRM}(u) - \boldsymbol{E}^{DRM}(u_S^*) \leq \frac{\max(1, V_{\max})}{2} \|u - u_S^*\|_{H^1(\Omega)}^2. \tag{B.3}$$

*Proof.* To show that $u_S^*$ satisfies estimate B.3, we first claim that for any $u \in H^1(\Omega)$,

$$\boldsymbol{E}^{\mathrm{DRM}}(u) - \boldsymbol{E}^{\mathrm{DRM}}(u_S^*) = \frac{1}{2} \int_\Omega \|\nabla u - \nabla u_S^*\|^2 dx + \frac{1}{2} \int_\Omega V(u_S^* - u)^2 \, dx. \tag{B.4}$$

In fact, by plugging in the first equation of B.1, one has that

$$\boldsymbol{E}^{\mathrm{DRM}}(u_S^*) = \frac{1}{2} \int_\Omega \|\nabla u_S^*\|^2 dx + \frac{1}{2} \int_\Omega V|u_S^*|^2 dx - \int_\Omega fu_S^* dx$$
$$= \frac{1}{2} \int_\Omega \|\nabla u_S^*\|^2 dx + \frac{1}{2} \int_\Omega V|u_S^*|^2 dx + \int_\Omega (\Delta u_S^* - Vu_S^*)u_S^* dx$$
$$= \frac{1}{2} \int_\Omega \|\nabla u_S^*\|^2 dx + \int_\Omega (\Delta u_S^*)u_S^* dx - \frac{1}{2} \int_\Omega V|u_S^*|^2 dx.$$

Furthermore, applying Green's formula to the true solution $u_S^*$ yields:

$$
\begin{aligned}
\boldsymbol{E}^{\text{DRM}}(u_S^*) &= \frac{1}{2}\int_\Omega \|\nabla u_S^*\|^2 dx + \int_\Omega (\Delta u_S^*)u_S^* dx - \frac{1}{2}\int_\Omega V|u_S^*|^2 dx \\
&= \int_{\partial\Omega} \frac{\partial u_S^*}{\partial n}u_S^* dx - \frac{1}{2}\int_\Omega \|\nabla u_S^*\|^2 dx - \frac{1}{2}\int_\Omega V|u_S^*|^2 dx \\
&= -\frac{1}{2}\int_\Omega \|\nabla u_S^*\|^2 dx - \frac{1}{2}\int_\Omega V|u_S^*|^2 dx,
\end{aligned}
$$

where the last identity above follows from the second equality in B.1. Now for any $u \in H^1(\Omega)$, applying Green's formula to $u$ and the true solution $u_S^*$ implies:

$$
\begin{aligned}
\boldsymbol{E}^{\text{DRM}}(u) - \boldsymbol{E}^{\text{DRM}}(u_S^*) &= \frac{1}{2}\int_\Omega \|\nabla u\|^2 dx + \frac{1}{2}\int_\Omega V|u|^2 dx - \int_\Omega fu dx + \frac{1}{2}\int_\Omega \|\nabla u_S^*\|^2 dx + \frac{1}{2}\int_\Omega V|u_S^*|^2 dx \\
&= \frac{1}{2}\int_\Omega \|\nabla u\|^2 dx + \frac{1}{2}\int_\Omega V|u|^2 dx + \int_\Omega (\Delta u_S^* - Vu_S^*)u dx + \frac{1}{2}\int_\Omega \|\nabla u_S^*\|^2 dx + \frac{1}{2}\int_\Omega V|u_S^*|^2 dx \\
&= \frac{1}{2}\int_\Omega \|\nabla u\|^2 dx + \int_\Omega (\Delta u_S^*)u dx + \frac{1}{2}\int_\Omega \|\nabla u_S^*\|^2 dx + \frac{1}{2}\int_\Omega V\left(u_S^* - u\right)^2 dx \\
&= \frac{1}{2}\int_\Omega \|\nabla u\|^2 dx + \int_{\partial\Omega} \frac{\partial u_S^*}{\partial n}u dx - \int_\Omega \nabla u_S^* \cdot \nabla u dx + \frac{1}{2}\int_\Omega \|\nabla u_S^*\|^2 dx + \frac{1}{2}\int_\Omega V\left(u_S^* - u\right)^2 dx \\
&= \frac{1}{2}\int_\Omega \|\nabla u - \nabla u_S^*\|^2 dx + \frac{1}{2}\int_\Omega V(u_S^* - u)^2 \, dx,
\end{aligned}
$$

where the last identity above again follows from the second equality in B.1. This completes our proof of identity B.4. Using the assumptions on the potential function $V$ then implies:

$$
\begin{aligned}
\boldsymbol{E}^{\text{DRM}}(u) - \boldsymbol{E}^{\text{DRM}}(u_S^*) &\le \frac{\max(1, V_{\max})}{2}\Big[\int_\Omega \|\nabla u - \nabla u_S^*\|^2 dx + \int_\Omega (u_S^* - u)^2 \, dx\Big] \\
&= \frac{\max(1, V_{\max})}{2}\|u - u_S^*\|_{H^1(\Omega)}^2, \\
\boldsymbol{E}^{\text{DRM}}(u) - \boldsymbol{E}^{\text{DRM}}(u_S^*) &\ge \frac{\min(1, V_{\min})}{2}\Big[\int_\Omega \|\nabla u - \nabla u_S^*\|^2 dx + \int_\Omega (u_S^* - u)^2 \, dx\Big] \\
&= \frac{\min(1, V_{\min})}{2}\|u - u_S^*\|_{H^1(\Omega)}^2.
\end{aligned}
$$

This completes our proof of B.1. $\qquad\square$

### Regularity Results of the PINN Objective Function

**Theorem B.2.** *We consider the static Schrödinger equation on the unit hypercube on $\mathbb{R}^d$ with the Neumann boundary condition:*

$$
\begin{aligned}
-\Delta u + Vu &= f \text{ on } \Omega, \\
u &= 0 \text{ on } \partial\Omega.
\end{aligned}
\tag{B.5}
$$

*where $f \in L^2(\Omega)$ and $V \in L^\infty(\Omega)$ with $V^2 - \Delta V > C_{\min}, 0 < C_{\min} < V(x) \le V_{\max}$ and $-\Delta V(x) \le V_{\max}$. Then there exists a unique solution $u_S^* \in H_0^1(\Omega)$ to the following minimization problem (Brezis, 2010):*

$$
u_S^* = \underset{u \in H_0^1(\Omega)}{\arg\min} \boldsymbol{E}^{PINN}(u) := \underset{u \in H_0^1(\Omega)}{\arg\min} \left\{ \int_\Omega |\Delta u - Vu + f|^2 dx \right\}.
\tag{B.6}
$$

*Then for any $u \in H_0^1(\Omega)$, we have:*

$$
\min\{1, C_{\min}\}\|u - u_S^*\|_{H^2(\Omega)}^2 \le \boldsymbol{E}^{PINN}(u) - \boldsymbol{E}^{PINN}(u_S^*) \le 2(1 + V_{\max} + V_{\max}^2)\|u - u_S^*\|_{H^2(\Omega)}^2.
\tag{B.7}
$$

*Proof.* For any $u \in H_0^1(\Omega)$, we let $\tilde{u} = u - u^*$, then we have $\tilde{u} \in H_0^1(\Omega)$.

$$
\begin{aligned}
\boldsymbol{E}^{\text{PINN}}(u) - \boldsymbol{E}^{\text{PINN}}(u_S^*) &= \int_\Omega |\Delta u - Vu - \Delta u^* + Vu^*|^2 dx = \int_\Omega |\Delta\tilde{u} - V\tilde{u}|^2 dx \\
&= \int_\Omega (\Delta\tilde{u})^2 dx + \int_\Omega V^2\tilde{u}^2 dx - 2\int_\Omega V\tilde{u}\Delta\tilde{u}dx.
\end{aligned}
\tag{B.8}
$$

Using Green's formula, we have:

$$
\int_\Omega V\tilde{u}\Delta\tilde{u}dx + \int_\Omega \nabla(V\tilde{u}) \cdot \nabla\tilde{u}dx = \int_{\partial\Omega} \frac{\partial\tilde{u}}{\partial n} V\tilde{u}ds = 0,
$$

where the last equality above follows from the fact that $\tilde{u} \in H_0^1(\Omega)$. This further implies:

$$
\begin{aligned}
\boldsymbol{E}^{\text{PINN}}(u) - \boldsymbol{E}^{\text{PINN}}(u_S^*) &= \int_\Omega (\Delta\tilde{u})^2 dx + \int_\Omega V^2\tilde{u}^2 dx + 2\int_\Omega \nabla(V\tilde{u}) \cdot \nabla\tilde{u}dx \\
&= \int_\Omega (\Delta\tilde{u})^2 dx + \int_\Omega V^2\tilde{u}^2 dx + 2\int_\Omega V\|\nabla\tilde{u}\|^2 dx + 2\int_\Omega \tilde{u}\nabla V \cdot \nabla\tilde{u}dx.
\end{aligned}
$$

Using Green's formula again, we have:

$$
2\int_\Omega \tilde{u}\nabla V \cdot \nabla\tilde{u}dx = \int_\Omega \nabla(u^2) \cdot \nabla V dx = \int_{\partial\Omega} \frac{\partial V}{\partial n}\tilde{u}^2 ds - \int_\Omega \tilde{u}^2\Delta V dx = -\int_\Omega \tilde{u}^2\Delta V dx.
$$

Then we can further deduce that:

$$
\boldsymbol{E}^{\text{PINN}}(u) - \boldsymbol{E}^{\text{PINN}}(u_S^*) = \int_\Omega (\Delta\tilde{u})^2 dx + \int_\Omega (V^2 - \Delta V)\tilde{u}^2 dx + 2\int_\Omega V\|\nabla\tilde{u}\|^2 dx.
$$

For we have assumed $V \in L^\infty(\Omega)$ with $0 < C_{\min} < V^2 - \Delta V, 0 < C_{\min} < V(x) \leq V_{\max}$ and $-\Delta V(x) \leq V_{\max}$, thus we have

$$
\min\{1, C_{\min}\}\|u - u_S^*\|_{H^2(\Omega)}^2 \leq \boldsymbol{E}^{\text{PINN}}(u) - \boldsymbol{E}^{\text{PINN}}(u_S^*) \leq 2(1 + V_{\max} + V_{\max}^2)\|u - u_S^*\|_{H^2(\Omega)}^2.
\tag{B.9}
$$

$\square$

## B.2 AUXILIARY DEFINITIONS AND LEMMATA ON GENERALIZATION ERROR

To bound the generalization error, we use the localized Rademacher complexity (Bartlett et al., 2005). Recall that the Rademacher complexity of a function class $\boldsymbol{G}$ is defined by

$$
R_n(\boldsymbol{G}) = \mathbb{E}_Z\mathbb{E}_\sigma\Big[\sup_{g\in\boldsymbol{G}}\Big|\frac{1}{n}\sum_{j=1}^n \sigma_j g(Z_j)\Big| \,\Big|\, Z_1, \cdots, Z_n\Big],
$$

where $Z_i$ are i.i.d samples according to the data distributions and $\sigma_j$ are i.i.d Rademacher random variables which take the value $1$ with probability $\frac{1}{2}$ and value $-1$ with probability $\frac{1}{2}$.

The following important symmetrization lemma makes the connection between the uniform law of large numbers and the Rademacher complexity.

**Lemma B.1** (Symmetrization Lemma). *Let $\boldsymbol{F}$ be a set of functions. Then*

$$
\mathbb{E}\sup_{u\in\boldsymbol{F}}\Big|\frac{1}{n}\sum_{j=1}^n u(X_j) - \mathbb{E}_{X\sim\boldsymbol{P}_\Omega}u(X)\Big| \leq 2R_n(\boldsymbol{F}).
$$

**Lemma B.2** (Ledoux-Talagrand contraction (Ledoux & Talagrand, 2013, Theorem 4.12)). *Assume that $\phi : \mathbb{R}\to\mathbb{R}$ is $L$-Lipschitz with $\phi(0) = 0$. Let $\{\sigma_i\}_{i=1}^n$ be independent Rademacher random variables. Then for any $T \subset \mathbb{R}^n$*

$$
\mathbb{E}_\sigma\Big[\sup_{(t_1,\cdots,t_n)\in T}\sum_{i=1}^n \sigma_i\phi(t_i)\Big] \leq 2L \cdot \mathbb{E}_\sigma\Big[\sup_{(t_1,\cdots,t_n)\in T}\sum_{i=1}^n \sigma_i t_i\Big].
$$

Let $(E, \rho)$ be a metric space with metric $\rho$. A $\delta$-*cover* of a set $A \subset E$ with respect to $\rho$ is a collection of points $\{x_1, \cdots, x_n\} \subset A$ such that for every $x \in A$, there exists $i \in \{1, \cdots, n\}$ such that $\rho(x, x_i) \leq \delta$. The $\delta$-covering number $\boldsymbol{N}(\delta, A, \rho)$ is the cardinality of the smallest $\delta$-cover of the set $A$ with respect to the metric $\rho$. Equivalently, the $\delta$-covering number $\boldsymbol{N}(\delta, A, \rho)$ is the minimal number of balls $B_\rho(x, \delta)$ of radius $\delta$ needed to cover the set $A$.

**Theorem B.3** (Dudley's Integral theorem). *Let $\boldsymbol{F}$ be a function class such that $\sup_{f \in \boldsymbol{F}} \|f\|_{n,2} \leq M$. Then the Rademacher complexity $R_n(\boldsymbol{F})$ satisfies that*

$$R_n(\boldsymbol{F}) \leq \inf_{0 \leq \delta \leq M} \left\{ 4\delta + \frac{12}{\sqrt{n}} \int_\delta^M \sqrt{\log \boldsymbol{N}(\epsilon, \boldsymbol{F}, \|\cdot\|_{n,2})} \, d\epsilon \right\}.$$

**Lemma B.3** (Talagrand Concentration Inequality). *Consider a function class $\mathcal{F}$ defined on a probability measure $\mu$ such that for all $f \in \mathcal{F}$, we have $\|f\|_\infty \leq \beta, \mathbb{E}_\mu[f] = 0, \mathbb{E}_\mu[f^2] \leq \sigma^2$. Then for any $t > 0$, we can have the following concentration results.*

$$\mathbb{P}_{z_1, \cdots, z_n \sim \mu} \left[ \sup_{f \in \mathcal{F}} \frac{1}{n} \sum_{i=1}^n f(z_i) \geq 2 \sup_{f \in \mathcal{F}} \mathbb{E}_{z_1', \cdots, z_n' \sim \mu} \frac{1}{n} \sum_{i=1}^n f(z_i') + \sqrt{\frac{2t\sigma^2}{n}} + \frac{2t\beta}{n} \right] \leq e^{-t}.$$

**Lemma B.4** (Peeling lemma ([Bartlett et al., 2005](#))). *Consider some measurable function class $\mathcal{F}$. Assume that there exists a sub-root function $\phi(r)$ satisfying*

$$R_n(\{f \in \mathcal{F} \mid \mathbb{E}[f] \leq r\}) \leq \phi(r) \ (\forall \, r > 0). \tag{B.10}$$

*Then we have*

$$\mathbb{E}_{\sigma_i, z_n} \left[ \sup_{f \in \mathcal{F}} \frac{\frac{1}{n} \sum_{i=1}^n \sigma_i f(z_i)}{\mathbb{E}[f] + r} \right] \leq \frac{4\phi(r)}{r}.$$

*Proof.* Denote $\mathcal{F}(r) = \{f \in \mathcal{F} \mid \mathbb{E}[f] \leq r\}$ to be the localized set with radius $r$. Then for a fixed set of datapoints $\{z_i\}_{i=1}^n$ and a fixed set of Rademacher random variables $\{\sigma_i\}_{i=1}^n$, we have:

$$\mathbb{E}_{\sigma_i, z_n} \left[ \sup_{f \in \mathcal{F}} \frac{\frac{1}{n} \sum_{i=1}^n \sigma_i f(z_i)}{\mathbb{E}[f] + r} \right] \leq \mathbb{E}_{\sigma_i, z_n} \left[ \sup_{f \in \mathcal{F}(r)} \frac{\frac{1}{n} \sum_{i=1}^n \sigma_i f(z_i)}{r} \right] + \sum_{j=0}^\infty \mathbb{E}_{\sigma_i, z_n} \left[ \sup_{f \in \mathcal{F}(r4^{j+1}) \backslash \mathcal{F}(r4^j)} \frac{\frac{1}{n} \sum_{i=1}^n \sigma_i f(z_i)}{r4^j + r} \right]$$

$$\leq \frac{R_n(\mathcal{F}(r))}{r} + \sum_{j=0}^\infty \frac{R_n(\mathcal{F}(r4^{j+1}))}{r4^j + r} \leq \frac{\phi(r)}{r} + \sum_{j=0}^\infty \frac{\phi(r4^{j+1})}{r4^j + r}$$

$$\leq \frac{\phi(r)}{r} + \sum_{j=0}^\infty \frac{2^{j+1}\phi(r)}{r4^j + r} \leq \frac{4\phi(r)}{r}.$$

$\square$

We also modify the peeling lemma above, as we aim to apply it to derive the upper bound for the Modified Deep Ritz Method (MDRM).

**Lemma B.5** (Peeling Lemma For MDRM). *Given some measurable function class $\mathcal{F}$ and two continuous mappings $g, h : \mathcal{F} \to \mathbb{R}$, we define a class $\mathcal{F}$ of vector-valued functions by:*

$$\mathcal{F} := \{(g \circ f, h \circ f) \mid f \in \boldsymbol{F}\}.$$

*For any $f \in \boldsymbol{F}$, we use $g_f$ and $h_f$ to denote the two compositions $g \circ f$ and $h \circ f$, respectively. For any $r > 0$, the localized set $\mathcal{F}_r$ is defined by:*

$$\mathcal{F}_r = \{(g_f, h_f) \in \mathcal{F} \mid \mathbb{E}_x[g_f(x)] + \mathbb{E}_y[h_f(y)] \leq r\}.$$

*Moreover, the modified Rademacher Complexity of $\mathcal{F}_r$ is defined by:*

$$R_{n,m}(\mathcal{F}_r) := R_n\Big(\{g_f | (g_f, h_f) \in \mathcal{F}_r\}\Big) + R_m(\{h_f | (g_f, h_f) \in \mathcal{F}_r\}\Big).$$

*Assume that there exists some function $\phi : [0, \infty) \to [0, \infty)$ and some $r^\star > 0$, such that for any $r > r^\star$, we have:*

$$\phi(4r) \leq 2\phi(r) \text{ and } R_{n,m}(\mathcal{F}_r) \leq \phi(r).$$

*Then for any $r > r^\star$, we have:*

$$\mathbb{E}_{\sigma, \tau}\Big[\mathbb{E}_{x,y}[\sup_{f \in \boldsymbol{F}} \frac{\frac{1}{n} \sum_{i=1}^n \sigma_i g_f(x_i) + \frac{1}{m} \sum_{j=1}^m \tau_j h_f(y_j)}{\mathbb{E}_x[g_f(x)] + \mathbb{E}_y[h_f(y)] + r}]\Big] \leq \frac{4\phi(r)}{r}.$$

*Proof.* The proof is the same as the original peeling lemma, thus we omit the detailed proof here. $\square$

### B.2.1 Local Rademacher Complexity of Truncated Fourier Basis

**Definition B.1.** *(Fourier Series) Given a domain $\Omega \subseteq [0,1]^d$. For any $z \in \mathbb{N}^d$, we consider the corresponding Fourier basis function $\phi_z(x) := e^{2\pi i \langle z, x \rangle}$ $(x \in \Omega)$. With respect to the Fourier basis, any function $f \in L^2(\Omega)$ can be decomposed as the following sum:*

$$f(x) := \sum_{z \in \mathbb{N}^d} f_z \phi_z(x). \tag{B.11}$$

*where for any $z \in \mathbb{N}^d$, the Fourier coefficient $f_z = \int_\Omega f(x)\overline{\phi_z(x)}dx$.*

**Definition B.2.** *(Truncated Fourier Series) For a fixed positive integer $\xi \in \mathbb{Z}^+$, we define the space $F_\xi(\Omega)$ of truncated Fourier series as follows:*

$$F_\xi(\Omega) := \left\{ f = \sum_{z \in \mathbb{N}^d} f_z \phi_z \,\Big|\, f_z = 0, \ \forall \, \|z\|_\infty > \xi \right\}. \tag{B.12}$$

*Equivalently, we can decompose any $f \in F_\xi(\Omega)$ as $f := \sum_{\|z\|_\infty \le \xi} f_z \phi_z$.*

**Lemma B.6.** *(Local Rademacher Complexity of Localized Truncated Fourier Series) For a fixed $\xi \in \mathbb{Z}^+$, we consider a localized class of functions $\boldsymbol{F}_{\rho,\xi}(\Omega) = \left\{ f \in F_\xi(\Omega) \,\Big|\, \|f\|_{H^1(\Omega)}^2 \le \rho \right\}$, where $\rho > 0$ is fixed. Then we have the following upper bound on the local Rademacher complexity:*

$$R_n(\boldsymbol{F}_{\rho,\xi}(\Omega)) = \mathbb{E}_X \left[ \mathbb{E}_\sigma \left[ \sup_{f \in \boldsymbol{F}_{\rho,\xi}(\Omega)} \frac{1}{n} \sum_{i=1}^n \sigma_i f(X_i) \,\Big|\, X_1, \cdots, X_n \right] \right] \lesssim \sqrt{\frac{\rho}{n}} \xi^{\frac{d-2}{2}}. \tag{B.13}$$

*Proof.* Take an arbitrary function $f \in \boldsymbol{F}_{\rho,\xi}(\Omega)$. Let $f = \sum_{\|z\|_\infty \le \xi} f_z \phi_z$ be the Fourier basis expansion of $f$. $\rho \ge \|f\|_{H^1(\Omega)}^2$ implies constraint $\sum_{\|z\|_\infty \le \xi} |f_z|^2 \|z\|^2 \lesssim \rho$ on the Fourier coefficients(Adams & Fournier, 2003).

On the other hand, substituting the Fourier expansion into the average sum $\frac{1}{n}\sum_{i=1}^n \sigma_i f(X_i)$ and using Cauchy-Schwarz inequality imply:

$$\frac{1}{n}\sum_{i=1}^n \sigma_i f(X_i) = \frac{1}{n}\sum_{i=1}^n \sigma_i \sum_{\|z\|_\infty \le \xi} f_z \phi_z(X_i) = \frac{1}{n} \sum_{\|z\|_\infty \le \xi} \sum_{i=1}^n \sigma_i f_z \phi_z(X_i)$$

$$\le \frac{1}{n} \Big( \sum_{\|z\|_\infty \le \xi} |f_z|^2 \|z\|^2 \Big)^{\frac{1}{2}} \Big( \sum_{\|z\|_\infty \le \xi} \Big| \sum_{i=1}^n \frac{\sigma_i}{\|z\|} \phi_z(X_i) \Big|^2 \Big)^{\frac{1}{2}}$$

$$\lesssim \frac{\sqrt{\rho}}{n} \Big( \sum_{\|z\|_\infty \le \xi} \Big| \sum_{i=1}^n \frac{\sigma_i}{\|z\|} \phi_z(X_i) \Big|^2 \Big)^{\frac{1}{2}}.$$

where we have used the constraint $\sum_{\|z\|_\infty \le \xi} |f_z|^2 \|z\|^2 \lesssim \rho$ in the last step above. Moreover, by taking expectation with respect to the i.i.d Rademacher random variables $\sigma_i$ $(1 \le i \le n)$ and the uniformly sampled data points $\{X_i\}_{i=1}^n$ on both sides and applying Jensen's inequality, we can deduce that:

$$\mathbb{E}_X \mathbb{E}_\sigma \Big[ \frac{1}{n}\sum_{i=1}^n \sigma_i f(X_i) \Big] \lesssim \frac{\sqrt{\rho}}{n} \mathbb{E}_{X,\sigma} \left[ \Big( \sum_{\|z\|_\infty \le \xi} \Big| \sum_{i=1}^n \frac{\sigma_i}{\|z\|} \phi_z(X_i) \Big|^2 \Big)^{\frac{1}{2}} \right]$$

$$\le \frac{\sqrt{\rho}}{n} \left( \mathbb{E}_{X,\sigma} \Big[ \sum_{\|z\|_\infty \le \xi} \Big| \sum_{i=1}^n \frac{\sigma_i}{\|z\|} \phi_z(X_i) \Big|^2 \Big] \right)^{\frac{1}{2}}.$$

Using independence between the random variables $\sigma_i$ $(1 \leq i \leq n)$, we can further simplify the expectation inside the square root above as below:

$$
\begin{aligned}
\mathbb{E}_{X,\sigma}\Big[ \sum_{\|z\|_\infty \leq \xi} \Big| \sum_{i=1}^n \frac{\sigma_i}{\|z\|} \phi_z(X_i) \Big|^2 \Big] &= \sum_{\|z\|_\infty \leq \xi} \mathbb{E}_{X,\sigma}\Big[ \Big| \sum_{i=1}^n \frac{\sigma_i}{\|z\|} \phi_z(X_i) \Big|^2 \Big] \\
&= \sum_{\|z\|_\infty \leq \xi} \sum_{i=1}^n \mathbb{E}_{X,\sigma}\Big[ \frac{\sigma_i^2}{\|z\|^2} \Big| \phi_z(X_i) \Big|^2 \Big] \\
&= \sum_{\|z\|_\infty \leq \xi} \sum_{i=1}^n \frac{|\Omega|}{\|z\|^2} \lesssim n \sum_{\|z\|_\infty \leq \xi} \frac{1}{\|z\|^2} \lesssim n \frac{\xi^d}{\xi^2} = n\xi^{d-2}.
\end{aligned}
$$

Combining the two bounds above yields the desired upper bound:

$$
\mathbb{E}_X\Big[ \mathbb{E}_\sigma\Big[ \sup_{f \in \boldsymbol{F}_{\rho,\xi}(\Omega)} \frac{1}{n} \sum_{i=1}^n \sigma_i f(X_i) \,\Big|\, X_1, \cdots, X_n \Big] \Big] \lesssim \frac{\sqrt{\rho}}{n} \sqrt{n\xi^{d-2}} = \sqrt{\frac{\rho}{n}} \xi^{\frac{d-2}{2}}.
$$

$\square$

**Lemma B.7.** *(Local Rademacher Complexity of Localized Truncated Fourier Series' Gradient) For a fixed $\xi \in \mathbb{Z}^+$, we consider a localized class of functions $\boldsymbol{G}_{\rho,\xi}(\Omega) = \{\|\nabla f\| \mid f \in F_{\rho,\xi}(\Omega)\}$, where $\rho > 0$ is fixed. Then for any sample $\{X_i\}_{i=1}^n \subset \Omega$, we have the following upper bound on the local Rademacher complexity:*

$$
R_n(\boldsymbol{G}_{\rho,\xi}(\Omega)) = \mathbb{E}_X\Big[ \mathbb{E}_\sigma\Big[ \sup_{f \in \boldsymbol{F}_{\rho,\xi}(\Omega)} \frac{1}{n} \sum_{i=1}^n \sigma_i \|\nabla f(X_i)\| \,\Big|\, X_1, \cdots, X_n \Big] \Big] \lesssim \sqrt{\frac{\rho}{n}} \xi^{\frac{d}{2}}. \quad \text{(B.14)}
$$

*Proof.* Take an arbitrary function $f \in \boldsymbol{F}_{\rho,\xi}(\Omega)$. Let $f = \sum_{\|z\|_\infty \leq \xi} f_z \phi_z$ be the Fourier basis expansion of $f$. Similarly, the norm restriction condition $\|f\|_{H^1(\Omega)}^2 \leq \rho$ can be reduced to the following condition about Fourier coefficients:

$$
\sum_{\|z\|_\infty \leq \xi} |f_z|^2 \|z\|^2 \lesssim \rho.
$$

Moreover, substituting the Fourier expansion into the average sum $\frac{1}{n} \sum_{i=1}^n \sigma_i \|\nabla f(X_i)\|$ and using Cauchy-Schwarz inequality imply:

$$
\begin{aligned}
\frac{1}{n} \sum_{i=1}^n \sigma_i \|\nabla f(X_i)\| &= \frac{1}{n} \sum_{i=1}^n \sigma_i \Big\| \sum_{\|z\|_\infty \leq \xi} f_z \nabla \phi_z(X_i) \Big\| \leq \frac{1}{n} \sum_{\|z\|_\infty \leq \xi} \sum_{i=1}^n \sigma_i \|f_z \nabla \phi_z(X_i)\| \\
&\leq \frac{1}{n} \Big( \sum_{\|z\|_\infty \leq \xi} |f_z|^2 \|z\|^2 \Big)^{\frac{1}{2}} \Big( \sum_{\|z\|_\infty \leq \xi} \Big| \sum_{i=1}^n \frac{\sigma_i}{\|z\|} \|\nabla \phi_z(X_i)\| \Big|^2 \Big)^{\frac{1}{2}} \\
&\lesssim \frac{\sqrt{\rho}}{n} \Big( \sum_{\|z\|_\infty \leq \xi} \Big| \sum_{i=1}^n \frac{\sigma_i}{\|z\|} \|\nabla \phi_z(X_i)\| \Big|^2 \Big)^{\frac{1}{2}}.
\end{aligned}
$$

where we have used the constraint $\sum_{\|z\|_\infty \leq \xi} |f_z|^2 \|z\|^2 \lesssim \rho$ in the last step above. Moreover, by taking expectation with respect to the i.i.d Rademacher random variables $\sigma_i$ $(1 \leq i \leq n)$ and the uniformly sampled data points $\{X_i\}_{i=1}^n$ on both sides and applying Jensen's inequality, we can deduce that:

$$
\begin{aligned}
\mathbb{E}_X \mathbb{E}_\sigma\Big[ \frac{1}{n} \sum_{i=1}^n \sigma_i \|\nabla f(X_i)\| \Big] &\lesssim \frac{\sqrt{\rho}}{n} \mathbb{E}_{X,\sigma}\Big[ \Big( \sum_{\|z\|_\infty \leq \xi} \Big| \sum_{i=1}^n \frac{\sigma_i}{\|z\|} \|\nabla \phi_z(X_i)\| \Big|^2 \Big)^{\frac{1}{2}} \Big] \\
&\leq \frac{\sqrt{\rho}}{n} \Big( \mathbb{E}_{X,\sigma}\Big[ \sum_{\|z\|_\infty \leq \xi} \Big| \sum_{i=1}^n \frac{\sigma_i}{\|z\|} \|\nabla \phi_z(X_i)\| \Big|^2 \Big] \Big)^{\frac{1}{2}}.
\end{aligned}
$$

Using independence between the random variables $\sigma_i$ $(1 \leq i \leq n)$, we can further simplify the expectation inside the square root above as below:

$$
\mathbb{E}_{X,\sigma}\Big[\sum_{\|z\|_\infty \leq \xi}\Big|\sum_{i=1}^n \frac{\sigma_i}{\|z\|}\|\nabla\phi_z(X_i)\|\Big|^2\Big] = \sum_{\|z\|_\infty \leq \xi}\mathbb{E}_{X,\sigma}\Big[\Big|\sum_{i=1}^n \frac{\sigma_i}{\|z\|}\|\nabla\phi_z(X_i)\|\Big|^2\Big]
$$

$$
= \sum_{\|z\|_\infty \leq \xi}\sum_{i=1}^n \mathbb{E}_{X,\sigma}\Big[\frac{\sigma_i^2}{\|z\|^2}\|\nabla\phi_z(X_i)\|^2\Big]
$$

$$
= \sum_{\|z\|_\infty \leq \xi}\sum_{i=1}^n |\Omega|\frac{4\pi^2\|z\|^2}{\|z\|^2} \lesssim n \sum_{\|z\|_\infty \leq \xi} 1 \lesssim n\xi^d.
$$

Combining the two bounds above yields the desired upper bound:

$$
\mathbb{E}_X\Big[\mathbb{E}_\sigma\Big[\sup_{f \in \boldsymbol{F}_{\rho,\xi}(\Omega)}\frac{1}{n}\sum_{i=1}^n \sigma_i\|\nabla f(X_i)\| \,\Big|\, X_1,\cdots,X_n\Big]\Big] \lesssim \frac{\sqrt{\rho}}{n}\sqrt{n\xi^d} = \sqrt{\frac{\rho}{n}}\xi^{\frac{d}{2}}.
$$

$\square$

**Lemma B.8.** *(Local Rademacher Complexity of Localized Truncated Fourier Series' Laplacian) For a fixed $\xi \in \mathbb{Z}^+$, we consider a localized class of functions $\boldsymbol{J}_{\rho,\xi}(\Omega) := \Big\{f \in F_\xi(\Omega) \,\Big|\, \|f\|_{H^2(\Omega)}^2 \leq \rho\Big\}$, where $\rho > 0$ is fixed. Correspondingly, we define a localized class of Laplacians $\boldsymbol{K}_{\rho,\xi}(\Omega) := \{\Delta f \mid f \in J_{\rho,\xi}(\Omega)\}$. Then for any sample $\{X_i\}_{i=1}^n \subset \Omega$, we have the following upper bound on the local Rademacher complexity:*

$$
R_n(\boldsymbol{K}_{\rho,\xi}(\Omega)) = \mathbb{E}_X\Big[\mathbb{E}_\sigma\Big[\sup_{f \in \boldsymbol{F}_{\rho,\xi}(\Omega)}\frac{1}{n}\sum_{i=1}^n \sigma_i\Delta f(X_i) \,\Big|\, X_1,\cdots,X_n\Big]\Big] \lesssim \sqrt{\frac{\rho}{n}}\xi^{\frac{d}{2}}. \tag{B.15}
$$

*Proof.* Take an arbitrary function $f \in \boldsymbol{J}_{\rho,\xi}(\Omega)$. Let $f = \sum_{\|z\|_\infty \leq \xi} f_z\phi_z$ be the Fourier basis expansion of $f$. Similarly, the norm restriction condition $\|f\|_{H^2(\Omega)}^2 \leq \rho$ can be reduced to the following condition about Fourier coefficients:

$$
\sum_{\|z\|_\infty \leq \xi} |f_z|^2\|z\|^4 \lesssim \rho.
$$

Moreover, substituting the Fourier expansion into the average sum $\frac{1}{n}\sum_{i=1}^n \sigma_i\Delta f(X_i)$ and using Cauchy-Schwarz inequality imply:

$$
\frac{1}{n}\sum_{i=1}^n \sigma_i\Delta f(X_i) = \frac{1}{n}\sum_{i=1}^n \sigma_i\sum_{\|z\|_\infty \leq \xi} f_z\Delta\phi_z(X_i) = \frac{1}{n}\sum_{\|z\|_\infty \leq \xi}\sum_{i=1}^n \sigma_i f_z\Delta\phi_z(X_i)
$$

$$
\leq \frac{1}{n}\Big(\sum_{\|z\|_\infty \leq \xi} |f_z|^2\|z\|^4\Big)^{\frac{1}{2}}\Big(\sum_{\|z\|_\infty \leq \xi}\Big|\sum_{i=1}^n \frac{\sigma_i}{\|z\|^2}\Delta\phi_z(X_i)\Big|^2\Big)^{\frac{1}{2}}
$$

$$
\lesssim \frac{\sqrt{\rho}}{n}\Big(\sum_{\|z\|_\infty \leq \xi}\Big|\sum_{i=1}^n \frac{\sigma_i}{\|z\|^2}\Delta\phi_z(X_i)\Big|^2\Big)^{\frac{1}{2}}.
$$

where we have used the constraint $\sum_{\|z\|_\infty \leq \xi} |f_z|^2\|z\|^4 \lesssim \rho$ in the last step above. Moreover, by taking expectation with respect to the i.i.d Rademacher random variables $\sigma_i$ $(1 \leq i \leq n)$ and the uniformly sampled data points $\{X_i\}_{i=1}^n$ on both sides and applying Jensen's inequality, we can deduce that:

$$
\mathbb{E}_X\mathbb{E}_\sigma\Big[\frac{1}{n}\sum_{i=1}^n \sigma_i\Delta f(X_i)\Big] \lesssim \frac{\sqrt{\rho}}{n}\mathbb{E}_{X,\sigma}\Big[\Big(\sum_{\|z\|_\infty \leq \xi}\Big|\sum_{i=1}^n \frac{\sigma_i}{\|z\|^2}\Delta\phi_z(X_i)\Big|^2\Big)^{\frac{1}{2}}\Big]
$$

$$
\leq \frac{\sqrt{\rho}}{n}\Big(\mathbb{E}_{X,\sigma}\Big[\sum_{\|z\|_\infty \leq \xi}\Big|\sum_{i=1}^n \frac{\sigma_i}{\|z\|^2}\Delta\phi_z(X_i)\Big|^2\Big]\Big)^{\frac{1}{2}}.
$$

Using independence between the random variables $\sigma_i$ $(1 \leq i \leq n)$, we can further simplify the expectation inside the square root above as below:

$$
\mathbb{E}_{X,\sigma}\Big[\sum_{\|z\|_\infty \leq \xi}\Big|\sum_{i=1}^n \frac{\sigma_i}{\|z\|^2}\Delta\phi_z(X_i)\Big|^2\Big] = \sum_{\|z\|_\infty \leq \xi}\mathbb{E}_{X,\sigma}\Big[\Big|\sum_{i=1}^n \frac{\sigma_i}{\|z\|^2}\Delta\phi_z(X_i)\Big|^2\Big]
$$

$$
= \sum_{\|z\|_\infty \leq \xi}\sum_{i=1}^n \mathbb{E}_{X,\sigma}\Big[\frac{\sigma_i^2}{\|z\|^4}|\Delta\phi_z(X_i)|^2\Big]
$$

$$
= \sum_{\|z\|_\infty \leq \xi}\sum_{i=1}^n |\Omega|\frac{16\pi^4\|z\|^4}{\|z\|^4} \lesssim n\sum_{\|z\|_\infty \leq \xi} 1 \lesssim n\xi^d.
$$

Combining the two bounds above yields the desired upper bound:

$$
\mathbb{E}_X\Big[\mathbb{E}_\sigma\Big[\sup_{f\in\boldsymbol{F}_{\rho,\xi}(\Omega)}\frac{1}{n}\sum_{i=1}^n \sigma_i\Delta f(X_i)\,\Big|\,X_1,\cdots,X_n\Big]\Big] \lesssim \frac{\sqrt{\rho}}{n}\sqrt{n\xi^d} = \sqrt{\frac{\rho}{n}}\xi^{\frac{d}{2}}.
$$

$\square$

### B.2.2 LOCAL RADEMACHER COMPLEXITY OF THE DEEP NEURAL NETWORK MODEL

In this section we aim to bound the local Rademacher Complexity of a Deep Neural Network. We first bound the covering number of the function space composed by the gradient of all possible neural networks and then apply a Duley Integral to achieve the final bound.

**Definition B.3.** *Let $\eta_l$ denote the l-ReLU activiation function. Here we use $\eta_3 := \max\{0,x\}^3$(E & Yu, 2018) as the activation function to ensure smoothness. We can define the space consisting of all neural network models with depth L, width W, sparsity constraint S and norm constraint B as follows:*

$$
\Phi(L,W,S,B) := \Big\{(\mathcal{W}^{(L)}\eta_3(\cdot)+b^{(L)})\cdots(\mathcal{W}^{(1)}x+b^{(1)})\,|\,\mathcal{W}^{(L)}\in\mathbb{R}^{1\times W}, b^{(L)}\in\mathbb{R}, \quad\text{(B.16)}
$$

$$
\mathcal{W}^{(1)}\in\mathbb{R}^{W\times d}, b^{(1)}\in\mathbb{R}^W, \mathcal{W}^{(l)}\in\mathbb{R}^{W\times W}, b^{(l)}\in\mathbb{R}^W(1<l<L), \quad\text{(B.17)}
$$

$$
\sum_{l=1}^L(\|\mathcal{W}^{(l)}\|_0+\|b^{(l)}\|_0) \leq S, \max_l \|\mathcal{W}^{(l)}\|_{\infty,\infty}\vee\|b^{(l)}\|_\infty \leq B\Big\}. \quad\text{(B.18)}
$$

*where $\|\cdot\|_0$ measures the number of nonzero entries in a matrix and $\|\cdot\|_{\infty,\infty}$ measures the maximum of the absolute values of the entries in a matrix.*
*For any $d\in\mathbb{Z}^+$, we refer to an arbitrary element in $\Phi(L,W,S,B)$ as a ReLU3 Deep Neural Network. Then for any index $1\leq k\leq L$, we use $F_k$ to denote the $k-$ReLU3 Deep Neural Network composed by the first k layers, i.e:*

$$
F_k(x) := (\mathcal{W}_F^{(k)}\eta_3(\cdot)+b_F^{(k)})\cdots(\mathcal{W}_F^{(1)}x+b_F^{(1)}).
$$

*Also, we use $\Phi_k(L,W,S,B)$ to denote the space consisting of all $F_k$. In particular, when $k=L$, we have:*

$$
F(x) := F_L(x) = (\mathcal{W}_F^{(L)}\eta_3(\cdot)+b_F^{(L)})\cdots(\mathcal{W}_F^{(1)}x+b_F^{(1)}), \text{ and } \Phi_L(L,W,S,B)=\Phi(L,W,S,B).
$$

*Furthermore, given that the domain $\Omega\subset[0,1]^d$ is bounded, we have $\sup_{x\in\Omega}\|x\|_\infty = 1$.*

**Lemma B.9.** *(Upper bound on $\infty$-norm of functions in DNN space) For any $1\leq k\leq L$, the following inequality holds:*

$$
\sup_{x\in\Omega,\ F_k\in\Phi_k(L,W,S,B)}\|F_k(x)\|_\infty \leq W^{\frac{3^{k-1}-1}{2}}(B\vee d)^{\frac{5\cdot 3^{k-1}-1}{2}}2^{\frac{3^k-1}{2}-k+1}.
$$

*Proof.* We use induction to prove this claim.
Base cases: When $k=1$, we have that for any $x\in\Omega$ and any $F_1\in\Phi_1(L,W,S,B)$, the following holds:

$$
\|F_1(x)\|_\infty = \|\mathcal{W}_F^{(1)}x+b_F^{(1)}\|_\infty \leq \|\mathcal{W}_F^{(1)}\|_\infty\|x\|_\infty + \|b_F^{(1)}\|_\infty
$$

$$
\leq d\|\mathcal{W}_F^{(1)}\|_{\infty,\infty} + B \leq dB + B \leq 2(B\vee d)^2. \quad\text{(B.19)}
$$

When $k = 2$, we have that for any $x \in \Omega$ and any $F_2 \in \Phi_2(L, W, S, B)$, the following holds:

$$\|F_2(x)\|_\infty = \|\mathcal{W}_F^{(2)} \eta_3(F_1(x)) + b_F^{(2)}\|_\infty \leq \|\mathcal{W}_F^{(2)}\|_\infty \|\eta_3(F_1(x))\|_\infty + \|b_F^{(2)}\|_\infty \leq W\|\mathcal{W}_F^{(2)}\|_{\infty,\infty}\|F_1(x)\|_\infty^3 + B.$$

By applying the bound proved in the case when $k = 1$, we have:

$$\|F_2(x)\|_\infty \leq WB(dB + B)^3 + B = WB^4(d+1)^3 + B$$
$$= WB^4(d^3 + 3d^2 + 3d + 1) + B \leq 8W(B \vee d)^7.$$

where the last inequality follows from the assumption that $W \geq 2$.

Inductive Step: Now we assume that the claim has been proved for $k-1$, where $3 \leq k \leq L$. Similarly, for any $x \in \Omega$ and any $F_k \in \Phi_k(L, W, S, B)$, we have:

$$\|F_k(x)\|_\infty = \|\mathcal{W}_F^{(k)} \eta_3(F_{k-1}(x)) + b_F^{(k)}\|_\infty \leq \|\mathcal{W}_F^{(k)}\|_\infty \|\eta_3(F_{k-1}(x))\|_\infty + \|b_F^{(k)}\|_\infty$$
$$\leq W\|\mathcal{W}_F^{(k)}\|_{\infty,\infty}\|F_{k-1}(x)\|_\infty^3 + B \leq WB\|F_{k-1}(x)\|_\infty^3 + B.$$

Using inductive hypothesis, we can further deduce that:

$$\|F_k(x)\|_\infty \leq WB \times W^{\frac{3^{k-1}-3}{2}}(B \vee d)^{\frac{5 \cdot 3^{k-1}-3}{2}} 2^{\frac{3^k-3}{2}-3k+6} + B$$
$$\leq W^{\frac{3^{k-1}-1}{2}}(B \vee d)^{\frac{5 \cdot 3^{k-1}-1}{2}} 2^{\frac{3^k-3}{2}-3k+6} + B \vee d$$
$$\leq W^{\frac{3^{k-1}-1}{2}}(B \vee d)^{\frac{5 \cdot 3^{k-1}-1}{2}} [2^{\frac{3^k-3}{2}-3k+6} + 1]$$
$$\leq W^{\frac{3^{k-1}-1}{2}}(B \vee d)^{\frac{5 \cdot 3^{k-1}-1}{2}} 2^{\frac{3^k-3}{2}-k+2} \quad (k \geq 3)$$
$$= W^{\frac{3^{k-1}-1}{2}}(B \vee d)^{\frac{5 \cdot 3^{k-1}-1}{2}} 2^{\frac{3^k-1}{2}-k+1}.$$

Taking supremum with respect to $x \in \Omega$ and $F_k \in \Phi_k(L, W, S, B)$ on the LHS implies that the given upper bound also holds for $k$. By induction, the claim is proved. $\qquad\square$

We also need to show that the ReLU3 activation function is a Lipschitzness functions over a bounded domain.

**Lemma B.10.** *For any $k \in \mathbb{Z}^+$, consider the $k-$ReLU activation function $\eta_k$ defined on some bounded domain $\mathcal{D} \subset \mathbb{R}^d$ (i.e, $\sup_{x \in \mathcal{D}} \|x\|_\infty \leq C$ for some $C > 0$). Then we have that for any $x, y \in \mathcal{D}$, the following inequalities hold:*

$$\|\eta_1(x) - \eta_1(y)\|_\infty \leq \|x - y\|_\infty,$$
$$\|\eta_2(x) - \eta_2(y)\|_\infty \leq 2C\|x - y\|_\infty,$$
$$\|\eta_3(x) - \eta_3(y)\|_\infty \leq 3C^2\|x - y\|_\infty.$$

*Proof.* This is because $|\nabla \eta_1(x)| = |\max\{1, 0\}| = 1$, $|\nabla \eta_2(x)| = |2\max\{x, 0\}| \leq 2C$ and $|\nabla \eta_3(x)| = |3\max\{x, 0\}^2| \leq 3C^2$. $\qquad\square$

**Lemma B.11.** *(Relation between the covering number of DNN space and parameter space) For any $1 \leq k \leq L$, suppose that a pair of different two networks $F_k, G_k \in \Phi_k(L, W, S, B)$ are given by:*

$$F_k(x) := (\mathcal{W}_F^{(k)} \eta_3(\cdot) + b_F^{(k)}) \cdots (\mathcal{W}_F^{(1)} x + b_F^{(1)}),$$
$$G_k(x) := (\mathcal{W}_G^{(k)} \eta_3(\cdot) + b_G^{(k)}) \cdots (\mathcal{W}_G^{(1)} x + b_G^{(1)}).$$

*Furthermore, assume that the $\| \ \|_\infty$ norm of the distance between the parameter spaces of $F_k$ and $G_k$ is uniformly upper bounded by $\delta$, i.e*

$$\|W_F^{(l)} - W_G^{(l)}\|_{\infty,\infty} \leq \delta, \ \|b_F^{(l)} - b_G^{(l)}\|_\infty \leq \delta, \ (\forall\, 1 \leq l \leq k). \tag{B.20}$$

*Then we have:*

$$\sup_{x \in \Omega} \|F_k(x) - G_k(x)\|_\infty \leq \delta W^{\frac{3^{k-1}-1}{2}}(B \vee d)^{\frac{5 \cdot 3^{k-1}-1}{2}} 2^{\frac{3^k-1}{2}-k+1} 3^{k-1}. \tag{B.21}$$

*Proof.* Let's prove the claim by using induction on $k$.

Base Case: When $k = 1$, we have that for any $x \in \Omega$ and any $F_1, G_1 \in \Phi_1(L, W, S, B)$ satisfying constraint B.20, the following holds:

$$
\begin{aligned}
\|F_1(x) - G_1(x)\|_\infty &= \|\mathcal{W}_F^{(1)} x + b_F^{(1)} - \mathcal{W}_G^{(1)} x - b_G^{(1)}\|_\infty \\
&\leq \|\mathcal{W}_F^{(1)} - \mathcal{W}_G^{(1)}\|_\infty \|x\|_\infty + \|b_F^{(1)} - b_G^{(1)}\|_\infty \\
&\leq \delta d + \delta = \delta(d + 1) \leq 2\delta(B \vee d) \leq 2\delta(B \vee d)^2.
\end{aligned}
\tag{B.22}
$$

When $k = 2$, we have that for any $x \in \Omega$ and any $F_2, G_2 \in \Phi_2(L, W, S, B)$ satisfying constraint B.20, the following inequality holds:

$$
\begin{aligned}
\|F_2(x) - G_2(x)\|_\infty &= \|\mathcal{W}_F^{(2)} \eta_3(F_1(x)) + b_F^{(2)} - \mathcal{W}_G^{(2)} \eta_3(G_1(x)) - b_G^{(2)}\|_\infty \\
&\leq \|\mathcal{W}_F^{(2)} \eta_3(F_1(x)) - \mathcal{W}_G^{(2)} \eta_3(G_1(x))\|_\infty + \|b_F^{(2)} - b_G^{(2)}\|_\infty \\
&\leq \|\mathcal{W}_F^{(2)} \eta_3(F_1(x)) - \mathcal{W}_G^{(2)} \eta_3(F_1(x))\|_\infty + \|\mathcal{W}_G^{(2)} \eta_3(F_1(x)) - \mathcal{W}_G^{(2)} \eta_3(G_1(x))\|_\infty + \delta.
\end{aligned}
$$

By applying the upper bound proved in equation B.19, we can upper bound the first part $\|\mathcal{W}_F^{(2)} \eta_3(F_1(x)) - \mathcal{W}_G^{(2)} \eta_3(F_1(x))\|_\infty$ by:

$$
\begin{aligned}
\|\mathcal{W}_F^{(2)} \eta_3(F_1(x)) - \mathcal{W}_G^{(2)} \eta_3(F_1(x))\|_\infty &\leq \|\mathcal{W}_F^{(2)} - \mathcal{W}_G^{(2)}\|_\infty \|\eta_3(F_1(x))\|_\infty \\
&\leq W\delta \|F_1(x)\|_\infty^3 \leq \delta W[2(B \vee d)^2]^3.
\end{aligned}
$$

By applying the Lipschitz condition proved in Lemma B.10 and the bound proved in equation B.22, we can further upper bound the second part $\|\mathcal{W}_G^{(2)} \eta_3(F_1(x)) - \mathcal{W}_G^{(2)} \eta_3(G_1(x))\|_\infty$ by:

$$
\begin{aligned}
\|\mathcal{W}_G^{(2)} \eta_3(F_1(x)) - \mathcal{W}_G^{(2)} \eta_3(G_1(x))\|_\infty &\leq \|\mathcal{W}_G^{(2)}\|_\infty \|\eta_3(F_1(x)) - \eta_3(G_1(x))\|_\infty \\
&\leq WB \times 3 \sup_{F_1 \in \Phi_1(L, W, S, B)} \|F_1(x)\|_\infty^2 \times \|F_1(x) - G_1(x)\|_\infty \\
&\leq WB \times 3[2(B \vee d)^2]^2 \times 2\delta(B \vee d) \\
&\leq 24\delta W(B \vee d)^6.
\end{aligned}
$$

Summing the two upper bounds above yields:

$$
\|F_2(x) - G_2(x)\|_\infty \leq 8\delta W(B \vee d)^6 + 24\delta W(B \vee d)^6 + \delta \leq 24\delta W(B \vee d)^7.
$$

where we again use the assumption $d \geq 2$ in the last step.

Inductive Step: Now we assume that the claim has been proved for $k - 1$, where $k \geq 3$. For any $x \in \Omega$ and $F_k \in \Phi_k(L, W, S, B)$, we have that:

$$
\begin{aligned}
\|F_k(x) - G_k(x)\|_\infty &= \|\mathcal{W}_F^{(k)} \eta_3(F_{k-1}(x)) + b_F^{(k)} - \mathcal{W}_G^{(k)} \eta_3(G_{k-1}(x)) - b_G^{(k)}\|_\infty \\
&\leq \|\mathcal{W}_F^{(k)} \eta_3(F_{k-1}(x)) - \mathcal{W}_G^{(k)} \eta_3(G_{k-1}(x))\|_\infty + \|b_F^{(k)} - b_G^{(k)}\|_\infty \\
&\leq \|\mathcal{W}_F^{(k)} \eta_3(F_{k-1}(x)) - \mathcal{W}_G^{(k)} \eta_3(G_{k-1}(x))\|_\infty + \delta.
\end{aligned}
$$

Applying triangle inequality helps us upper bound the first term above as follows:

$$
\begin{aligned}
&\|\mathcal{W}_F^{(k)} \eta_3(F_{k-1}(x)) - \mathcal{W}_G^{(k)} \eta_3(G_{k-1}(x))\|_\infty \\
&\leq \|\mathcal{W}_F^{(k)} \eta_3(F_{k-1}(x)) - \mathcal{W}_G^{(k)} \eta_3(F_{k-1}(x))\|_\infty + \|\mathcal{W}_G^{(k)} \eta_3(F_{k-1}(x)) - \mathcal{W}_G^{(k)} \eta_3(G_{k-1}(x))\|_\infty \\
&\leq \|\mathcal{W}_F^{(k)} - \mathcal{W}_G^{(k)}\|_\infty \|\eta_3(F_{k-1}(x))\|_\infty + \|\mathcal{W}_G^{(k)}\|_\infty \|\eta_3(F_{k-1}(x)) - \eta_3(G_{k-1}(x))\|_\infty \\
&\leq \delta W \|F_{k-1}(x)\|_\infty^3 + BW \|\eta_3(F_{k-1}(x)) - \eta_3(G_{k-1}(x))\|_\infty.
\end{aligned}
$$

From Lemma B.9, we can upper bound the first term $\delta W \|F_{k-1}(x)\|_\infty^3$ by:

$$
\delta W \|F_{k-1}(x)\|_\infty^3 \leq \delta W^{\frac{3^{k-1}-1}{2}} (B \vee d)^{\frac{5 \cdot 3^{k-1}-3}{2}} 2^{\frac{3^k-3}{2}-3k+6}.
$$

Moreover, applying Lemma B.10 and the inductive hypothesis let us upper bound the second term $BW\|\eta_3(F_{k-1}(x)) - \eta_3(G_{k-1}(x))\|_\infty$ as follows:

$BW\|\eta_3(F_{k-1}(x)) - \eta_3(G_{k-1}(x))\|_\infty$

$\leq BW \times 3 \sup_{x\in\Omega,\ F_{k-1}\in\Phi_{k-1}(L,W,S,B)} \|F_{k-1}(x)\|_\infty^2 \times \|F_{k-1}(x) - G_{k-1}(x)\|_\infty$

$\leq 3BW \times W^{3^{k-2}-1}(B\vee d)^{5\times 3^{k-2}-1}2^{3^{k-1}-1-2k+4}\|F_{k-1}(x) - G_{k-1}(x)\|_\infty$

$\leq 3BW \times W^{3^{k-2}-1}(B\vee d)^{5\times 3^{k-2}-1}2^{3^{k-1}-1-2k+4} \times \delta W^{\frac{3^{k-2}-1}{2}}(B\vee d)^{\frac{5\cdot 3^{k-2}-1}{2}}2^{\frac{3^{k-1}-1}{2}-k+2}3^{k-2}$

$\leq 3^{k-1}\delta W^{\frac{3^{k-1}-1}{2}}(B\vee d)^{\frac{5\times 3^{k-1}-1}{2}}2^{\frac{3^k-1}{2}-3k+5}.$

Combining the two upper bounds derived above yields:

$$\|F_k(x) - G_k(x)\|_\infty \leq \delta W^{\frac{3^{k-1}-1}{2}}(B\vee d)^{\frac{5\cdot 3^{k-1}-3}{2}}2^{\frac{3^k-3}{2}-3k+6}$$
$$+ 3^{k-1}\delta W^{\frac{3^{k-1}-1}{2}}(B\vee d)^{\frac{5\times 3^{k-1}-1}{2}}2^{\frac{3^k-1}{2}-3k+5} + \delta$$
$$\leq \delta 3^{k-1}W^{\frac{3^{k-1}-1}{2}}(B\vee d)^{\frac{5\times 3^{k-1}-1}{2}}2^{\frac{3^k-1}{2}-k+1},$$

where the last inequality above follows from $k \geq 3$. Taking supremum with respect to $x \in \Omega$ on the LHS implies the given upper bound also holds for $k$. By induction, the claim is proved. $\square$

**Theorem B.4.** *(Bounding the DNN space covering number) Fix some sufficiently large $N \in \mathbb{Z}^+$. Consider a Deep Neural Network space $\Phi(L, W, S, B)$ with $L = O(1), W = O(N), S = O(N)$ and $B = O(N)$. Then the $\log$ value of the covering number of this DNN space with respect to the inf-norm $\|F(x)\|_\infty := \sup_{x\in\Omega}|F(x)|$, which is denoted by $\mathcal{N}(\delta, \Phi(L, W, S, B), \|\cdot\|_\infty)$, can be upper bounded by:*

$$\log\mathcal{N}(\delta, \Phi(L, W, S, B), \|\cdot\|_\infty) = O\left(S\left[\log(\delta^{-1}) + 3^L\log(WB)\right]\right). \qquad (B.23)$$

*Proof.* We firstly fix a sparsity pattern (i.e, the locations of the non-zero entries are fixed). By picking $k = L$ in Lemma B.11, we get the following upper bound on the covering number with respect to $\|\cdot\|_\infty$:

$$\left(\frac{\delta}{3^{L-1}W^{\frac{3^{L-1}-1}{2}}(B\vee d)^{\frac{5\times 3^{L-1}-1}{2}}2^{\frac{3^L-1}{2}-L+2}}\right)^{-S}.$$

Furthermore, note that the number of feasible configurations is upper bounded by $\binom{(W+1)^L}{S} \leq (W+1)^{LS}$.(Schmidt-Hieber et al., 2020; Farrell et al., 2021) Plug in the previous inequality and yields:

$$\log\mathcal{N}(\delta, \Phi(L, W, S, B), \|\cdot\|_\infty) \leq \log\left[(W+1)^{LS}\left(\frac{\delta}{3^{L-1}W^{\frac{3^{L-1}-1}{2}}(B\vee d)^{\frac{5\times 3^{L-1}-1}{2}}2^{\frac{3^L-1}{2}-L+1}}\right)^{-S}\right]$$
$$\leq S\log\left[\delta^{-1}(W+1)^L 3^{L-1}W^{\frac{3^{L-1}-1}{2}}(B\vee d)^{\frac{5\times 3^{L-1}-1}{2}}2^{\frac{3^L-1}{2}-L+1}\right]$$
$$\lesssim S\left[\log(\delta^{-1}) + L\log(3W) + 3^L\log(W(B\vee d)) + 3^L\log 2\right].$$

Note that here the dimension $d$ is some constant. Thus, by plugging in thee given magnitudes $L = O(1), W = O(N), S = O(N)$ and $B = O(N)$, we can further deduce that:

$$\log\mathcal{N}(\delta, \Phi(L, W, S, B), \|\cdot\|_\infty) \lesssim S\left[\log(\delta^{-1}) + 3^L\log(WB)\right].$$

This finishes our proof. $\square$

Now let's consider upper bounding the covering number of the $l_2$ norm of the sparse Deep Neural Networks' gradients. Note that for any $1 \leq k \leq L - 1$, any $k-$ReLU3 Deep Neural Network $F_k \in \Phi_k(L, W, S, B)$ is a map from $\mathbb{R}^d$ to $\mathbb{R}^W$. For any $1 \leq l \leq W$, we use $F_{k,l}(x)$ to denote the

$l$-th component of the map $F_k$. This helps us write the map $F_k(x)$ and its Jacobian matrix $J[F_k](x)$ explicitly as:

$$F_k(x) = [F_{k,1}(x), F_{k,2}(x), \cdots, F_{k,W}(x)]^T \in \mathbb{R}^W.$$

$$J[F_k](x) = \begin{bmatrix} \frac{\partial}{\partial x_1} F_{k,1}(x) & \frac{\partial}{\partial x_2} F_{k,1}(x) & \cdots & \frac{\partial}{\partial x_d} F_{k,1}(x) \\ \frac{\partial}{\partial x_1} F_{k,2}(x) & \frac{\partial}{\partial x_2} F_{k,2}(x) & \cdots & \frac{\partial}{\partial x_d} F_{k,2}(x) \\ \cdots & \cdots & \ddots & \\ \frac{\partial}{\partial x_1} F_{k,W}(x) & \frac{\partial}{\partial x_2} F_{k,W}(x) & \cdots & \frac{\partial}{\partial x_d} F_{k,W}(x) \end{bmatrix} \in \mathbb{R}^{W \times d}.$$

In particular, when $k = L$, we have that any $F_L \in \Phi_L(L, W, S, B) = \Phi(L, W, S, B)$ is a map from $\mathbb{R}^d$ to $\mathbb{R}$. Thus, its Jacobian can be explicitly written as the following row vector:

$$J[F_L](x) = [\frac{\partial}{\partial x_1} F_L(x), \frac{\partial}{\partial x_2} F_L(x), \cdots \frac{\partial}{\partial x_d} F_L(x)] \in \mathbb{R}^{1 \times d}.$$

**Lemma B.12.** *(Upper bound on $\infty$-norm of Jacobian/Gradient of elements in the DNN space) For any $1 \le k \le L$, the following inequality holds:*

$$\sup_{x \in \Omega, F_k \in \Phi_k(L, W, S, B)} \|J[F_k](x)\|_\infty \le W^{\frac{3^{k-1}-1}{2}} (B \vee d)^{\frac{5 \cdot 3^{k-1}-1}{2}} 2^{\frac{3^k-1}{2} - k + 1} 3^{k-1}.$$

*Proof.* We use induction on $k$ to prove the claim.
Base case: $k = 1$. By the definition of Jacobian matrix, we have that for any $x \in \Omega$ and any $F_1 \in \Phi_1(L, W, S, B)$, the following holds:

$$\|J[F_1](x)\|_\infty = \|\mathcal{W}_F^{(1)}\|_\infty \le dB \le 2(B \vee d)^2.$$

Inductive Step: Assume that the claim has been proved for $k - 1$, where $2 \le k \le L$. For any $x \in \Omega$ and any $F_k \in \Phi_k(L, W, S, B)$, by applying the Chain Rule, we can write the Jacobian matrix $J[F_k](x)$ as $J[F_k](x) = \mathcal{W}_F^{(k)} J[\eta_3 \circ F_{k-1}](x)$, where the ReLU3 activation function $\eta_3$ is applied to each component $F_{k-1,l}$ ($1 \le l \le W$) of the map $F_k$. Then we have the following upper bound:

$$\|J[F_k](x)\|_\infty \le \|\mathcal{W}_F^{(k)}\|_\infty \|J[\eta_3 \circ F_{k-1}](x)\|_\infty \le WB\|J[\eta_3 \circ F_{k-1}](x)\|_\infty. \tag{B.24}$$

Note that the composition $\eta_3 \circ F_{k-1}$ is a map from $\mathbb{R}^d$ to $\mathbb{R}^W$. Hence, the Jacobian matrix $J[\eta_3 \circ F_{k-1}](x)$ is of shape $\mathbb{R}^{W \times d}$. Applying the Chain Rule again implies:

$$\|J[\eta_3 \circ F_{k-1}](x)\|_\infty = \sup_{1 \le l \le W} (\sum_{j=1}^d |3\eta_2(F_{k-1,l}(x)) \frac{\partial F_{k-1,l}(x)}{\partial x_j}|).$$

Furthermore, for any $1 \le l \le W$, the summation on the RHS above can be upper bounded by:

$$\sum_{j=1}^d |3\eta_2(F_{k-1,l}(x)) \frac{\partial F_{k-1,l}(x)}{\partial x_j}| \le 3\|F_{k-1}(x)\|_\infty^2 (\sum_{j=1}^d |\frac{\partial}{\partial x_j} F_{k-1,l}(x)|) \le 3\|F_{k-1}(x)\|_\infty^2 \|J[F_{k-1}](x)\|_\infty.$$

Now let's take supremum with respect to $l$ and apply the inductive hypothesis and Lemma B.9. This yields:

$$\|J[\eta_3 \circ F_{k-1}](x)\|_\infty \le 3W^{3^{k-2}-1} (B \vee d)^{5 \cdot 3^{k-2}-1} 2^{3^{k-1}-1-2k+4} \times W^{\frac{3^{k-2}-1}{2}} (B \vee d)^{\frac{5 \cdot 3^{k-2}-1}{2}} 2^{\frac{3^{k-1}-1}{2} - k + 2} 3^{k-2}$$
$$= W^{\frac{3^{k-1}-1}{2}-1} (B \vee d)^{\frac{5 \cdot 3^{k-1}-1}{2}-1} 2^{\frac{3^k-1}{2} - 3k + 5} 3^{k-1}. \tag{B.25}$$

By substituting equation B.25 into equation B.24, we can derive the final bound:

$$\|J[F_k](x)\|_\infty \le WB\|J[\eta_3 \circ F_{k-1}](x)\|_\infty \le W^{\frac{3^{k-1}-1}{2}} (B \vee d)^{\frac{5 \cdot 3^{k-1}-1}{2}} 2^{\frac{3^k-1}{2} - 3k + 5} 3^{k-1}$$
$$\le W^{\frac{3^{k-1}-1}{2}} (B \vee d)^{\frac{5 \cdot 3^{k-1}-1}{2}} 2^{\frac{3^k-1}{2} - k + 1} 3^{k-1}.$$

where the last inequality above follows from $k \ge 2$. Taking supremum with respect to $x \in \Omega$ and $F_k \in \Phi_k(L, W, S, B)$ on the LHS implies that the given upper bound also holds for $k$. By induction, the claim is proved. $\qquad \square$

For the convenience of the following proof, we first prove this lemma for vector 2 norm and $\infty$ norm.

**Lemma B.13.** *Given any two row vectors $\boldsymbol{u}, \boldsymbol{v} \in \mathbb{R}^{1 \times d}$, we have:*

$$\left| \|\boldsymbol{u}\| - \|\boldsymbol{v}\| \right| \leq \|\boldsymbol{u} - \boldsymbol{v}\|_\infty.$$

*Proof.* Assume that the two vectors $\boldsymbol{u}, \boldsymbol{v} \in \mathbb{R}^d$ can be explicitly written as $\boldsymbol{u} = [u_1, u_2, \cdots, u_d]$ and $v = [v_1, v_2, \cdots, v_d]$, respectively. By applying Cauchy-Schwarz inequality, we have:

$$
\begin{aligned}
\left| \|\boldsymbol{u}\| - \|\boldsymbol{v}\| \right|^2 &= \left| \sqrt{\sum_{i=1}^d u_i^2} - \sqrt{\sum_{i=1}^d v_i^2} \right|^2 \\
&= \sum_{i=1}^d u_i^2 + \sum_{i=1}^d v_i^2 - 2 \sqrt{\sum_{i=1}^d u_i^2} \sqrt{\sum_{i=1}^d v_i^2} \\
&\leq \sum_{i=1}^d u_i^2 + \sum_{i=1}^d v_i^2 - 2 \sum_{i=1}^d u_i v_i = \sum_{i=1}^d |u_i - v_i|^2 \\
&\leq \left( \sum_{i=1}^d |u_i - v_i| \right)^2 = \|\boldsymbol{u} - \boldsymbol{v}\|_\infty^2.
\end{aligned}
$$

Taking the square root on both sides yields the desired inequality. $\qquad \square$

Then we upper bound the Lipschitz constant of the gradient of the neural network. Given a DNN space $\Phi(L, W, S, B)$, we define a corresponding DNN Gradient space $\nabla\Phi(L, W, S, B)$ as:

$$\nabla\Phi(L, W, S, B) := \{\|\nabla F\| \mid F \in \Phi(L, W, S, B)\}. \tag{B.26}$$

**Lemma B.14.** *(Relation between the covering number of the DNN Gradient space and parameter space) For any $1 \leq k \leq L$, suppose that a pair of different two networks $F_k, G_k \in \Phi_k(L, W, S, B)$ are given by:*

$$
\begin{aligned}
F_k(x) &:= (\mathcal{W}_F^{(k)} \eta_3(\cdot) + b_F^{(k)}) \cdots (\mathcal{W}_F^{(1)} x + b_F^{(1)}), \\
G_k(x) &:= (\mathcal{W}_G^{(k)} \eta_3(\cdot) + b_G^{(k)}) \cdots (\mathcal{W}_G^{(1)} x + b_G^{(1)}).
\end{aligned}
$$

*Furthermore, assume that the $\| \ \|_\infty$ norm of the distance between the parameter spaces is uniformly upper bounded by $\delta$, i.e*

$$\|W_F^{(l)} - W_G^{(l)}\|_{\infty,\infty} \leq \delta, \ \|b_F^{(l)} - b_G^{(l)}\|_\infty \leq \delta, \ (\forall \ 1 \leq l \leq k). \tag{B.27}$$

*Then we have:*

$$\sup_{x \in \Omega} \|J[F_k](x) - J[G_k](x)\|_\infty \leq \delta W^{\frac{3^{k-1}-1}{2}} (B \vee d)^{\frac{5 \cdot 3^{k-1}-1}{2}} 2^{\frac{3^k-1}{2}-k+1} 3^{2k-2}. \tag{B.28}$$

*In particular, when $k = L$, we have:*

$$\sup_{x \in \Omega} \left| \|\nabla F_L(x)\| - \|\nabla G_L(x)\| \right| \leq \delta W^{\frac{3^{L-1}-1}{2}} (B \vee d)^{\frac{5 \cdot 3^{L-1}-1}{2}} 2^{\frac{3^L-1}{2}-L+1} 3^{2L-2}. \tag{B.29}$$

*Proof.* We use induction on $k$ to prove the claim.
Base case: When $k = 1$, we have that for any $x \in \Omega$ and any $F_1, G_1 \in \Phi_1(L, W, S, B)$, the following holds:

$$\|J[F_1](x) - J[G_1](x)\|_\infty = \|\mathcal{W}_F^{(1)} - \mathcal{W}_G^{(1)}\|_\infty \leq \delta d \leq 2\delta(B \vee d)^2.$$

Inductive Step: assume that the claim has been proved for $k - 1$, where $2 \leq k \leq L$. Then for any $x \in \Omega$ and $F_k, G_k \in \Phi_k(L, W, S, B)$ satisfying constraint B.27, applying the Chain Rule and

triangle inequality help us upper bound the inf-norm $\|J[F_k](x) - J[G_k](x)\|_\infty$ by:

$$\|J[F_k](x) - J[G_k](x)\|_\infty = \|\mathcal{W}_F^{(k)} J[\eta_3 \circ F_{k-1}](x) - \mathcal{W}_G^{(k)} J[\eta_3 \circ G_{k-1}](x)\|_\infty$$
$$\leq \|\mathcal{W}_F^{(k)} J[\eta_3 \circ F_{k-1}](x) - \mathcal{W}_G^{(k)} J[\eta_3 \circ F_{k-1}](x)\|_\infty + \|\mathcal{W}_G^{(k)} J[\eta_3 \circ F_{k-1}](x) - \mathcal{W}_G^{(k)} J[\eta_3 \circ G_{k-1}](x)\|_\infty$$
$$\leq \|\mathcal{W}_F^{(k)} - \mathcal{W}_G^{(k)}\|_\infty \|J[\eta_3 \circ F_{k-1}](x)\|_\infty + \|\mathcal{W}_G^{(k)}\|_\infty \|J[\eta_3 \circ F_{k-1}](x) - J[\eta_3 \circ G_{k-1}](x)\|_\infty$$
$$\leq \delta W \|J[\eta_3 \circ F_{k-1}](x)\|_\infty + BW \|J[\eta_3 \circ F_{k-1}](x) - J[\eta_3 \circ G_{k-1}](x)\|_\infty.$$
$$\text{(B.30)}$$

Using equation B.25 helps us upper bound the first term by:

$$\delta W \|J[\eta_3 \circ F_{k-1}](x)\|_\infty \leq \delta W^{\frac{3^{k-1}-1}{2}} (B \vee d)^{\frac{5 \cdot 3^{k-1}-1}{2}-1} 2^{\frac{3^{k-1}-1}{2}-3k+5} 3^{k-1}. \qquad \text{(B.31)}$$

Note that the two compositions $\eta_3 \circ F_{k-1}$ and $\eta_3 \circ G_{k-1}$ both map from $\mathbb{R}^d$ to $\mathbb{R}^W$. Hence, the two Jacobian matrices $J[\eta_3 \circ F_{k-1}](x)$ and $J[\eta_3 \circ G_{k-1}](x)$ are of shape $\mathbb{R}^{W \times d}$. Applying the Chain Rule again implies:

$$\|J[\eta_3 \circ F_{k-1}](x) - J[\eta_3 \circ G_{k-1}](x)\|_\infty = \sup_{1 \leq l \leq W} \Big( \sum_{j=1}^d |3\eta_2(F_{k-1,l}(x)) \frac{\partial F_{k-1,l}(x)}{\partial x_j} - 3\eta_2(G_{k-1,l}(x)) \frac{\partial G_{k-1,l}(x)}{\partial x_j}| \Big).$$

For any $1 \leq l \leq W$, the summation on the RHS above can be upper bounded by:

$$\sum_{j=1}^d |3\eta_2(F_{k-1,l}(x)) \frac{\partial F_{k-1,l}(x)}{\partial x_j} - 3\eta_2(G_{k-1,l}(x)) \frac{\partial G_{k-1,l}(x)}{\partial x_j}|$$
$$\leq \sum_{j=1}^d |3\eta_2(F_{k-1,l}(x)) \frac{\partial F_{k-1,l}(x)}{\partial x_j} - 3\eta_2(G_{k-1,l}(x)) \frac{\partial F_{k-1,l}(x)}{\partial x_j}|$$
$$+ \sum_{j=1}^d |3\eta_2(G_{k-1,l}(x)) \frac{\partial F_{k-1,l}(x)}{\partial x_j} - 3\eta_2(G_{k-1,l}(x)) \frac{\partial G_{k-1,l}(x)}{\partial x_j}|$$
$$\leq \sum_{j=1}^d |3\eta_2(F_{k-1,l}(x)) - 3\eta_2(G_{k-1,l}(x))| |\frac{\partial F_{k-1,l}(x)}{\partial x_j}| + \sum_{j=1}^d |3\eta_2(G_{k-1,l}(x))| |\frac{\partial F_{k-1,l}(x)}{\partial x_j} - \frac{\partial G_{k-1,l}(x)}{\partial x_j}|.$$

We denote the two summations above by $T_1$ and $T_2$, respectively:

$$T_1 := \sum_{j=1}^d |3\eta_2(F_{k-1,l}(x)) - 3\eta_2(G_{k-1,l}(x))| |\frac{\partial F_{k-1,l}(x)}{\partial x_j}|,$$

$$T_2 := \sum_{j=1}^d |3\eta_2(G_{k-1,l}(x))| |\frac{\partial F_{k-1,l}(x)}{\partial x_j} - \frac{\partial G_{k-1,l}(x)}{\partial x_j}|.$$

For the first sum $T_1$, applying Lemma B.9, Lemma B.10, Lemma B.11 and Lemma B.12 yields the following upper bound:

$$T_1 \leq 6 \Big( \sup_{x \in \Omega, \ F_{k-1} \in \Phi_{k-1}(L,W,S,B)} \|F_{k-1}(x)\|_\infty \Big) \|F_{k-1}(x) - G_{k-1}(x)\|_\infty \sum_{j=1}^d |\frac{\partial F_{k-1,l}(x)}{\partial x_j}|$$
$$\leq 6 \Big( \sup_{x \in \Omega, \ F_{k-1} \in \Phi_{k-1}(L,W,S,B)} \|F_{k-1}(x)\|_\infty \Big) \|F_{k-1}(x) - G_{k-1}(x)\|_\infty \|J[F_{k-1}](x)\|_\infty$$
$$\leq 3 \times W^{\frac{3^{k-2}-1}{2}} (B \vee d)^{\frac{5 \cdot 3^{k-2}-1}{2}} 2^{\frac{3^{k-1}-1}{2}-k+2} \times \delta W^{\frac{3^{k-2}-1}{2}} (B \vee d)^{\frac{5 \cdot 3^{k-2}-1}{2}} 2^{\frac{3^{k-1}-1}{2}-k+2} 3^{k-2}$$
$$\times W^{\frac{3^{k-2}-1}{2}} (B \vee d)^{\frac{5 \cdot 3^{k-2}-1}{2}} 2^{\frac{3^{k-1}-1}{2}-k+2} 3^{k-2} = \delta W^{\frac{3^{k-1}-3}{2}} (B \vee d)^{\frac{5 \cdot 3^{k-1}-3}{2}} 2^{\frac{3^k-3}{2}-3k+6} 3^{2k-3}.$$

For the second sum $T_2$, applying Lemma B.9 and inductive hypothesis yields:

$$T_2 \leq 3 \Big( \sup_{x \in \Omega, \ G_{k-1} \in \Phi_{k-1}(L,W,S,B)} \|G_{k-1}(x)\|_\infty \Big)^2 \|J[F_{k-1}](x) - J[G_{k-1}](x)\|_\infty$$
$$\leq 3 \times W^{3^{k-2}-1} (B \vee d)^{5 \cdot 3^{k-2}-1} 2^{3^{k-1}-1-2k+4} \times \delta W^{\frac{3^{k-2}-1}{2}} (B \vee d)^{\frac{5 \cdot 3^{k-2}-1}{2}} 2^{\frac{3^{k-1}-1}{2}-k+2} 3^{2k-4}$$
$$= \delta W^{\frac{3^{k-1}-3}{2}} (B \vee d)^{\frac{5 \cdot 3^{k-1}-3}{2}} 2^{\frac{3^k-3}{2}-3k+6} 3^{2k-3}.$$

Combining the two upper bounds on $T_1$ and $T_2$ yields:

$$\sum_{j=1}^{d} |3\eta_2(F_{k-1,l}(x))\frac{\partial F_{k-1,l}(x)}{\partial x_j} - 3\eta_2(G_{k-1,l}(x))\frac{\partial G_{k-1,l}(x)}{\partial x_j}|$$

$$\leq T_1 + T_2 \leq 2 \times \delta W^{\frac{3^{k-1}-3}{2}}(B \vee d)^{\frac{5 \cdot 3^{k-1}-3}{2}} 2^{\frac{3^k-3}{2}-3k+6} 3^{2k-3}.$$

By taking supremum with respect to $1 \leq l \leq W$ on the LHS yields:

$$BW\|J[\eta_3 \circ F_{k-1}](x) - J[\eta_3 \circ G_{k-1}](x)\|_\infty \leq \delta W^{\frac{3^{k-1}-1}{2}}(B \vee d)^{\frac{5 \cdot 3^{k-1}-1}{2}} 2^{\frac{3^k-1}{2}-3k+6} 3^{2k-3}. \tag{B.32}$$

By adding the two upper bounds in B.31 and B.32, we can deduce that:

$$\|J[F_k](x) - J[G_k](x)\|_\infty \leq \delta W^{\frac{3^{k-1}-1}{2}}(B \vee d)^{\frac{5 \cdot 3^{k-1}-1}{2}-1} 2^{\frac{3^k-1}{2}-3k+5} 3^{k-1}$$

$$+ \delta W^{\frac{3^{k-1}-1}{2}}(B \vee d)^{\frac{5 \cdot 3^{k-1}-1}{2}} 2^{\frac{3^k-1}{2}-3k+6} 3^{2k-3}$$

$$\leq \delta W^{\frac{3^{k-1}-1}{2}}(B \vee d)^{\frac{5 \cdot 3^{k-1}-1}{2}} 2^{\frac{3^k-1}{2}-k+1} 3^{2k-2}.$$

where the last inequality above follows from $k \geq 2$. Taking supremum with respect to $x \in \Omega$ on the LHS implies the given upper bound also holds for $k$. By induction, the claim is proved.

In particular, when $k = L$, we have $\nabla F_L(x) = J[F_L](x)^T$ for any $x \in \Omega$. Applying Lemma B.13 then yields:

$$\sup_{x \in \Omega} \left| \|\nabla F_L(x)\| - \|\nabla G_L(x)\| \right| = \sup_{x \in \Omega} \left| \|\nabla J[F_L](x)^T\| - \|\nabla J[G_L](x)^T\| \right|$$

$$\leq \sup_{x \in \Omega} \|J[F_L](x) - J[G_L](x)\|_\infty$$

$$\leq \delta W^{\frac{3^{L-1}-1}{2}}(B \vee d)^{\frac{5 \cdot 3^{L-1}-1}{2}} 2^{\frac{3^L-1}{2}-k+1} 3^{2L-2}.$$

This finishes our proof of the Lemma. $\qquad \square$

**Theorem B.5.** *(Bounding the DNN Gradient space covering number) Fix some sufficiently large $N \in \mathbb{Z}^+$. Consider a Deep Neural Network space $\Phi(L, W, S, B)$ with $L = O(1), W = O(N), S = O(N)$ and $B = O(N)$. Then the $\log$ value of the covering number of the DNN Gradient space with respect to the $\|\cdot\|_\infty$ norm $\|F(x)\|_\infty := \sup_{x \in \Omega} |F(x)|$, which is denoted by $\mathcal{N}(\delta, \nabla \Phi(L, W, S, B), \|\cdot\|_\infty)$, can be upper bounded by:*

$$\log \mathcal{N}(\delta, \nabla \Phi(L, W, S, B), \|\cdot\|_\infty) = O\left(S\left[\log(\delta^{-1}) + 3^L \log(WB)\right]\right). \tag{B.33}$$

*Proof.* We firstly fix a sparsity pattern (i.e, the locations of the non-zero entries are fixed). Using equation B.29 in Lemma B.14, yields the following upper bound on the covering number with respect to $\|\cdot\|_\infty$:

$$\left(\frac{\delta}{W^{\frac{3^{L-1}-1}{2}}(B \vee d)^{\frac{5 \cdot 3^{L-1}-1}{2}} 2^{\frac{3^L-1}{2}-L+1} 3^{2L-2}}\right)^{-S}.$$

Furthermore, note that the number of feasible configurations is upper bounded by: $\binom{(W+1)^L}{S} \leq (W+1)^{LS}$.(Schmidt-Hieber et al., 2020; Farrell et al., 2021). Plug this inequality into the previous estimation then yields:

$$\log \mathcal{N}(\delta, \Phi(L, W, S, B), \|\cdot\|_\infty) \leq \log\left[(W+1)^{LS}\left(\frac{\delta}{W^{\frac{3^{L-1}-1}{2}}(B \vee d)^{\frac{5 \cdot 3^{L-1}-1}{2}} 2^{\frac{3^L-1}{2}-L+1} 3^{2L-2}}\right)^{-S}\right]$$

$$\leq S\log\left[\delta^{-1}(W+1)^L 3^{2L-2} W^{\frac{3^{L-1}-1}{2}}(B \vee d)^{\frac{5 \cdot 3^{L-1}-1}{2}} 2^{\frac{3^L-1}{2}-L+1}\right]$$

$$\lesssim S\left[\log(\delta^{-1}) + 2L\log(3W) + 3^L \log(W(B \vee d)) + 3^L \log 2\right].$$

Note that here the dimension $d$ is some constant. Thus, by plugging in thee given magnitudes $L = O(1), W = O(N), S = O(N)$ and $B = O(N)$, we can further deduce that:

$$\log \mathcal{N}(\delta, \Phi(L, W, S, B), \|\cdot\|_\infty) \lesssim S\left[\log(\delta^{-1}) + 3^L \log(WB)\right].$$

This finishes our proof. $\qquad \square$

Now let's consider upper bounding the covering number of the Laplacian of the sparse Deep Neural Networks. Note that for any $1 \le k \le L - 1$, any $k-$ReLU3 Deep Neural Network $F_k \in \Phi_k(L, W, S, B)$ is a vector-valued function mapping from $\mathbb{R}^d$ to $\mathbb{R}^W$. Moreover, we define the Laplacian of $F_k(x)$, which is denoted by $\Delta[F_k](x)$, as follows:

$$\Delta[F_k](x) = [\Delta F_{k,1}(x), \Delta F_{k,2}(x), \cdots, \Delta F_{k,W}(x)]^T \in \mathbb{R}^W,$$

where for any $1 \le l \le W$, we have:

$$\Delta F_{k,l}(x) = \sum_{j=1}^d \frac{\partial^2}{\partial x_j^2} F_{k,l}(x).$$

In particular, when $k = L$, we have that any $F_L \in \Phi_L(L, W, S, B) = \Phi(L, W, S, B)$ is a scalar-valued function mapping from $\mathbb{R}^d$ to $\mathbb{R}$. Thus, its Laplacian can be explicitly written as:

$$\Delta[F_L](x) = \Delta F_L(x) = \sum_{j=1}^d \frac{\partial^2}{\partial x_j^2} F_L(x).$$

For both Lemma B.15 and Lemma B.16 below, we consider a fixed Deep Neural Network space $\Phi(L, W, S, B)$ with $L = O(1), W = O(N), S = O(N)$ and $B = O(N)$, where $N \in \mathbb{Z}^+$ is fixed and sufficiently large.

**Lemma B.15.** *(Upper bound on $\infty$-norm of Laplacian of elements in the DNN space) For any $1 \le k \le L$, we have the following upper bound:*

$$\sup_{x \in \Omega, F_k \in \Phi_k(L, W, S, B)} \|\Delta[F_k](x)\|_\infty = O\Big(W^{\frac{3^{k-1}-1}{2}} (B \vee d)^{\frac{5 \cdot 3^{k-1}-1}{2}}\Big).$$

*Proof.* We use induction on $k$ to prove the claim.

Base case: $k = 1$. Note that any $F_1 \in \Phi_1(L, W, S, B)$ is a linear transform, so the Laplacian $\Delta[F_1](x)$ must be the zero vector for any $x \in \Omega$. This implies:

$$\|\Delta[F_1](x)\|_\infty = 0 \lesssim (B \vee d)^2.$$

Inductive Step: Assume that the claim has been proved for $k - 1$, where $2 \le k \le L$. For any $x \in \Omega$ and any $F_k \in \Phi_k(L, W, S, B)$, using linearity of the Laplacian operator implies:

$$\Delta[F_k](x) = \mathcal{W}_F^{(k)} \Delta[\eta_3 \circ F_{k-1}](x).$$

Taking the inf-norm on both sides of the identity above implies:

$$\|\Delta[F_k](x)\|_\infty \le \|\mathcal{W}_F^{(k)}\|_\infty \|\Delta[\eta_3 \circ F_{k-1}](x)\|_\infty \le WB \|\Delta[\eta_3 \circ F_{k-1}](x)\|_\infty.$$

It now remains to upper bound the term $\|\Delta[\eta_3 \circ F_{k-1}](x)\|_\infty$. For any $1 \le l \le W$, we will use the Chain Rule to write the $l$-th component $\Big(\Delta[\eta_3 \circ F_{k-1}](x)\Big)_l$ in an explicit form. For any $1 \le j \le d$, we have:

$$\frac{\partial}{\partial x_j} \eta_3[F_{k-1,l}(x)] = 3\eta_2[F_{k-1,l}(x)] \frac{\partial}{\partial x_j} F_{k-1,l}(x).$$

Differentiating with respect to $x_j$ on both sides above yields:

$$\frac{\partial^2}{\partial x_j^2} \eta_3[F_{k-1,l}(x)] = 6\eta_1[F_{k-1,l}(x)] \Big(\frac{\partial}{\partial x_j} F_{k-1,l}(x)\Big)^2 + 3\eta_2[F_{k-1,l}(x)] \frac{\partial^2}{\partial x_j^2} F_{k-1,l}(x). \quad \text{(B.34)}$$

Summing the expression above from $j = 1$ to $j = d$ implies:

$$\Big|\Big(\Delta[\eta_3 \circ F_{k-1}](x)\Big)_l\Big| = \Big|\sum_{j=1}^d \frac{\partial^2}{\partial x_j^2} \eta_3[F_{k-1,l}(x)]\Big|$$

$$= \Big|6\eta_1[F_{k-1,l}(x)] \sum_{j=1}^d \Big(\frac{\partial}{\partial x_j} F_{k-1,l}(x)\Big)^2 + 3\eta_2[F_{k-1,l}(x)] \sum_{j=1}^d \frac{\partial^2}{\partial x_j^2} F_{k-1,l}(x)\Big|$$

$$\le 6\Big|\eta_1[F_{k-1,l}(x)]\Big| \Big(\sum_{j=1}^d \Big|\frac{\partial}{\partial x_j} F_{k-1,l}(x)\Big|\Big)^2 + 3\Big|\eta_2[F_{k-1,l}(x)]\Big| \Big|\sum_{j=1}^d \frac{\partial^2}{\partial x_j^2} F_{k-1,l}(x)\Big|.$$

We denote the two summations above by $U_1$ and $U_2$, respectively:

$$U_1 := 6\Big|\eta_1[F_{k-1,l}(x)]\Big| \left(\sum_{j=1}^{d}\Big|\frac{\partial}{\partial x_j}F_{k-1,l}(x)\Big|\right)^2.$$

$$U_2 := 3\Big|\eta_2[F_{k-1,l}(x)]\Big| \left|\sum_{j=1}^{d}\frac{\partial^2}{\partial x_j^2}F_{k-1,l}(x)\right|.$$

On the one hand, by applying Lemma B.9 and Lemma B.12, we can upper bound $U_1$ by:

$$U_1 \le 6\Big(\sup_{x\in\Omega,\ F_{k-1}\in\Phi_{k-1}(L,W,S,B)}\|F_{k-1}(x)\|_\infty\Big)\Big(\sup_{x\in\Omega,F_{k-1}\in\Phi_{k-1}(L,W,S,B)}\|J[F_{k-1}](x)\|_\infty\Big)^2$$

$$\le 6\times W^{\frac{3^{k-2}-1}{2}}(B\vee d)^{\frac{5\cdot 3^{k-2}-1}{2}}2^{\frac{3^{k-1}-1}{2}-k+2}\times W^{3^{k-2}-1}(B\vee d)^{5\cdot 3^{k-2}-1}2^{3^{k-1}-1-2k+4}3^{2k-4}$$

$$\lesssim W^{\frac{3^{k-1}-3}{2}}(B\vee d)^{\frac{5\cdot 3^{k-1}-3}{2}},$$

where the last step above follows from $k\le L$ and $L=O(1)$.
On the other hand, by applying Lemma B.9 and the inductive hypothesis, we have:

$$U_2 \le 3\Big(\sup_{x\in\Omega,\ F_{k-1}\in\Phi_{k-1}(L,W,S,B)}\|F_{k-1}(x)\|_\infty\Big)^2\|\Delta[F_{k-1}](x)\|_\infty$$

$$\lesssim 3\times W^{3^{k-2}-1}(B\vee d)^{5\cdot 3^{k-2}-1}2^{3^{k-1}-1-2k+4}\times W^{\frac{3^{k-2}-1}{2}}(B\vee d)^{\frac{5\cdot 3^{k-2}-1}{2}}$$

$$\lesssim W^{\frac{3^{k-1}-3}{2}}(B\vee d)^{\frac{5\cdot 3^{k-1}-3}{2}},$$

where the last step above follows from $k\le L$ and $L=O(1)$.
Summing the two bounds on $U_1$ and $U_2$ implies that for any $1\le l\le W$, we have:

$$\left|\Big(\Delta[\eta_3\circ F_{k-1}](x)\Big)_l\right| \le U_1+U_2 \lesssim W^{\frac{3^{k-1}-3}{2}}(B\vee d)^{\frac{5\cdot 3^{k-1}-3}{2}}. \tag{B.35}$$

Taking supremum with respect to $1\le l\le W$ then yields:

$$\|\Delta[F_k](x)\|_\infty \le WB\|\Delta[\eta_3\circ F_{k-1}](x)\|_\infty \lesssim W^{\frac{3^{k-1}-1}{2}}(B\vee d)^{\frac{5\cdot 3^{k-1}-1}{2}}.$$

Taking supremum with respect to $x\in\Omega$ and $F_k\in\Phi_k(L,W,S,B)$ on the LHS implies that the given upper bound also holds for $k$. By induction, the claim is proved. $\square$

**Lemma B.16.** *(Relation between the covering number of the DNN Laplacian space and parameter space) For any $1\le k\le L$, suppose that a pair of different two networks $F_k, G_k\in\Phi_k(L,W,S,B)$ are given by:*

$$F_k(x) := (\mathcal{W}_F^{(k)}\eta_3(\cdot)+b_F^{(k)})\cdots(\mathcal{W}_F^{(1)}x+b_F^{(1)}),$$
$$G_k(x) := (\mathcal{W}_G^{(k)}\eta_3(\cdot)+b_G^{(k)})\cdots(\mathcal{W}_G^{(1)}x+b_G^{(1)}).$$

*Furthermore, assume that the $\|\ \|_\infty$ norm of the distance between the parameter spaces is uniformly upper bounded by $\delta$, i.e*

$$\|W_F^{(l)}-W_G^{(l)}\|_{\infty,\infty}\le\delta,\ \|b_F^{(l)}-b_G^{(l)}\|_\infty\le\delta,\ (\forall\ 1\le l\le k). \tag{B.36}$$

*Then we have:*

$$\sup_{x\in\Omega}\|\Delta[F_k](x)-\Delta[G_k](x)\|_\infty = O\Big(\delta W^{\frac{3^{k-1}-1}{2}}(B\vee d)^{\frac{5\cdot 3^{k-1}-1}{2}}\Big). \tag{B.37}$$

*Proof.* We use induction on $k$ to prove the claim.
Base case: $k=1$. Note that any $F_1\in\Phi_1(L,W,S,B)$ is a linear transform, so the Laplacian $\Delta[F_1](x)$ must be the zero vector for any $x\in\Omega$. Hence, for any $x\in\Omega$ and any $F_1, G_1\in\Phi_1(L,W,S,B)$, we have:

$$\|\Delta[F_1](x)-\Delta[G_1](x)\|_\infty = 0 \lesssim \delta(B\vee d)^2.$$

Inductive Step: assume that the claim has been proved for $k-1$, where $2 \leq k \leq L$. Then for any $x \in \Omega$ and $F_k, G_k \in \Phi_k(L, W, S, B)$ satisfying constraint B.36, applying linearity of the Laplacian operator indicates:

$$
\begin{aligned}
\|\Delta[F_k](x) - \Delta[G_k](x)\|_\infty &= \|\mathcal{W}_F^{(k)}\Delta[\eta_3 \circ F_{k-1}](x) - \mathcal{W}_G^{(k)}\Delta[\eta_3 \circ G_{k-1}](x)\|_\infty \\
&= \left\|\left(\mathcal{W}_F^{(k)} - \mathcal{W}_G^{(k)}\right)\Delta[\eta_3 \circ F_{k-1}](x)\right\|_\infty \\
&\quad + \left\|\mathcal{W}_G^{(k)}\left(\Delta[\eta_3 \circ F_{k-1}](x) - \Delta[\eta_3 \circ G_{k-1}](x)\right)\right\|_\infty \\
&\leq \|\mathcal{W}_F^{(k)} - \mathcal{W}_G^{(k)}\|_\infty\|\Delta[\eta_3 \circ F_{k-1}](x)\|_\infty \\
&\quad + \|\mathcal{W}_G^{(k)}\|_\infty\|\Delta[\eta_3 \circ F_{k-1}](x) - \Delta[\eta_3 \circ G_{k-1}](x)\|_\infty.
\end{aligned}
$$

For the first term $\|\mathcal{W}_F^{(k)} - \mathcal{W}_G^{(k)}\|_\infty\|\Delta[\eta_3 \circ F_{k-1}](x)\|_\infty$, applying the bound in equation B.35 and equation B.36 yields:

$$
\begin{aligned}
\|\mathcal{W}_F^{(k)} - \mathcal{W}_G^{(k)}\|_\infty\|\Delta[\eta_3 \circ F_{k-1}](x)\|_\infty &\lesssim \delta W \times W^{\frac{3^{k-1}-3}{2}}(B \vee d)^{\frac{5 \cdot 3^{k-1}-3}{2}} \\
&= \delta W^{\frac{3^{k-1}-1}{2}}(B \vee d)^{\frac{5 \cdot 3^{k-1}-3}{2}}.
\end{aligned} \tag{B.38}
$$

For the second term $\|\mathcal{W}_G^{(k)}\|_\infty\|\Delta[\eta_3 \circ F_{k-1}](x) - \Delta[\eta_3 \circ G_{k-1}](x)\|_\infty$, we need to upper bound the norm $\|\Delta[\eta_3 \circ F_{k-1}](x) - \Delta[\eta_3 \circ G_{k-1}](x)\|_\infty$ at first. Note that for any $1 \leq l \leq W$, we can use equation B.34 to write the $l$-th component of $\Delta[\eta_3 \circ F_{k-1}](x) - \Delta[\eta_3 \circ G_{k-1}](x)$ as:

$$
\begin{aligned}
\left(\Delta[\eta_3 \circ F_{k-1}](x) - \Delta[\eta_3 \circ G_{k-1}](x)\right)_l &= \sum_{j=1}^d \frac{\partial^2}{\partial x_j^2}\eta_3[F_{k-1,l}(x)] - \sum_{j=1}^d \frac{\partial^2}{\partial x_j^2}\eta_3[G_{k-1,l}(x)] \\
&= 6\eta_1[F_{k-1,l}(x)]\sum_{j=1}^d\left(\frac{\partial}{\partial x_j}F_{k-1,l}(x)\right)^2 - 6\eta_1[G_{k-1,l}(x)]\sum_{j=1}^d\left(\frac{\partial}{\partial x_j}G_{k-1,l}(x)\right)^2 \\
&\quad + 3\eta_2[F_{k-1,l}(x)]\sum_{j=1}^d\frac{\partial^2}{\partial x_j^2}F_{k-1,l}(x) - 3\eta_2[G_{k-1,l}(x)]\sum_{j=1}^d\frac{\partial^2}{\partial x_j^2}G_{k-1,l}(x) \\
&= 6\eta_1[F_{k-1,l}(x)]\sum_{j=1}^d\left(\frac{\partial}{\partial x_j}F_{k-1,l}(x)\right)^2 - 6\eta_1[G_{k-1,l}(x)]\sum_{j=1}^d\left(\frac{\partial}{\partial x_j}F_{k-1,l}(x)\right)^2 \\
&\quad + 6\eta_1[G_{k-1,l}(x)]\sum_{j=1}^d\left(\frac{\partial}{\partial x_j}F_{k-1,l}(x)\right)^2 - 6\eta_1[G_{k-1,l}(x)]\sum_{j=1}^d\left(\frac{\partial}{\partial x_j}G_{k-1,l}(x)\right)^2 \\
&\quad + 3\eta_2[F_{k-1,l}(x)]\sum_{j=1}^d\frac{\partial^2}{\partial x_j^2}F_{k-1,l}(x) - 3\eta_2[G_{k-1,l}(x)]\sum_{j=1}^d\frac{\partial^2}{\partial x_j^2}F_{k-1,l}(x) \\
&\quad + 3\eta_2[G_{k-1,l}(x)]\sum_{j=1}^d\frac{\partial^2}{\partial x_j^2}F_{k-1,l}(x) - 3\eta_2[G_{k-1,l}(x)]\sum_{j=1}^d\frac{\partial^2}{\partial x_j^2}G_{k-1,l}(x).
\end{aligned}
$$

We denote the four summations above by $V_1, V_2, V_3$ and $V_4$, respectively:

$$
V_1 := 6\eta_1[F_{k-1,l}(x)]\sum_{j=1}^d\left(\frac{\partial}{\partial x_j}F_{k-1,l}(x)\right)^2 - 6\eta_1[G_{k-1,l}(x)]\sum_{j=1}^d\left(\frac{\partial}{\partial x_j}F_{k-1,l}(x)\right)^2,
$$

$$
V_2 := 6\eta_1[G_{k-1,l}(x)]\sum_{j=1}^d\left(\frac{\partial}{\partial x_j}F_{k-1,l}(x)\right)^2 - 6\eta_1[G_{k-1,l}(x)]\sum_{j=1}^d\left(\frac{\partial}{\partial x_j}G_{k-1,l}(x)\right)^2,
$$

$$
V_3 := 3\eta_2[F_{k-1,l}(x)]\sum_{j=1}^d\frac{\partial^2}{\partial x_j^2}F_{k-1,l}(x) - 3\eta_2[G_{k-1,l}(x)]\sum_{j=1}^d\frac{\partial^2}{\partial x_j^2}F_{k-1,l}(x),
$$

$$
V_4 := 3\eta_2[G_{k-1,l}(x)]\sum_{j=1}^d\frac{\partial^2}{\partial x_j^2}F_{k-1,l}(x) - 3\eta_2[G_{k-1,l}(x)]\sum_{j=1}^d\frac{\partial^2}{\partial x_j^2}G_{k-1,l}(x).
$$

By applying Lemma B.9, Lemma B.10, Lemma B.11 and Lemma B.12, we can upper bound $V_1$ by:

$$V_1 = 6\Big(\eta_1[F_{k-1,l}(x)] - \eta_1[G_{k-1,l}(x)]\Big) \sum_{j=1}^{d} \Big(\frac{\partial}{\partial x_j} F_{k-1,l}(x)\Big)^2$$

$$\leq 6|F_{k-1,l}(x) - G_{k-1,l}(x)| \left(\sum_{j=1}^{d}\Big|\frac{\partial}{\partial x_j}F_{k-1,l}(x)\Big|\right)^2 \leq 6\|F_{k-1}(x) - G_{k-1}(x)\|_\infty \|J[F_{k-1}](x)\|_\infty^2$$

$$\lesssim \delta W^{\frac{3^{k-2}-1}{2}}(B \vee d)^{\frac{5\cdot 3^{k-2}-1}{2}} 2^{\frac{3^{k-1}-1}{2}-k+2} 3^{k-2} \times W^{3^{k-2}-1}(B \vee d)^{5\cdot 3^{k-2}-1} 2^{3^{k-1}-1-2k+4} 3^{2k-4}$$

$$\lesssim \delta W^{\frac{3^{k-1}-3}{2}}(B \vee d)^{\frac{5\cdot 3^{k-1}-3}{2}}.$$

where the last step above follows from $k \leq L$ and $L = O(1)$.

Furthermore, note that for any $1 \leq j \leq d$, we can upper bound the difference $\Big(\frac{\partial}{\partial x_j}F_{k-1,l}(x)\Big)^2 - \Big(\frac{\partial}{\partial x_j}G_{k-1,l}(x)\Big)^2$ as follows:

$$\Big(\frac{\partial}{\partial x_j}F_{k-1,l}(x)\Big)^2 - \Big(\frac{\partial}{\partial x_j}G_{k-1,l}(x)\Big)^2 \leq \left|\Big(\frac{\partial}{\partial x_j}F_{k-1,l}(x)\Big)^2 - \Big(\frac{\partial}{\partial x_j}G_{k-1,l}(x)\Big)^2\right|$$

$$= \left|\frac{\partial}{\partial x_j}F_{k-1,l}(x) + \frac{\partial}{\partial x_j}G_{k-1,l}(x)\right|\left|\frac{\partial}{\partial x_j}F_{k-1,l}(x) - \frac{\partial}{\partial x_j}G_{k-1,l}(x)\right| \tag{B.39}$$

$$\leq \left(\left|\frac{\partial}{\partial x_j}F_{k-1,l}(x)\right| + \left|\frac{\partial}{\partial x_j}G_{k-1,l}(x)\right|\right)\left|\frac{\partial}{\partial x_j}F_{k-1,l}(x) - \frac{\partial}{\partial x_j}G_{k-1,l}(x)\right|.$$

Note that $\eta_1(G_{k-1,l}(x)) \geq 0$. Combining the non-negativity with equation B.39, Lemma B.9, Lemma B.12 and Lemma B.14 helps us upper bound $V_2$ by:

$$V_2 = 6\eta_1[G_{k-1,l}(x)]\sum_{j=1}^{d}\left[\Big(\frac{\partial}{\partial x_j}F_{k-1,l}(x)\Big)^2 - \Big(\frac{\partial}{\partial x_j}G_{k-1,l}(x)\Big)^2\right]$$

$$\leq 6\|G_{k-1}(x)\|_\infty \sum_{j=1}^{d}\left(\left|\frac{\partial}{\partial x_j}F_{k-1,l}(x)\right| + \left|\frac{\partial}{\partial x_j}G_{k-1,l}(x)\right|\right)\left|\frac{\partial}{\partial x_j}F_{k-1,l}(x) - \frac{\partial}{\partial x_j}G_{k-1,l}(x)\right|$$

$$\leq 6\|G_{k-1}(x)\|_\infty\left(\sum_{j=1}^{d}\left|\frac{\partial}{\partial x_j}F_{k-1,l}(x)\right| + \sum_{j=1}^{d}\left|\frac{\partial}{\partial x_j}G_{k-1,l}(x)\right|\right)\left(\sum_{j=1}^{d}\left|\frac{\partial}{\partial x_j}F_{k-1,l}(x) - \frac{\partial}{\partial x_j}G_{k-1,l}(x)\right|\right)$$

$$\leq 6\|G_{k-1}(x)\|_\infty\Big(\|J[F_{k-1}](x)\|_\infty + \|J[G_{k-1}](x)\|_\infty\Big)\Big\|J[F_{k-1}](x) - J[G_{k-1}](x)\Big\|_\infty$$

$$\leq 6W^{\frac{3^{k-2}-1}{2}}(B \vee d)^{\frac{5\cdot 3^{k-2}-1}{2}}2^{\frac{3^{k-1}-1}{2}-k+2} \times 2W^{\frac{3^{k-2}-1}{2}}(B \vee d)^{\frac{5\cdot 3^{k-2}-1}{2}}2^{\frac{3^{k-1}-1}{2}-k+2}3^{k-2}$$

$$\times \delta W^{\frac{3^{k-2}-1}{2}}(B \vee d)^{\frac{5\cdot 3^{k-2}-1}{2}}2^{\frac{3^{k-1}-1}{2}-k+1}3^{2k-4} \lesssim \delta W^{\frac{3^{k-1}-3}{2}}(B \vee d)^{\frac{5\cdot 3^{k-1}-3}{2}}.$$

where the last step above follows from $k \leq L$ and $L = O(1)$.
Moreover, using Lemma B.9, Lemma B.10 and Lemma B.15 helps us upper bound $V_3$ by:

$$V_3 = \Big(3\eta_2[F_{k-1,l}(x)] - 3\eta_2[G_{k-1,l}(x)]\Big)\sum_{j=1}^{d}\frac{\partial^2}{\partial x_j^2}F_{k-1,l}(x)$$

$$\leq \left|3\eta_2[F_{k-1,l}(x)] - 3\eta_2[G_{k-1,l}(x)]\right|\left|\sum_{j=1}^{d}\frac{\partial^2}{\partial x_j^2}F_{k-1,l}(x)\right|$$

$$\leq 6\Big(\sup_{x\in\Omega,\ F_{k-1}\in\Phi_{k-1}(L,W,S,B)}\|F_{k-1}(x)\|_\infty\Big)\|F_{k-1}(x) - G_{k-1}(x)\|_\infty\|\Delta[F_{k-1}](x)\|_\infty$$

$$\lesssim 6W^{\frac{3^{k-2}-1}{2}}(B \vee d)^{\frac{5\cdot 3^{k-2}-1}{2}}2^{\frac{3^{k-1}-1}{2}-k+2} \times \delta W^{\frac{3^{k-2}-1}{2}}(B \vee d)^{\frac{5\cdot 3^{k-2}-1}{2}}2^{\frac{3^{k-1}-1}{2}-k+2}3^{k-2}$$

$$\times W^{\frac{3^{k-2}-1}{2}}(B \vee d)^{\frac{5\cdot 3^{k-2}-1}{2}} \lesssim \delta W^{\frac{3^{k-1}-3}{2}}(B \vee d)^{\frac{5\cdot 3^{k-1}-3}{2}}.$$

where the last step above follows from $k \leq L$ and $L = O(1)$.

Finally, applying Lemma B.9 and inductive hypothesis helps us upper bound $V_4$ by:

$$V_4 = 3\eta_2[G_{k-1,l}(x)]\Big(\sum_{j=1}^{d}\frac{\partial^2}{\partial x_j^2}F_{k-1,l}(x) - \sum_{j=1}^{d}\frac{\partial^2}{\partial x_j^2}G_{k-1,l}(x)\Big)$$

$$\leq 3\|G_{k-1}(x)\|_\infty^2\|\Delta[F_{k-1}](x) - \Delta[G_{k-1}](x)\|_\infty$$

$$\lesssim 3W^{3^{k-2}-1}(B \vee d)^{5\cdot 3^{k-2}-1}2^{3^{k-1}-1-2k+4} \times \delta W^{\frac{3^{k-2}-1}{2}}(B \vee d)^{\frac{5\cdot 3^{k-2}-1}{2}}$$

$$\lesssim \delta W^{\frac{3^{k-1}-3}{2}}(B \vee d)^{\frac{5\cdot 3^{k-1}-3}{2}}.$$

where the last step above follows from $k \leq L$ and $L = O(1)$.

Combining the four bounds on $V_1, V_2, V_3$ and $V_4$ implies:

$$\Big(\Delta[\eta_3 \circ F_{k-1}](x) - \Delta[\eta_3 \circ G_{k-1}](x)\Big)_l = \sum_{i=1}^{4}V_i \lesssim \delta W^{\frac{3^{k-1}-3}{2}}(B \vee d)^{\frac{5\cdot 3^{k-1}-3}{2}}.$$

Taking supremum with respect to $1 \leq l \leq W$ gives us an upper bound on the second term $\|\mathcal{W}_G^{(k)}\|_\infty\|\Delta[\eta_3 \circ F_{k-1}](x) - \Delta[\eta_3 \circ G_{k-1}](x)\|_\infty$:

$$\|\mathcal{W}_G^{(k)}\|_\infty\|\Delta[\eta_3 \circ F_{k-1}](x) - \Delta[\eta_3 \circ G_{k-1}](x)\|_\infty \lesssim WB \times \delta W^{\frac{3^{k-1}-3}{2}}(B \vee d)^{\frac{5\cdot 3^{k-1}-3}{2}}$$

$$= \delta W^{\frac{3^{k-1}-1}{2}}(B \vee d)^{\frac{5\cdot 3^{k-1}-1}{2}}. \tag{B.40}$$

Combining the two bounds derived in equation B.38 and equation B.40 then implies:

$$\|\Delta[F_k](x) - \Delta[G_k](x)\|_\infty \lesssim \delta W^{\frac{3^{k-1}-1}{2}}(B \vee d)^{\frac{5\cdot 3^{k-1}-3}{2}} + \delta W^{\frac{3^{k-1}-1}{2}}(B \vee d)^{\frac{5\cdot 3^{k-1}-1}{2}}$$

$$\lesssim \delta W^{\frac{3^{k-1}-1}{2}}(B \vee d)^{\frac{5\cdot 3^{k-1}-1}{2}}.$$

Taking supremum with respect to $x \in \Omega$ on the LHS implies that the given upper bound also holds for $k$. By induction, the claim is proved. $\qquad\square$

Given a Neural Network function space $\Phi(L, W, S, B)$, we define a corresponding Neural Network Laplacian space $\Delta\Phi(L, W, S, B)$ as:

$$\Delta\Phi(L, W, S, B) := \{\Delta F \mid F \in \Phi(L, W, S, B)\}. \tag{B.41}$$

**Theorem B.6.** *(Bounding the Neural Network Laplacian space covering number) Fix some suffi-ciently large $N \in \mathbb{Z}^+$. Consider a Deep Neural Network space $\Phi(L, W, S, B)$ with $L = O(1), W = O(N), S = O(N)$ and $B = O(N)$. Then the* log *value of the covering number of the DNN Laplacian space with respect to the $\|\cdot\|_\infty$ norm $\|F(x)\|_\infty := \sup_{x\in\Omega}|F(x)|$, which is denoted by $\mathcal{N}(\delta, \Delta\Phi(L, W, S, B), \|\cdot\|_\infty)$, can be upper bounded by:*

$$\log\mathcal{N}(\delta, \Delta\Phi(L, W, S, B), \|\cdot\|_\infty) = O\Big(S\big[\log(\delta^{-1}) + 3^L\log(WB)\big]\Big). \tag{B.42}$$

*Proof.* We firstly fix a sparsity pattern (i.e, the locations of the non-zero entries are fixed). Applying Lemma B.16 yields that there exists some constant $C = O(1)$, such that the covering number with respect to $\|\cdot\|_\infty$ can be upper bounded by:

$$\Big(\frac{\delta}{CW^{\frac{3^{L-1}-1}{2}}(B \vee d)^{\frac{5\cdot 3^{L-1}-1}{2}}}\Big)^{-S}.$$

Furthermore, note that the number of feasible configurations is upper bounded by $\binom{(W+1)^L}{S} \leq (W+1)^{LS}$ (Schmidt-Hieber et al., 2020; Farrell et al., 2021). Then we plug this into the pervious estimation and yields:

$$\log\mathcal{N}(\delta, \Phi(L, W, S, B), \|\cdot\|_\infty) \leq \log\Big[(W+1)^{LS}\Big(\frac{\delta}{CW^{\frac{3^{L-1}-1}{2}}(B \vee d)^{\frac{5\cdot 3^{L-1}-1}{2}}}\Big)^{-S}\Big]$$

$$\leq S\log\Big[\delta^{-1}(W+1)^L W^{\frac{3^{L-1}-1}{2}}(B \vee d)^{\frac{5\cdot 3^{L-1}-1}{2}}\Big]$$

$$\lesssim S\Big[\log(\delta^{-1}) + L\log(W) + 3^L\log(W(B \vee d))\Big].$$

Note that here the dimension $d$ is some constant. Thus, by plugging in thee given magnitudes $L = O(1), W = O(N), S = O(N)$ and $B = O(N)$, we can further deduce that:

$$\log \mathcal{N}(\delta, \Delta\Phi(L, W, S, B), \|\cdot\|_\infty) \lesssim S\Big[\log(\delta^{-1}) + 3^L \log(WB)\Big].$$

This finishes our proof. □

**Lemma B.17** (Local Rademacher Complexity Bound of DNN Estimator for Deep Ritz Method).
*Consider a Deep Neural Network space $\boldsymbol{F}(\Omega) = \Phi(L, W, S, B)$ with $L = O(1), W = O(N), S = O(N)$ and $B = O(N)$, where $N \in \mathbb{Z}^+$ is fixed to be sufficiently large. Moreover, assume that the gradients and function value of $\boldsymbol{F}(\Omega), u^*, V$ and $f$ are uniformly bounded*

$$\max\Big\{ \sup_{u\in\boldsymbol{F}(\Omega)} \|u\|_{L^\infty(\Omega)}, \sup_{u\in\boldsymbol{F}(\Omega)} \|\nabla u\|_{L^\infty(\Omega)}, \|u^*\|_{L^\infty(\Omega)}, \|\nabla u^*\|_{L^\infty(\Omega)}, V_{max}, \|f\|_{L^\infty(\Omega)} \Big\} \leq C.$$
(B.43)

*For any $\rho > 0$, we consider a localized set $\boldsymbol{L}_\rho$ defined by:*

$$\boldsymbol{L}_\rho(\Omega) := \{u : u \in \boldsymbol{F}(\Omega), \|u - u^*\|_{H^1}^2 \leq \rho\}.$$

*Then for any $\rho \gtrsim n^{-2}$, the Rademacher complexity of a localized function space $\boldsymbol{S}_\rho(\Omega) := \Big\{ h := |\Omega| \cdot \Big[ \frac{1}{2}\Big(\|\nabla u\|^2 - \|\nabla u^*\|^2\Big) + \frac{1}{2}V(|u|^2 - |u^*|^2) - f(u - u^*) \Big] \ \Big| \ u \in \boldsymbol{L}_\rho(\Omega) \Big\}$ can be upper bounded by a sub-root function*

$$\phi(\rho) := O\left( \sqrt{\frac{S3^L\rho}{n} \log\left(BWn\right)} \right).$$

i.e. *we have*

$$\phi(4\rho) \leq 2\phi(\rho) \text{ and } R_n(\boldsymbol{S}_\rho(\Omega)) \leq \phi(\rho).$$
(B.44)

*holds for all $\rho \gtrsim n^{-2}$.*

*Proof.* Firstly, we will check that for any $u \in \boldsymbol{L}_\rho(\Omega)$, the corresponding function $h$ in $\boldsymbol{S}_\rho(\Omega)$ is Lipschitz with respect to $u - u^*$ and $\|\nabla u\| - \|\nabla u^*\|$. Note that for any $u_1, u_2 \in \boldsymbol{L}_\rho(\Omega)$ with corresponding functions $h_1, h_2 \in \boldsymbol{S}_\rho(\Omega)$, applying boundedness condition B.43 yields:

$$|h_1(x) - h_2(x)| \leq \frac{1}{2}\Big|\|\nabla u_1(x)\|^2 - \|\nabla u_2(x)\|^2\Big| + \frac{1}{2}|V(x)||u_1(x)^2 - u_2(x)^2| + |f(x)||u_1(x) - u_2(x)|$$

$$\leq C\Big|\|\nabla u_1(x)\| - \|\nabla u_2(x)\|\Big| + (C^2 + C)|u_1(x) - u_2(x)|$$

$$= C\left|\Big(\|\nabla u_1(x)\| - \|\nabla u^*(x)\|\Big) - \Big(\|\nabla u_2(x)\| - \|\nabla u^*(x)\|\Big)\right|$$

$$+ (C^2 + C)\Big|(u_1(x) - u^*(x)) - (u_2(x) - u^*(x))\Big|.$$

Let's pick $L = C^2 + C > C$. Applying the Talagrand Contraction Lemma B.2 helps us upper bound the local Rademacher complexity $R_n(\boldsymbol{S}_\rho(\Omega))$ by

$$R_n(\boldsymbol{S}_\rho(\Omega)) = \mathbb{E}_x\mathbb{E}_\sigma \left[ \sup_{u\in\boldsymbol{L}_\rho(\Omega)} \frac{1}{n}\sum_{i=1}^n \sigma_i\Big[\frac{1}{2}\Big(\|\nabla u\|^2 - \|\nabla u^*\|^2\Big) + \frac{1}{2}V(|u|^2 - |u^*|^2) - f(u - u^*)\Big] \right]$$

$$\leq 2L\mathbb{E}_x\mathbb{E}_\sigma \left[ \sup_{u\in\boldsymbol{L}_\rho(\Omega)} \frac{1}{n}\sum_{i=1}^n \sigma_i\Big(u(x_i) - u^*(x_i)\Big) \right]$$

$$+ 2L\mathbb{E}_{x'}\mathbb{E}_{\sigma'} \left[ \sup_{u\in\boldsymbol{L}_\rho(\Omega)} \frac{1}{n}\sum_{i=1}^n \sigma_i'\Big(\|\nabla u(x_i')\| - \|\nabla u^*(x_i')\|\Big) \right]$$

$$\lesssim R_n\left(\Big\{u - u_* : u \in \boldsymbol{L}_\rho\Big\}\right) + R_n\left(\Big\{\|\nabla u\| - \|\nabla u^*\| : u \in \boldsymbol{L}_\rho\Big\}\right).$$

From the localization constraint $\rho \geq \|u - u^*\|_{H^1(\Omega)}^2 = \|u - u^*\|_{L^2(\Omega)}^2 + \|\nabla u - \nabla u^*\|_{L^2(\Omega)}^2$, we can deduce that

$$\|u - u^*\|_{L^2(\Omega)} \leq \sqrt{\rho} \text{ and } \|\nabla u - \nabla u^*\|_{L^2(\Omega)} \leq \sqrt{\rho}.$$
(B.45)

Moreover, note that $\Omega \subset [0,1]^d$. Applying triangle inequality yields:

$$\left\| \|\nabla u\| - \|\nabla u^*\| \right\|_{L^2(\Omega)}^2 = \int_\Omega \left| \|\nabla u(x)\| - \|\nabla u^*(x)\| \right|^2 dx \leq \int_\Omega \|\nabla u(x) - \nabla u^*(x)\|^2 dx$$

$$= \|\nabla u - \nabla u^*\|_{L^2(\Omega)}^2 \leq \rho \Rightarrow \left\| \|\nabla u\| - \|\nabla u^*\| \right\|_{L^2(\Omega)} \leq \sqrt{\rho}.$$
(B.46)

Using inequality B.45 and inequality B.46, we have:

$$R_n(\boldsymbol{S}_\rho(\Omega)) \lesssim R_n\left(\left\{u - u_* : u \in \boldsymbol{L}_\rho\right\}\right) + R_n\left(\left\{\|\nabla u\| - \|\nabla u^*\| : u \in \boldsymbol{L}_\rho\right\}\right)$$

$$\leq R_n\left(\left\{u - u^* : u \in \Phi(L,W,S,B), \|u - u^*\|_{L^2(\Omega)} \leq \sqrt{\rho}\right\}\right)$$

$$+ R_n\left(\left\{\|\nabla u\| - \|\nabla u^*\| : u \in \Phi(L,W,S,B), \left\|\|\nabla u\| - \|\nabla u^*\|\right\|_{L^2(\Omega)} \leq \sqrt{\rho}\right\}\right).$$

(Geer & van de Geer, 2000; Rakhlin et al., 2017) showed a "upper isometry" property, where the metric $\|\cdot\|_{L_2}$ is equivalent to $\|\cdot\|_{n,2}$ with high probability. Combining this fact with Theorem B.3, we can bound the local Rademacher complexities using Dudley integral:

$$R_n(\boldsymbol{S}_\rho(\Omega)) \lesssim R_n\left(\left\{u - u^* : u \in \Phi(L,W,S,B), \|u - u^*\|_{L^2(\Omega)} \leq \sqrt{\rho}\right\}\right)$$

$$+ R_n\left(\left\{\|\nabla u\| - \|\nabla u^*\| : u \in \Phi(L,W,S,B), \left\|\|\nabla u\| - \|\nabla u^*\|\right\|_{L^2(\Omega)} \leq \sqrt{\rho}\right\}\right)$$

$$\leq R_n\left(\left\{u - u^* : u \in \Phi(L,W,S,B), \|u - u^*\|_{n,2} \leq 2\sqrt{\rho}\right\}\right)$$

$$+ R_n\left(\left\{\|\nabla u\| - \|\nabla u^*\| : u \in \Phi(L,W,S,B), \left\|\|\nabla u\| - \|\nabla u^*\|\right\|_{n,2} \leq 2\sqrt{\rho}\right\}\right)$$

$$\lesssim \inf_{0<\alpha<2\sqrt{\rho}} \left\{4\alpha + \frac{12}{\sqrt{n}} \int_\alpha^{2\sqrt{\rho}} \sqrt{\log \mathcal{N}(\delta, \Phi(L,W,S,B), \|\cdot\|_{n,2})} d\delta\right\}$$

$$+ \inf_{0<\alpha<2\sqrt{\rho}} \left\{4\alpha + \frac{12}{\sqrt{n}} \int_\alpha^{2\sqrt{\rho}} \sqrt{\log \mathcal{N}(\delta, \nabla\Phi(L,W,S,B), \|\cdot\|_{n,2})} d\delta\right\}$$

$$\lesssim \inf_{0<\alpha<2\sqrt{\rho}} \left\{4\alpha + \frac{12}{\sqrt{n}} \int_\alpha^{2\sqrt{\rho}} \sqrt{\log \mathcal{N}(\delta, \Phi(L,W,S,B), \|\cdot\|_\infty)} d\delta\right\}$$

$$+ \inf_{0<\alpha<2\sqrt{\rho}} \left\{4\alpha + \frac{12}{\sqrt{n}} \int_\alpha^{2\sqrt{\rho}} \sqrt{\log \mathcal{N}(\delta, \nabla\Phi(L,W,S,B), \|\cdot\|_\infty)} d\delta\right\}.$$

For any $\rho \gtrsim \frac{1}{n^2}$, we pick $\alpha = \frac{1}{n} \lesssim \sqrt{\rho}$ and plug in the upper bounds proved in Theorem B.4 and Theorem B.5, which implies:

$$R_n(\boldsymbol{S}_\rho(\Omega)) \lesssim \frac{1}{n} + \frac{1}{\sqrt{n}} \int_{\frac{1}{n}}^{2\sqrt{\rho}} \sqrt{S\left[\log(\delta^{-1}) + 3^L \log(WB)\right]} d\delta + \frac{1}{\sqrt{n}} \int_{\frac{1}{n}}^{2\sqrt{\rho}} \sqrt{S\left[\log(\delta^{-1}) + 3^L \log(WB)\right]} d\delta$$

$$\lesssim \sqrt{\frac{S3^L\rho}{n} \log(BWn)}.$$

$\square$

**Lemma B.18** (Local Rademacher Complexity Bound of DNN Estimator for Physics Informed Neural Network). *Consider a Deep Neural Network space $\boldsymbol{F}(\Omega) = \Phi(L,W,S,B)$ with $L = O(1), W = O(N), S = O(N)$ and $B = O(N)$, where $N \in \mathbb{Z}^+$ is fixed to be sufficiently large. Moreover, assume that the gradients and function value of $\boldsymbol{F}(\Omega), u^*, V$ and $f$ are uniformly bounded*

$$\max\left\{\sup_{u\in\boldsymbol{F}(\Omega)} \|u\|_{L^\infty(\Omega)}, \sup_{u\in\boldsymbol{F}(\Omega)} \|\Delta u\|_{L^\infty(\Omega)}, \|u^*\|_{L^\infty(\Omega)}, \|\Delta u^*\|_{L^\infty(\Omega)}, V_{max}, \|f\|_{L^\infty(\Omega)}\right\} \leq C.$$
(B.47)

*For any $\rho > 0$, we consider a localized set $\boldsymbol{M}_\rho$ defined by:*

$$\boldsymbol{M}_\rho(\Omega) := \{u : u \in \boldsymbol{F}(\Omega), \|u - u^*\|_{H^2}^2 \leq \rho\}.$$

*Then for any $\rho \gtrsim n^{-2}$, the Rademacher complexity of a localized function space $\boldsymbol{T}_\rho(\Omega) := \Big\{ h :=$ $|\Omega| \cdot \big[ (\Delta u - Vu + f)^2 - (\Delta u^* - Vu^* + f)^2 \big] \; \Big| \; u \in \boldsymbol{M}_\rho(\Omega) \Big\}$ can be upper bounded by a sub-root function*

$$\phi(\rho) := O\left( \sqrt{\frac{S3^L\rho}{n} \log(BWn)} \right).$$

i.e. *we have*

$$\phi(4\rho) \leq 2\phi(\rho) \text{ and } R_n(\boldsymbol{T}_\rho(\Omega)) \leq \phi(\rho). \tag{B.48}$$

*holds for all $\rho \gtrsim n^{-2}$.*

*Proof.* Firstly, we will check that for any $u \in L_\rho(\Omega)$, the corresponding function $h$ in $\boldsymbol{S}_\rho(\Omega)$ is Lipschitz with respect to $u - u^*$ and $\Delta u - \Delta u^*$. Note that for any $u_1, u_2 \in L_\rho(\Omega)$ with corresponding functions $h_1, h_2 \in \boldsymbol{S}_\rho(\Omega)$, applying boundedness condition B.47 yields:

$$
\begin{aligned}
|h_1(x) - h_2(x)| &\leq |\Delta u_1 - \Delta u_2 - V(u_1 - u_2)||\Delta u_1 - Vu_1 + \Delta u_2 - Vu_2 + 2f| \\
&\leq (2C^2 + 4C)\left(|\Delta u_1 - \Delta u_2| + C|u_1 - u_2|\right) \\
&= (2C^2 + 4C)\Big|(\Delta u_1(x) - \Delta u^*(x)) - (\Delta u_2(x) - \Delta u^*(x))\Big| \\
&\quad + (2C^3 + 4C^2)\Big|(u_1(x) - u^*(x)) - (u_2(x) - u^*(x))\Big|.
\end{aligned}
$$

Let's pick $L = \max\{2C^2 + 4C, 2C^3 + 4C^2\}$. Applying the Talagrand Contraction Lemma B.2 helps us upper bound the local Rademacher complexity $R_n(\boldsymbol{T}_\rho(\Omega))$ by

$$
\begin{aligned}
R_n(\boldsymbol{T}_\rho(\Omega)) &= \mathbb{E}_x\mathbb{E}_\sigma\left[ \sup_{u \in \boldsymbol{M}_\rho(\Omega)} \frac{1}{n}\sum_{i=1}^n \sigma_i \left[ (\Delta u - Vu + f)^2 - (\Delta u^* - Vu^* + f)^2 \right] \right] \\
&\leq 2L\mathbb{E}_x\mathbb{E}_\sigma\left[ \sup_{u \in \boldsymbol{M}_\rho(\Omega)} \frac{1}{n}\sum_{i=1}^n \sigma_i\Big( u(x_i) - u^*(x_i) \Big) \right] \\
&\quad + 2L\mathbb{E}_{x'}\mathbb{E}_{\sigma'}\left[ \sup_{u \in \boldsymbol{M}_\rho(\Omega)} \frac{1}{n}\sum_{i=1}^n \sigma_i'\Big( \Delta u(x_i') - \Delta u^*(x_i') \Big) \right] \\
&\lesssim R_n\left( \Big\{ u - u_* : u \in \boldsymbol{M}_\rho \Big\} \right) + R_n\left( \Big\{ \Delta u - \Delta u^* : u \in \boldsymbol{M}_\rho \Big\} \right) \\
&\lesssim R_n\left( \Big\{ u - u^* : u \in \Phi(L, W, S, B), \|u - u^*\|_{L^2(\Omega)} \leq \sqrt{\rho} \Big\} \right) \\
&\quad + R_n\left( \Big\{ \Delta u - \Delta u^* : u \in \Phi(L, W, S, B), \|\Delta u - \Delta u^*\|_{L^2(\Omega)} \leq \sqrt{\rho} \Big\} \right) \\
&\leq R_n\left( \Big\{ u - u^* : u \in \Phi(L, W, S, B), \|u - u^*\|_{n,2} \leq 2\sqrt{\rho} \Big\} \right) \\
&\quad + R_n\left( \Big\{ \Delta u - \Delta u^* : u \in \Phi(L, W, S, B), \|\Delta u - \Delta u^*\|_{n,2} \leq 2\sqrt{\rho} \Big\} \right) \\
&\lesssim \inf_{0 < \alpha < 2\sqrt{\rho}} \Big\{ 4\alpha + \frac{12}{\sqrt{n}} \int_\alpha^{2\sqrt{\rho}} \sqrt{\log \mathcal{N}(\delta, \Phi(L, W, S, B), \|\cdot\|_{n,2})} d\delta \Big\} \\
&\quad + \inf_{0 < \alpha < 2\sqrt{\rho}} \Big\{ 4\alpha + \frac{12}{\sqrt{n}} \int_\alpha^{2\sqrt{\rho}} \sqrt{\log \mathcal{N}(\delta, \Delta\Phi(L, W, S, B), \|\cdot\|_{n,2})} d\delta \Big\} \\
&\lesssim \inf_{0 < \alpha < 2\sqrt{\rho}} \Big\{ 4\alpha + \frac{12}{\sqrt{n}} \int_\alpha^{2\sqrt{\rho}} \sqrt{\log \mathcal{N}(\delta, \Phi(L, W, S, B), \|\cdot\|_\infty)} d\delta \Big\} \\
&\quad + \inf_{0 < \alpha < 2\sqrt{\rho}} \Big\{ 4\alpha + \frac{12}{\sqrt{n}} \int_\alpha^{2\sqrt{\rho}} \sqrt{\log \mathcal{N}(\delta, \Delta\Phi(L, W, S, B), \|\cdot\|_\infty)} d\delta \Big\}.
\end{aligned}
$$

For any $\rho \gtrsim \frac{1}{n^2}$, we pick $\alpha = \frac{1}{n} \lesssim \sqrt{\rho}$ and plug in the upper bounds proved in Theorem B.4 and Theorem B.5, which implies:

$$R_n(\boldsymbol{T}_\rho(\Omega)) \lesssim \frac{1}{n} + \frac{1}{\sqrt{n}} \int_{\frac{1}{n}}^{2\sqrt{\rho}} \sqrt{S\Big[\log(\delta^{-1}) + 3^L\log(WB)\Big]}d\delta + \frac{1}{\sqrt{n}} \int_{\frac{1}{n}}^{2\sqrt{\rho}} \sqrt{S\Big[\log(\delta^{-1}) + 3^L\log(WB)\Big]}d\delta$$

$$\lesssim \sqrt{\frac{S3^L\rho}{n}\log(BWn)}.$$

$\square$

## B.3 Auxiliary definitions and lemmata On Approximation Error

### B.3.1 Approximation using Truncated Fourier Basis

**Lemma B.19.** *Given $\alpha > 0$ and a fixed integer $\xi \in \mathbb{Z}^+$. For any function $f \in H^\alpha(\Omega)$, we let $f_\xi = \sum_{\|z\|_\infty \le \xi} f_z \phi_z$ be the best approximation of $f$ in the space $F_\xi(\Omega)$. Then for any $0 < \beta \le \alpha$, we have the following inequality:*

$$\|f - f_\xi\|^2_{H^\beta(\Omega)} \le \xi^{-2(\alpha-\beta)}\|f\|^2_{H^\alpha}.$$

*Proof.* For $f \in H^\alpha(\Omega)$, we know the Fourier coefficient satisfies

$$\sum_{\|z\|_\infty \ge \xi} |f_z|^2 \|z\|^{2\alpha} \lesssim \|f\|^2_{H^\alpha}.$$

We directly construct $f_\xi = \sum_{\|z\|_\infty \le \xi} f_z \phi_z$ to be the truncated Fourier series of the function $f$, then we have

$$\|f - f_\xi\|^2_{H^\beta(\Omega)} \lesssim \sum_{\|z\|_\infty \ge \xi} |f_z|^2 \|z\|^{2\beta} \le \xi^{-2(\alpha-\beta)} \sum_{\|z\|_\infty \ge \xi} |f_z|^2 \|z\|^{2\alpha} \le \xi^{-2(\alpha-\beta)}\|f\|^2_{H^\alpha}.$$

$\square$

### B.3.2 Approximation using Neural Network

In this section, we aim to provide approximation bound for deep neural network. Our proof of the approximation upper bound is based on the observation that the B-spline approximation(De Boor & De Boor, 1978; Schumaker, 2007) can be formulated as a ReLU3 neural network efficiently(Suzuki, 2018; Gühring et al., 2020; Duan et al., 2021; Jiao et al., 2021a). Although the proof of the approximation of the neural network to the Sobolev spaces is a standard approach, we still demonstrate the proof sketch here.

**Definition B.4.** *(Univariate and Multivariate B-splines) Fix an arbitrary integer $l \in \mathbb{Z}^+$. Consider a corresponding uniform partition $\pi_l$ of $[0,1]$:*

$$\pi_l : 0 = t_0^{(l)} < t_1^{(l)} < \cdots < t_{l-1}^{(l)} < t_l^{(l)} = 1,$$

*where $t_i^{(l)} = \frac{i}{l}$ ($\forall\, 0 \le i \le l$). Now for any $k \in \mathbb{Z}^+$, we can define an extended partition $\pi_{l,k}$ as:*

$$\pi_{l,k} : t_{-k+1}^{(l)} = \cdots t_{-1}^{(l)} = 0 = t_0^{(l)} < t_1^{(l)} < \cdots < t_{l-1}^{(l)} < t_l^{(l)} = 1 = t_{l+1}^{(l)} = \cdots = t_{l+k-1}^{(l)}.$$

*Based on the extended partition $\pi_{l,k}$, the univariate B-splines of order $k$ with respect to partition $\pi_l$ are defined by:*

$$N_{l,i}^{(k)}(x) := (-1)^k (t_{i+k}^{(l)} - t_i^{(l)}) \cdot \Big[t_i^{(l)}, \cdots, t_{i+k}^{(l)}\Big] \max\{(x-t), 0\}^{k-1}, \ x \in [0,1], \ i \in I_{l,k}, \quad (B.49)$$

*where $I_{l,k} = \{-k+1, -k+2, \cdots, l-1\}$ and $\Big[t_i^{(l)}, \cdots, t_{i+k}^{(l)}\Big]$ denotes the divided difference operator.*

*Equivalently, for any $x \in [0,1]$, we can rewrite the univariate B-splines $N_{l,i}^{(k)}(x)$ in an explicit form:*

$$N_{l,i}^{(k)}(x) = \begin{cases} \frac{l^{k-1}}{(k-1)!} \sum_{j=0}^{k} (-1)^j \binom{k}{j} \max\left\{x - \frac{i+j}{l}, 0\right\}^{k-1}, & (0 \le i \le l-k+1) \\ \sum_{j=0}^{k-1} a_{ij} \max\left\{x - \frac{j}{l}, 0\right\}^{k-1} + \sum_{n=1}^{k-2} b_{in}x^n + b_{i0}, & (-k+1 \le i \le 0) \\ \sum_{j=l-k+1}^{l} c_{ij} \max\left\{x - \frac{j}{l}, 0\right\}^{k-1}, & (l-k+1 \le i \le l-1) \end{cases} \quad (B.50)$$

where $\{a_{ij} \mid -k+1 \leq i \leq 0, \, 0 \leq j \leq k-1\}$, $\{b_{in} \mid -k+1 \leq i \leq 0, \, 1 \leq n \leq k-2\}$ and $\{c_{ij} \mid l-k+1 \leq i \leq l-1, \, l-k+1 \leq j \leq l-1\}$ are some fixed constants.

For any index vector $\boldsymbol{i} = (i_1, i_2, \cdots, i_d) \in I_{l,k}^d$, we can define a corresponding multivariate B-spline as a product of univariate B-splines:

$$N_{l,\boldsymbol{i}}^{(k)}(\boldsymbol{x}) := \Pi_{j=1}^d N_{l,i_j}^{(k)}(x_j). \tag{B.51}$$

**Definition B.5.** *(Interpolation Operator(Schumaker, 2007)) Take some domain $\Omega \subset [0,1]^d$ and two arbitrary integers $k, l \in \mathbb{Z}^+$. Consider the extended partition $\pi_{l,k}$ and the corresponding set of multivariate B-splines $\{N_{l,\boldsymbol{i}}^{(k)}(x)\}_{\boldsymbol{i} \in I_{l,k}^d}$ defined in Definition B.4. For any $\boldsymbol{i} \in I_{l,k}^d$, we define the domain $\Omega_{\boldsymbol{i}} := \{\boldsymbol{x} \in \Omega : x_j \in [t_{i_j}, t_{i_j+k}], \, 1 \leq j \leq d\}$. There exists a set of linear functionals $\{\lambda_{\boldsymbol{i}}\}_{\boldsymbol{i} \in I_{k,l}^d}$, where $\lambda_{\boldsymbol{i}} : L^1(\Omega) \to \mathbb{R} \, (\forall \, \boldsymbol{i} \in I_{k,l}^d)$, such that for any $\boldsymbol{i} \in I_{k,l}^d$ and $p \in [1, \infty]$, we have:*

$$\lambda_{\boldsymbol{i}}(N_{l,\boldsymbol{j}}^{(k)}) = \delta_{\boldsymbol{i},\boldsymbol{j}} \text{ and } |\lambda_{\boldsymbol{i}}(f)| \leq 9^{d(k-1)}(2k+1)^d \left(\frac{k}{l}\right)^{-\frac{d}{p}} \|f\|_{L^p(\Omega_{\boldsymbol{i}})}, \, \forall \, f \in L^p(\Omega). \tag{B.52}$$

*The corresponding interpolation operator $Q_{k,l}$ is defined as:*

$$Q_{k,l}f := \sum_{\boldsymbol{i} \in I_{k,l}^d} \lambda_{\boldsymbol{i}}(f) N_{l,\boldsymbol{i}}^{(k)}, \, \forall \, f \in L^1(\Omega).$$

**Theorem B.7.** *[(Schumaker, 2007)] Fix $f \in W^s(\Omega)$ with $\Omega \subseteq [0,1]^d, s \in \mathbb{Z}^+$ and $p \in [1, \infty)$. Then for any $k, l, r \in \mathbb{Z}^+$ with $k \geq s$ and $0 \leq r \leq s$, we have that there exists some constant $C = C(k, s, r, p, d)$, such that:*

$$\|f - Q_{k,l}f\|_{H^r(\Omega)} \leq C \left(\frac{1}{l}\right)^{s-r} \|f\|_{H^s(\Omega)}.$$

**Theorem B.8.** *(Approximation result of Deep Neural Network) Fix some dimension $d \in \mathbb{Z}^+$, some domain $\Omega \subseteq [0,1]^d$. We pick some $l = N^{\frac{1}{d}} \geq 2$, for any $s, r \in \mathbb{Z}^+$ with $0 \leq r \leq s$ and any function $u^* \in H^s(\Omega)$, there exists some sparse Deep Neural Network $u_{DNN} \in \Phi(L, W, S, B)$ with $L = O(1), W = O(N), S = O(N), B = O(N)$, such that:*

$$\|u_{DNN} - u^*\|_{H^r(\Omega)} \lesssim N^{-\frac{s-r}{d}} \|u^*\|_{H^s(\Omega)}. \tag{B.53}$$

*Proof.* We firstly show that the given function $u^*$ can be approximated well by some linear combination of multivariate splines, which is denoted by $u_{\mathrm{sp}}$. Note that $N$ is assumed to be sufficiently large. Hence, we may pick $l = \lceil N^{\frac{1}{d}} \rceil = \Theta(N^{\frac{1}{d}}) \in \mathbb{Z}^+$ to be the partition size of the B-splines. Moreover, by picking $k = 4$ and $p = 2$ in Theorem B.7, we have that the linear combination $u_{\mathrm{sp}} := Q_{4,l}u^* = \sum_{\boldsymbol{i} \in I_{4,l}^d} \lambda_{\boldsymbol{i}}(u^*) N_{l,\boldsymbol{i}}^{(4)}$ satisfies:

$$\|u^* - u_{\mathrm{sp}}\|_{H^r(\Omega)} = \|u^* - Q_{4,l}u^*\|_{H^r(\Omega)} \leq C \left(\frac{1}{l}\right)^{s-r} \|u^*\|_{H^s(\Omega)} = CN^{-\frac{s-r}{d}} \|u^*\|_{H^s(\Omega)}.$$

We will then show that the linear combination $u_{\mathrm{sp}} = \sum_{\boldsymbol{i} \in I_{4,l}^d} \lambda_{\boldsymbol{i}}(f) N_{l,\boldsymbol{i}}^{(4)}$ can be implemented by some Deep Neural Network $u_{\mathrm{DNN}} \in \Phi(L, W, S, B)$ with $L = O(1), W = O(N), S = O(N)$ and $B = O(\log N)$. Firstly, note that for $x \geq 0$, both $x$ and $x^2$ can be expressed in terms of the ReLU3 activation function $\eta_3$ with no error:

$$x = -\frac{1}{12}[\eta_3(x+3) - 5\eta_3(x+2) + 7\eta_3(x+1) - 3\eta_3(x) + 6],$$
$$x^2 = -\frac{1}{6}[\eta_3(x+2) - 4\eta_3(x+1) + 3\eta_3(x) - 4].$$

Applying the explicit formula listed in equation B.50 implies that for any $-3 \leq i \leq l-1$, the univariate B-spline function $N_{l,i}^{(4)}(x)$ $(x \in [0,1])$ can be implemented by some ReLU3 Deep Neural Network $v_{\mathrm{DNN}}$ with both scalar input and scalar output. We have that for $v_{\mathrm{DNN}}$, the depth $L_v$ is 2 and the maximum width $W_v$ is upper bounded by 11.

Secondly, for any $x, y \geq 0$, we have that the product operation $x \cdot y$ can be expressed in terms of the ReLU3 activation function $\eta_3$ with no error:

$$
\begin{aligned}
x \cdot y &= \frac{1}{2}[(x+y)^2 - x^2 - y^2] \\
&= -\frac{1}{12}\Big[\eta_3(x+y+2) - 4\eta_3(x+y+1) + 3\eta_3(x+y) \\
&\quad - \eta_3(x+2) + 4\eta_3(x+1) - 3\eta_3(x) - \eta_3(y+2) + 4\eta_3(y+1) - 3\eta_3(y) + 4\Big].
\end{aligned}
$$

In (Schumaker, 2007), it has been proved that the B-splines are always non-negative, i.e $N_{l,i}^{(4)}(x) \geq 0, \forall x \in [0,1]$. Therefore, by multiplying the non-negative univariate B-splines, we can implement any multivariate B-spline $N_{l,i}^{(4)} = \Pi_{j=1}^{d} N_{l,i_j}^{(4)}(x_j)$ with some ReLU3 Deep Neural Network $p_{\text{DNN}}$. We have that for $p_{\text{DNN}}$, the depth $L_p = \lceil \log_2 d \rceil + 2$ and the maximum width $W_p = \max\{11d, \frac{9}{2}d\}$. Hence, we can further claim that $u^* = \sum_{i \in I_{4,l}^d} \lambda_i(u^*) N_{l,i}^{(4)}$, which is a linear combination of the multivariate B-splines $N_{l,i}^{(4)}$, can be implemented by some ReLU3 Deep Neural Network $u_{\text{DNN}}$. It remains to check that $u_{\text{DNN}} \in \Phi(L, W, S, B)$ with $L = O(1), W = O(N), S = O(N)$ and $B = O(N)$. Note that we can ensure that the hidden layers of $u_{\text{DNN}}$ are of the same dimension $W$ by adding inactive neurons.

For the depth $L$ of $u_{\text{DNN}}$, we have that $L$ is equal to $L_p + 1$, where $L_p$ denotes the depth of the ReLU3 Deep Neural Network $p_{\text{DNN}}$. Thus, we have $L = L_p + 1 = \lceil \log_2 d \rceil + 3$, which implies that $L = O(1)$.

For the width $W$ of $u_{\text{DNN}}$, we have that $W \leq |I_{k,l}^d| W_p$, where $W_p$ denotes the width of the ReLU3 Deep Neural Network $p_{\text{DNN}}$. This implies:

$$
W \leq |I_{k,l}^d| \times 11d = 11d(l+k)^d = 11d(l+4)^d = O(l^d) \Rightarrow W = O(N).
$$

For the sparsity constraint $S$ of $u_{\text{DNN}}$, starting from the third layer, the number of activated neurons is half of the number of activated neurons at previous layer. This yields the following upper bound on $S$:

$$
S \leq 2(W + W + \sum_{j=0}^{L-2} \frac{W}{2^j}) \leq 8W \Rightarrow S = O(W) = O(N).
$$

For the norm constraint $B$ of $u_{\text{DNN}}$, we have the following upper bound on $B$ from equation B.50 and equation B.52:

$$
B = O(\max\{l^{k-1}, \sup_{i \in I_{k,l}^d} \lambda_i(u^*)\}) = O(\max\{l^3, l^d\}) = O(N).
$$

Now we have shown that parameters $L, W, S, B$ of the Deep Neural Network $u_{\text{DNN}}$ are of the desired magnitude, which completes our proof. □

## B.4 Final Upper Bound

In this subsection, we provide the proof of upper bounds for PINN and DRM. For both estimator, we first provide a meta-theorem to illustrate the approximation and generalization decomposition with a $O(1/n)$ fast rate generalization bound(Bartlett et al., 2005; Xu, 2020). Then we use truncated Fourier basis estimator and neural network estimator as example to obtain the final rate.

### B.4.1 Deep Ritz Methods

**Theorem B.9** (Meta-theorem for Upper Bounds of Deep Ritz Methods). *Let $u^* \in H^s(\Omega)$ denote the true solution to the PDE model with Dirichlet boundary condition:*

$$
\begin{aligned}
-\Delta u + V u &= f \text{ on } \Omega, \\
u &= 0 \text{ on } \partial\Omega,
\end{aligned}
\tag{B.54}
$$

*where $f \in L^2(\Omega)$ and $V \in L^\infty(\Omega)$ with $0 < V_{\min} \leq V(x) \leq V_{\max} > 0$. In Theorem B.1, it has been proved that $u^*$ can be obtained by minimizing the loss $\boldsymbol{E}(u)$:*

$$
u^* = \underset{u \in H_0^1(\Omega)}{\arg\min} \, \boldsymbol{E}(u) := \underset{u \in H_0^1(\Omega)}{\arg\min} \Big\{ \frac{1}{2} \int_\Omega \Big[ \|\nabla u\|^2 + V|u|^2 \Big] dx - \int_\Omega f u \, dx \Big\}.
$$

*For a fixed function space $\boldsymbol{F}(\Omega)$, consider the empirical loss induced by the Deep Ritz Method:*

$$\boldsymbol{E}_n(u) = \frac{1}{n}\sum_{j=1}^{n}\Big[|\Omega|\cdot\Big(\frac{1}{2}\|\nabla u(X_j)\|^2 + \frac{1}{2}V(X_j)|u(X_j)|^2 - f(X_j)u(X_j)\Big)\Big], \qquad \text{(B.55)}$$

*where $\{X_j\}_{j=1}^{n}$ are datapoints uniformly sampled from the domain $\Omega$. Then the Deep Ritz estimator associated with function space $\boldsymbol{F}(\Omega)$ is defined as the minimizer of $\boldsymbol{E}_n(u)$ over the function space $\boldsymbol{F}(\Omega)$:*

$$\hat{u}_{DRM} = \min_{u\in\boldsymbol{F}(\Omega)}\boldsymbol{E}_n(u).$$

*Moreover, we assume that there exists some constant $C > 0$ such that all function $u$ in the function space $\boldsymbol{F}(\Omega)$, the real solution $u^*$ and $f, V$ satisfy the following two conditions.*

- *The gradients and function value are uniformly bounded*

$$\max\Big\{\sup_{u\in\boldsymbol{F}(\Omega)}\|u\|_{L^\infty(\Omega)},\ \sup_{u\in\boldsymbol{F}(\Omega)}\|\nabla u\|_{L^\infty(\Omega)}, \|u^*\|_{L^\infty(\Omega)}, \|\nabla u^*\|_{L^\infty(\Omega)}, V_{max}, \|f\|_{L^\infty(\Omega)}\Big\} \le C.$$
$$\text{(B.56)}$$

- *All the functions in the function space $\boldsymbol{F}(\Omega)$ satisfies the boundary condition*

$$u = 0 \text{ on } \partial\Omega.$$

*At the the same time, for any $\rho > 0$, we assume the Rademacher complexity of a localized function space $\boldsymbol{S}_\rho(\Omega) := \Big\{h := |\Omega|\cdot\Big[\frac{1}{2}\Big(\|\nabla u\|^2 - \|\nabla u^*\|^2\Big) + \frac{1}{2}V(|u|^2 - |u^*|^2) - f(u-u^*)\Big] \;\Big|\; \|u - u^*\|_{H^1}^2 \le \rho\Big\}$ can be upper bounded by a sub-root function $\phi = \phi(\rho) : [0,\infty)\to[0,\infty)$, i.e.*

$$\phi(4\rho) \le 2\phi(\rho) \text{ and } R_n(\boldsymbol{S}_\rho(\Omega)) \le \phi(\rho)\ (\forall\ \rho > 0). \qquad \text{(B.57)}$$

*For all constant $t > 0$. We denote $r^*$ to be the solution of the fix point equation of local Rademacher complexity $r = \phi(r)$. There exist two constants $C_p, C_q$ such that with probability $1 - C_p\exp(-C_q t)$, we have the following upper bound for the Deep Ritz Estimator*

$$\|\hat{u}_{DRM} - u^*\|_{H^1}^2 \lesssim \inf_{u_{\boldsymbol{F}}\in\boldsymbol{F}(\Omega)}\Big(\boldsymbol{E}(u_{\boldsymbol{F}}) - \boldsymbol{E}(u^\star)\Big) + \max\Big\{r^*, \frac{t}{n}\Big\}.$$

*Proof.* To upper bound the excess risk $\Delta\boldsymbol{E}^{(n)} := \boldsymbol{E}(\hat{u}_{DRM}) - \boldsymbol{E}(u^*)$, following(Xu, 2020; Lu et al., 2021b; Duan et al., 2021), we decompose the excess risk into approximation error and generalization error with probability $1 - e^{-C_q t}$, where $C_q > 0$ is some constant:

$$\begin{aligned}
\Delta\boldsymbol{E}^{(n)}(\hat{u}_{DRM}) = \boldsymbol{E}(\hat{u}_{DRM}) - \boldsymbol{E}(u^\star) = {} & \big[\boldsymbol{E}(\hat{u}_{DRM}) - \boldsymbol{E}_n(\hat{u}_{DRM})\big] + \big[\boldsymbol{E}_n(\hat{u}_{DRM}) - \boldsymbol{E}_n(u_{\boldsymbol{F}})\big] \\
& + \big[\boldsymbol{E}_n(u_{\boldsymbol{F}}) - \boldsymbol{E}(u_{\boldsymbol{F}})\big] + \big[\boldsymbol{E}(u_{\boldsymbol{F}}) - \boldsymbol{E}(u^\star)\big] \\
\le {} & \big[\boldsymbol{E}(\hat{u}_{DRM}) - \boldsymbol{E}_n(\hat{u}_{DRM})\big] + \big[\boldsymbol{E}_n(u_{\boldsymbol{F}}) - \boldsymbol{E}(u_{\boldsymbol{F}})\big] + \big[\boldsymbol{E}(u_{\boldsymbol{F}}) - \boldsymbol{E}(u^\star)\big] \\
\le {} & \big[\boldsymbol{E}(\hat{u}_{DRM}) - \boldsymbol{E}(u^*) + \boldsymbol{E}_n(u^*) - \boldsymbol{E}_n(\hat{u}_{DRM})\big] \\
& + \frac{3}{2}\big[\boldsymbol{E}(u_{\boldsymbol{F}}) - \boldsymbol{E}(u^\star)\big] + \frac{t}{2n},
\end{aligned}$$
$$\text{(B.58)}$$

where the expectation is on all sampled data. The inequality of the third line is because $\hat{u}_{DRM}$ is the minimizer of the empirical loss $\boldsymbol{E}_n$ in the solution set $\boldsymbol{F}(\Omega)$, so we have $\boldsymbol{E}_n(\hat{u}_{DRM}) \le \boldsymbol{E}_n(u_{\boldsymbol{F}})$. The last inequality is based on the Bernstein inequality. The variance of $h = |\Omega|\cdot\Big[\frac{1}{2}\Big(\|\nabla u\|^2 - \|\nabla u^*\|^2\Big) + \frac{1}{2}V(|u|^2 - |u^*|^2) - f(u-u^*)\Big]$ can be upper bounded by $\frac{\alpha'}{\alpha}\big[\boldsymbol{E}(u_{\boldsymbol{F}}) - \boldsymbol{E}(u^\star)\big]$ due to the strong convexity of the variation objective (B.60). According to the Bernstein inequality, there exists some constant $C_q > 0$, such that with probability $1 - e^{-C_q t}$ we have:

$$\boldsymbol{E}_n(u_{\boldsymbol{F}}) - \boldsymbol{E}_n(u^*) - \boldsymbol{E}(u_{\boldsymbol{F}}) + \boldsymbol{E}(u^*) \le \sqrt{\frac{t\big[\boldsymbol{E}(u_{\boldsymbol{F}}) - \boldsymbol{E}(u^\star)\big]}{n}} \le \frac{1}{2}\big[\boldsymbol{E}(u_{\boldsymbol{F}}) - \boldsymbol{E}(u^\star)\big] + \frac{t}{2n}.$$

Note that B.58 holds for all function lies in the function space $\boldsymbol{F}$. Thus, we can take $u_{\boldsymbol{F}} := \arg\min_{u_0 \in \boldsymbol{F}(\Omega)} \left( \boldsymbol{E}(u_0) - \boldsymbol{E}(u^\star) \right)$ and finally get

$$\Delta \boldsymbol{E}^{(n)} \leq \underbrace{\boldsymbol{E}(\hat{u}_{\mathrm{DRM}}) - \boldsymbol{E}(u^*) + \boldsymbol{E}_n(u^*) - \boldsymbol{E}_n(u)}_{\Delta \boldsymbol{E}_{\mathrm{gen}}} + \frac{3}{2} \underbrace{\inf_{u_{\boldsymbol{F}} \in \boldsymbol{F}(\Omega)} \left( \boldsymbol{E}(u_{\boldsymbol{F}}) - \boldsymbol{E}(u^\star) \right)}_{\Delta \boldsymbol{E}_{\mathrm{app}}} + \frac{t}{2n}.$$

This inequality decompose the excess risk to the generalization error $\Delta \boldsymbol{E}_{\mathrm{gen}} := \boldsymbol{E}(\hat{u}_{\mathrm{DRM}}) - \boldsymbol{E}(u^*) + \boldsymbol{E}_n(u^*) - \boldsymbol{E}_n(\hat{u}_{\mathrm{DRM}})$ and the approximation error $\Delta \boldsymbol{E}_{\mathrm{app}} = \inf_{u_{\boldsymbol{F}} \in \boldsymbol{F}(\Omega)} \left( \boldsymbol{E}(u_{\boldsymbol{F}}) - \boldsymbol{E}(u^\star) \right)$.

From the lemmata proved in Section B.3, we already have an estimation of the approximation error's convergence rate. So now we'll focus on providing fast rate upper bounds of the generalization error for the two estimators using the localization technique(Bartlett et al., 2005; Xu, 2020). To achieve the fast generalization bound, we focus on the following normalized empirical process

$$\tilde{\boldsymbol{S}}_r(\Omega) := \left\{ \tilde{h}(x) := \frac{\mathbb{E}[h] - h(x)}{\mathbb{E}[h] + r} \mid h \in \boldsymbol{S}(\Omega) \right\} (r > 0).$$

First, we try to bound the expectation of the normalized empirical process. Applying the Symmetrization Lemma B.1, we can first bound the expectation as

$$\sup_{\tilde{h} \in \tilde{\boldsymbol{S}}_r(\Omega)} \mathbb{E}_{x'} \left[ \frac{1}{n} \sum_{i=1}^n \tilde{h}(x_i') \right] \leq \mathbb{E}_{x'} \left[ \sup_{h \in \boldsymbol{S}(\Omega)} \left| \frac{1}{n} \sum_{i=1}^n \frac{h(x_i') - \mathbb{E}[h]}{\mathbb{E}[h] + r} \right| \right] \leq 2 R_n(\hat{\boldsymbol{S}}_r(\Omega)).$$

where the function class $\hat{\boldsymbol{S}}_r(\Omega)$ is defined as:

$$\hat{\boldsymbol{S}}_r(\Omega) := \left\{ \hat{h}(x) := \frac{h(x)}{\mathbb{E}[h] + r} \mid h \in \boldsymbol{S}(\Omega) \right\},$$

where $\boldsymbol{S}(\Omega) = \left\{ h := |\Omega| \cdot \left[ \frac{1}{2} \left( \|\nabla u\|^2 - \|\nabla u^*\|^2 \right) + \frac{1}{2} V(|u|^2 - |u^*|^2) - f(u - u^*) \right] \right\}$. Then applying the Peeling Lemma B.4 to any function $h \in \boldsymbol{S}(\Omega)$ helps us upper bound the local Rademacher complexity $R_n(\hat{\boldsymbol{S}}_r(\Omega))$ with the function $\phi$ defined in equation B.57:

$$R_n(\hat{\boldsymbol{S}}_r(\Omega)) = \mathbb{E}_\sigma \left[ \mathbb{E}_x \left[ \sup_{h \in \boldsymbol{S}(\Omega)} \frac{\frac{1}{n} \sum_{i=1}^n \sigma_i h(x_i)}{\mathbb{E}[h] + r} \right] \right] \leq \frac{4\phi(r)}{r}.$$

Combining all inequalities derived above yields:

$$\sup_{\tilde{h} \in \tilde{\boldsymbol{S}}_r(\Omega)} \mathbb{E}_{x'} \left[ \frac{1}{n} \sum_{i=1}^n \tilde{h}(x_i') \right] \leq 2 R_n(\hat{\boldsymbol{S}}_r(\Omega)) \leq \frac{8\phi(r)}{r} (r > 0). \tag{B.59}$$

Secondly we'll apply the Talagrand concentration inequality, which requires us to verify the condition needed. We will first check that the expectation value $\mathbb{E}[h]$ is always non-negative for any $h \in \boldsymbol{S}(\Omega)$:

$$\mathbb{E}[h] = \frac{1}{|\Omega|} \int_\Omega |\Omega| \cdot \left( \frac{1}{2} \|\nabla u(x)\|^2 + \frac{1}{2} V(x)|u(x)|^2 - f(x)u(x) \right) dx$$

$$- \frac{1}{|\Omega|} \int_\Omega |\Omega| \cdot \left( \frac{1}{2} \|\nabla u^\star(x)\|^2 + \frac{1}{2} V(x)|u^\star(x)|^2 - f(x)u^\star(x) \right) dx$$

$$= \boldsymbol{E}(u) - \boldsymbol{E}(u^\star) \geq 0 \Rightarrow \mathbb{E}[h] \geq 0.$$

We will proceed to verify that any $\tilde{h} = \frac{\mathbb{E}[h] - h}{\mathbb{E}[h] + r} \in \tilde{\boldsymbol{S}}_r(\Omega)$ is of bounded inf-norm. We need to prove that any $h \in \boldsymbol{S}(\Omega)$ is of bounded inf-norm beforehand. Using boundedness condition listed in

equation B.56 implies:

$$
\begin{aligned}
\|h\|_\infty &= |\Omega| \| \frac{1}{2}\Big(\|\nabla u\|^2 - \|\nabla u^*\|^2\Big) + \frac{1}{2}V(|u|^2 - |u^*|^2) - f(u - u^*)\|_\infty \\
&\leq \frac{|\Omega|}{2}\Big(\|\nabla u\|_\infty^2 + \|\nabla u^*\|_\infty^2\Big) + \frac{|\Omega|}{2}V_{\max}\Big(\|u\|_\infty^2 + \|u^*\|_\infty^2\Big) + |\Omega|\|f\|_\infty\Big(\|u\|_\infty + \|u^*\|_\infty\Big) \\
&\leq \frac{|\Omega|}{2} \times 2C^2 + \frac{|\Omega|}{2}V_{\max} \times 2C^2 + 2|\Omega|C^2 = |\Omega|(V_{\max} + 3)C^2.
\end{aligned}
$$

By taking $M := |\Omega|(V_{\max} + 3)C^2$, we then have $\|h\|_\infty \leq M$ for all $h \in \boldsymbol{S}(\Omega)$. Note that the denominator can be lower bounded by $|\mathbb{E}[h] + r| \geq r > 0$. Combining these two inequalities help us upper bound the inf-norm $\|\tilde{h}\|_\infty = \sup_{x \in \Omega}|\tilde{h}(x)|$ as follows:

$$
\|\tilde{h}\|_\infty = \frac{\|\mathbb{E}[h] - h\|_\infty}{|\mathbb{E}[h] + r|} \leq \frac{2\|h\|_\infty}{r} \leq \frac{2M}{r} =: \beta.
$$

We will then check the normalized functions $\frac{\mathbb{E}[h] - h(x)}{\mathbb{E}[h] + r}$ in $\tilde{S}_r(\Omega)$ have bounded second moment, which is satisfied because of the regularity results of the PDE. We aim to show that there exist some constants $\alpha, \alpha' > 0$, such that for any $h \in \boldsymbol{S}(\Omega)$, the following inequality holds:

$$
\alpha\mathbb{E}[h^2] \leq \|u - u^*\|_{H^1(\Omega)}^2 \leq \alpha'\mathbb{E}[h]. \tag{B.60}
$$

The RHS of the inequality follows from strong convexity of the DRM objective function proved in Theorem B.1:

$$
\mathbb{E}[h] = \boldsymbol{E}(u) - \boldsymbol{E}(u^*) \geq \frac{\min\{1, V_{\min}\}}{4}\|u - u^*\|_{H^1(\Omega)}^2.
$$

The LHS of the inequality follows from boundedness condition listed in equation B.56 and the QM-AM inequality:

$$
\begin{aligned}
\mathbb{E}[h^2] &= \int_\Omega \left[\frac{1}{2}\Big(\|\nabla u\|^2 - \|\nabla u^*\|^2\Big) + \frac{1}{2}V(|u|^2 - |u^*|^2) - f(u - u^*)\right]^2 dx \\
&\leq \frac{3}{4}\int_\Omega \Big(\|\nabla u\|^2 - \|\nabla u^*\|^2\Big)^2 dx + \frac{3}{4}\int_\Omega V^2(|u|^2 - |u^*|^2)^2 dx + 3\int_\Omega f^2(u - u^*)^2 dx \\
&\leq \frac{3}{4}\int_\Omega \Big|\|\nabla u\| - \|\nabla u^*\|\Big|^2 (\|\nabla u\| + \|\nabla u^*\|)^2 dx + \frac{3}{4}V_{\max}^2\int_\Omega \Big||u| - |u^*|\Big|^2 (|u| + |u^*|)^2 dx \\
&\quad + 3C^2\int_\Omega (u - u^*)^2 dx \leq 3C^2\int_\Omega \|\nabla u - \nabla u^*\|^2 dx + 3C^2(1 + V_{\max}^2)\int_\Omega |u - u^*|^2 dx \\
&\leq 3C^2(1 + V_{\max}^2)\|u - u^*\|_{H^1(\Omega)}^2.
\end{aligned}
$$

By picking $\alpha' = \frac{4}{\min\{1, V_{\min}\}}$ and $\alpha = \frac{1}{3C^2(1 + V_{\max}^2)}$, we have finished proving inequality B.60. Then we can can upper bound the expectation $\mathbb{E}[\tilde{h}^2]$ as:

$$
\mathbb{E}[\tilde{h}^2] = \frac{\mathbb{E}[(h - \mathbb{E}[h])^2]}{|\mathbb{E}[h] + r|^2} = \frac{\mathbb{E}[h^2] - \mathbb{E}[h]^2}{|\mathbb{E}[h] + r|^2} \leq \frac{\mathbb{E}[h^2]}{|\mathbb{E}[h] + r|^2}.
$$

Using the fact that $\mathbb{E}[h] \geq 0$ and inequality B.60, we can lower bound the denominator $|\mathbb{E}[h] + r|^2$ as follows:

$$
|\mathbb{E}[h] + r|^2 \geq 2\mathbb{E}[h]r \geq \frac{2r\alpha}{\alpha'}\mathbb{E}[h^2].
$$

Therefore, we can deduce that:

$$
\mathbb{E}[\tilde{h}^2] \leq \frac{\mathbb{E}[h^2]}{|\mathbb{E}[h] + r|^2} \leq \frac{\mathbb{E}[h^2]}{\frac{2r\alpha}{\alpha'}\mathbb{E}[h^2]} = \frac{\alpha'}{2r\alpha} =: \sigma^2.
$$

Hence, any function in the localized class $\tilde{\boldsymbol{S}}_r(\Omega)$ is of bounded second moment.

It is easy to check that for any $\tilde{h} \in \tilde{\boldsymbol{S}}_r(\Omega)$, we have

$$\mathbb{E}[\tilde{h}] = \frac{\mathbb{E}[h] - \mathbb{E}[h]}{\mathbb{E}[h] + r} = 0,$$

*i.e.* any function in the localized class $\tilde{\boldsymbol{S}}_r(\Omega)$ is of zero mean.

Now we have verified that any function $\tilde{h} \in \tilde{\boldsymbol{S}}_r(\Omega)$ satisfies all the required conditions. By taking $\mu$ to be the uniform distribution on the domain $\Omega$ and applying Talagrand's Concentration inequality given in Lemma B.3, we have:

$$\mathbb{P}_x \left[ \sup_{\tilde{h} \in \tilde{\boldsymbol{S}}_r(\Omega)} \frac{1}{n} \sum_{i=1}^n \tilde{h}(x_i) \geq 2 \sup_{\tilde{h} \in \tilde{\boldsymbol{S}}_r(\Omega)} \mathbb{E}_{x'} \left[ \frac{1}{n} \sum_{i=1}^n \tilde{h}(x_i') \right] + \sqrt{\frac{2t\sigma^2}{n}} + \frac{2t\beta}{n} \right] \leq e^{-t}.$$

By using the upper bound deduced above and plugging in the expressions of $\beta$ and $\sigma$, we can rewrite Talagrand's Concentration Inequality in the following way. With probability at least $1 - e^{-t}$, the inequality below holds:

$$\frac{1}{n} \sum_{i=1}^n \tilde{h}(x_i) \leq \sup_{\tilde{h} \in \tilde{\boldsymbol{S}}_r(\Omega)} \frac{1}{n} \sum_{i=1}^n \tilde{h}(x_i) \leq 2 \sup_{\tilde{h} \in \tilde{\boldsymbol{S}}_r(\Omega)} \mathbb{E}_{x'} \left[ \frac{1}{n} \sum_{i=1}^n \tilde{h}(x_i') \right] + \sqrt{\frac{2t\sigma^2}{n}} + \frac{2t\beta}{n}$$

$$\leq \frac{16\phi(r)}{r} + \sqrt{\frac{t\alpha'}{n\alpha r}} + \frac{4Mt}{nr} =: \psi(r).$$

Let's pick the critical radius $r_0$ to be:

$$r_0 = \max\{2^{14} r^*, \frac{24Mt}{n}, \frac{36\alpha't}{\alpha n}\}. \tag{B.61}$$

Note that concavity of the function $\phi$ implies that $\phi(r) \leq r$ for any $r \geq r^*$. Combining this with the first inequality listed in B.57 yields:

$$\frac{16\phi(r)}{r} \leq \frac{2^{11}\phi(\frac{r_0}{2^{14}})}{2^{14}\frac{r_0}{2^{14}}} = \frac{1}{8} \times \frac{\phi(\frac{r_0}{2^{14}})}{\frac{r)}{2^{14}}} \leq \frac{1}{8}.$$

On the other hand, applying equation B.61 yields:

$$\sqrt{\frac{\alpha't}{n\alpha r_0}} \leq \sqrt{\frac{\alpha't}{n\alpha} \frac{\alpha n}{36\alpha't}} = \frac{1}{6},$$
$$\frac{4Mt}{nr_0} \leq \frac{4Mt}{n} \times \frac{n}{24Mt} = \frac{1}{6}.$$

Summing the three inequalities above implies:

$$\psi(r_0) = \frac{16\phi(r_0)}{r_0} + \sqrt{\frac{t\alpha'}{n\alpha r_0}} + \frac{4Mt}{nr_0} \leq \frac{1}{8} + \frac{1}{6} + \frac{1}{6} < \frac{1}{2}.$$

By picking $r = r_0$, we can further deduce that for any function $u \in \boldsymbol{F}(\Omega)$, the following inequality holds with probability $1 - e^{-t}$:

$$\frac{\boldsymbol{E}(u) - \boldsymbol{E}(u^*) - \boldsymbol{E}_n(u) + \boldsymbol{E}_n(u^*)}{\boldsymbol{E}(u) - \boldsymbol{E}(u^*) + r_0} = \frac{1}{n} \sum_{i=1}^n \tilde{h}(x_i) \leq \psi(r_0) < \frac{1}{2}.$$

Multiplying the denominator on both sides indicates:

$$\Delta\boldsymbol{E}_{\text{gen}} = \boldsymbol{E}(u) - \boldsymbol{E}(u^*) - \boldsymbol{E}_n(u) + \boldsymbol{E}_n(u^*) \leq \frac{1}{2}\left[\boldsymbol{E}(u) - \boldsymbol{E}(u^*)\right] + \frac{1}{2}r_0 = \frac{1}{2}\Delta\boldsymbol{E}^{(n)} + \frac{1}{2}r_0.$$

Substituting the upper bound above into the decomposition $\Delta\boldsymbol{E}^{(n)} \leq \Delta\boldsymbol{E}_{\text{gen}} + \frac{3}{2}\Delta\boldsymbol{E}_{\text{app}} + \frac{t}{2n}$ yields that with probability $1 - 2e^{-\min\{C_q, 1\}t}$, we have:

$$\Delta\boldsymbol{E}^{(n)} \leq \Delta\boldsymbol{E}_{\text{gen}} + \frac{3}{2}\Delta\boldsymbol{E}_{\text{app}} + \frac{t}{2n} \leq \frac{1}{2}\Delta\boldsymbol{E}^{(n)} + \frac{1}{2}r_0 + \frac{3}{2}\Delta\boldsymbol{E}_{\text{app}} + \frac{t}{2n}.$$

Simplifying the inequality above yields that with probability $1 - 2e^{-\min\{C_q, 1\}t}$, we have:

$$\Delta \boldsymbol{E}^{(n)} \leq r_0 + 3\Delta \boldsymbol{E}_{\text{app}} + \frac{t}{n} = 3 \inf_{u_{\boldsymbol{F}} \in \boldsymbol{F}(\Omega)} \left( \boldsymbol{E}(u_{\boldsymbol{F}}) - \boldsymbol{E}(u^\star) \right) + \max\{2^{14} r^*, 24M \frac{t}{n}, \frac{36\alpha'}{\alpha} \frac{t}{n}\} + \frac{t}{n}$$

$$\lesssim \inf_{u_{\boldsymbol{F}} \in \boldsymbol{F}(\Omega)} \left( \boldsymbol{E}(u_{\boldsymbol{F}}) - \boldsymbol{E}(u^\star) \right) + \max\left\{ r^*, \frac{t}{n} \right\}.$$

Moreover, using strong convexity of the DRM objective function proved in Theorem B.1 implies:

$$\Delta \boldsymbol{E}^{(n)} = \boldsymbol{E}(\hat{u}_{\text{DRM}}) - \boldsymbol{E}(u^*) \geq \frac{\min\{1, V_{\min}\}}{4} \|\hat{u}_{\text{DRM}} - u^*\|^2_{H^1(\Omega)}.$$

Combining the two bounds above yields that with probability $1 - 2e^{-\min\{C_q, 1\}t}$, we have:

$$\|\hat{u}_{\text{DRM}} - u^*\|^2_{H^1(\Omega)} \lesssim \inf_{u_{\boldsymbol{F}} \in \boldsymbol{F}(\Omega)} \left( \boldsymbol{E}(u_{\boldsymbol{F}}) - \boldsymbol{E}(u^\star) \right) + \max\left\{ r^*, \frac{t}{n} \right\}.$$

$\square$

**Deep Neural Network Estimator.** For any $N \in \mathbb{Z}^+$, there exists some Deep Neural Network in $\Phi(L, W, S, B)$ with $L = O(1)$, $W = O(N)$, $S = O(N)$, $B = O(N)$, such that the approximation error $\Delta \boldsymbol{E}_{\text{app}} = O(N^{-\frac{2(s-1)}{d}})$ and generalization error $\Delta \boldsymbol{E}_{\text{gen}} = O(\frac{N \log N}{n})$. With optimal selection $N = n^{\frac{d}{d+2s-2}}$ to balance the bias and variance, we can achieve $n^{-\frac{2s-2}{d+2s-2}} \log n$ convergence rate for the DNN estimator.

**Theorem B.10.** *(Final Upper Bound of DRM with Deep Neural Network Estimator) Under the assumptions in Theorem B.9, we consider the Deep Ritz objective with the sparse Deep Neural Network function space $\Phi(L, W, S, B)$, where the parameters $L = O(1)$, $W = O(n^{\frac{d}{d+2s-2}})$, $S = O(n^{\frac{d}{d+2s-2}})$, $B = O(n^{\frac{d}{d+2s-2}})$. Then we have that the DNN estimator $\hat{u}_{DRM}^{DNN} = \min_{u \in \Phi(L, W, S, B)} \boldsymbol{E}_n^{DRM}(u)$ satisfies the following upper bound with high probability:*

$$\|\hat{u}_{DRM}^{DNN} - u^*\|^2_{H^1} \lesssim n^{-\frac{2s-2}{d+2s-2}} \log n.$$

*Proof.* On the one hand, by taking $s = 1$ and $p = 2$ in Theorem B.8 proved above, we have that there exists some Deep Neural Network $u_{\text{DNN}} \in \Phi(L, W, S, B)$ with $L = O(1), W = O(N), S = O(N), B = O(N)$, such that.

$$\|u_{\text{DNN}} - u^*\|^2_{H^1(\Omega)} \leq N^{-\frac{2s-2}{d}} \|u^*\|_{H^s(\Omega)}.$$

Applying strong convexity of the DRM objective function proved in Theorem B.1 yields the following upper bound on the approximation error of DRM:

$$\Delta \boldsymbol{E}_{\text{app}} \lesssim \|u_{\text{DNN}} - u^*\|^2_{H^1(\Omega)} \leq N^{-\frac{2s-2}{d}}.$$

On the other hand, from Lemma B.17 proved above, we know that the function $\phi(\rho)$ that upper bounds the local Rademacher complexity of the Deep Neural Network space is of the same magnitude as $\sqrt{\frac{S3^L \rho}{n} \log(BWn)}$. By plugging in the magnitudes of $L, W, S, B$, we can determine the critical radius $r^*$:

$$\sqrt{\frac{r^* 3^L S}{n} \log(BWn)} \simeq \sqrt{\frac{r^* N}{n} (2\log N + \log n)} \simeq r^* \Rightarrow r^* \simeq \frac{N(\log N + \log n)}{n}.$$

Combining the two bounds above with Theorem B.9 yields that with high probability, we have:

$$\|\hat{u}_{\text{DRM}}^{\text{DNN}} - u^*\|^2_{H^1} \lesssim \Delta \boldsymbol{E}_{\text{app}} + r^* \lesssim N^{-\frac{2(s-1)}{d}} + \frac{N(\log N + \log n)}{n}.$$

By equating the two terms above, we can solve for the optimal $N$ that yields the desired bound:

$$N^{-\frac{2(s-1)}{d}} \simeq \frac{N}{n} \Rightarrow N \simeq n^{\frac{d}{d+2s-2}}.$$

Plugging in the optimal $N$ gives us the magnitudes of the four parameters $L = O(1)$, $W = O(n^{\frac{d}{d+2s-2}})$, $S = O(n^{\frac{d}{d+2s-2}})$, $B = O(n^{\frac{d}{d+2s-2}})$, as well as the final rate:

$$\|\hat{u}_{\text{DRM}}^{\text{DNN}} - u^*\|^2_{H^1} \lesssim N^{-\frac{2(s-1)}{d}} + \frac{N \log N}{n} \lesssim n^{-\frac{2(s-1)}{d+2(s-1)}} \log n.$$

$\square$

**Truncated Fourier Series Estimator.** For any $\xi \in \mathbb{Z}^+$, there exists some Truncated Fourier Series in $\boldsymbol{F}_\xi(\Omega)$ with approximation error $\Delta \boldsymbol{E}_{\text{app}} = O(\xi^{-2(s-1)})$ and generalization error $\Delta \boldsymbol{E}_{\text{gen}} = O(\frac{\xi^d}{n})$. With optimal selection $\xi = n^{\frac{1}{d+2s-2}}$ to balance the bias and variance, we can achieve $n^{-\frac{2s-2}{d+2s-2}}$ convergence rate for the Fourier estimator.

**Theorem B.11.** *(Final Upper Bound of DRM with Truncated Fourier Series Estimator) Under the assumptions in Theorem B.9, we consider the Deep Ritz objective with the Truncated Fourier Series function space $\boldsymbol{F}_\xi(\Omega)$, where the parameter $\xi = \Theta(n^{\frac{1}{d+2s-2}})$. Then we have that the Fourier estimator $\hat{u}_{DRM}^{Fourier} = \min_{u \in \boldsymbol{F}_\xi(\Omega)} \boldsymbol{E}_n^{DRM}(u)$ satisfies the following upper bound with high probability:*

$$\|\hat{u}_{DRM}^{Fourier} - u^*\|_{H^1}^2 \lesssim n^{-\frac{2s-2}{d+2s-2}}.$$

*Proof.* Let's firstly derive the function $\phi(\rho)$ that upper bounds the local Rademacher complexity of $\boldsymbol{S}_\rho(\Omega) := \left\{ h := |\Omega| \cdot \left[ \frac{1}{2}\left(\|\nabla u\|^2 - \|\nabla u^*\|^2\right) + \frac{1}{2}V(|u|^2 - |u^*|^2) - f(u - u^*) \right] \ \middle| \ u \in \boldsymbol{F}_{\rho,\xi}(\Omega) \right\}$, where $\boldsymbol{F}_{\rho,\xi}(\Omega) = \left\{ v \in F_\xi(\Omega) \ \middle| \ \|v\|_{H^1(\Omega)}^2 \leq \rho \right\}$ denotes the localized Truncated Fourier Series space. From Talagrand Contraction Lemma B.2, Lemma B.6 and Lemma B.7 proved above, we have

$$R_n(\boldsymbol{S}_\rho(\Omega)) \lesssim R_n\left(\left\{u - u^* : u \in \boldsymbol{F}_{\rho,\xi}(\Omega), \|u - u^*\|_{H^1(\Omega)} \leq \sqrt{\rho}\right\}\right)$$

$$+ R_n\left(\left\{\|\nabla u - \nabla u^*\| : u \in \boldsymbol{F}_{\rho,\xi}(\Omega), \|u - u^*\|_{H^1(\Omega)} \leq \sqrt{\rho}\right\}\right)$$

$$\lesssim \mathbb{E}_X\left[\mathbb{E}_\sigma\left[\sup_{u \in \boldsymbol{F}_{\rho,\xi}(\Omega)} \frac{1}{n}\sum_{i=1}^n \sigma_i(u(X_i) - \Pi_\xi u^*(X_i)) \middle| \|u - \Pi_\xi u^*\|_{H^1(\Omega)}^2 \leq \rho\right]\right]$$

$$+ \mathbb{E}_X\left[\mathbb{E}_\sigma\left[\sup_{u \in \boldsymbol{F}_{\rho,\xi}(\Omega)} \frac{1}{n}\sum_{i=1}^n \sigma_i\|\nabla u(X_i) - \nabla\Pi_\xi u^*(X_i)\| \middle| \|u - \Pi_\xi u^*\|_{H^1(\Omega)}^2 \leq \rho\right]\right]$$

$$+ \mathbb{E}_X\left[\mathbb{E}_\sigma\left[\frac{1}{n}\sum_{i=1}^n \sigma_i\|\nabla\Pi_{>\xi} u^*(X_i)\|\right]\right] + \mathbb{E}_X\left[\mathbb{E}_\sigma\left[\frac{1}{n}\sum_{i=1}^n \sigma_i\Pi_{>\xi} u^*(X_i)\right]\right]$$

$$\lesssim \mathbb{E}_X\left[\mathbb{E}_\sigma\left[\sup_{v \in \boldsymbol{F}_{\rho,\xi}(\Omega)} \frac{1}{n}\sum_{i=1}^n \sigma_i v(X_i) \middle| \|v\|_{H^1(\Omega)}^2 \leq \rho\right]\right]$$

$$+ \mathbb{E}_X\left[\mathbb{E}_\sigma\left[\sup_{v \in \boldsymbol{F}_{\rho,\xi}(\Omega)} \frac{1}{n}\sum_{i=1}^n \sigma_i\|\nabla v(X_i)\| \middle| \|v\|_{H^1(\Omega)}^2 \leq \rho\right]\right]$$

$$+ \mathbb{E}_X\left[\mathbb{E}_\sigma\left[\frac{1}{n}\sum_{i=1}^n \sigma_i\|\nabla\Pi_{>\xi} u^*(X_i)\|\right]\right] + \mathbb{E}_X\left[\mathbb{E}_\sigma\left[\frac{1}{n}\sum_{i=1}^n \sigma_i\Pi_{>\xi} u^*(X_i)\right]\right]$$

$$\lesssim \sqrt{\frac{\rho}{n}}\xi^{\frac{d}{2}} + \sqrt{\frac{\|\Pi_{>\xi} u^*\|_{H^1}^2}{n}} \lesssim \sqrt{\frac{\rho}{n}}\xi^{\frac{d}{2}} + \sqrt{\frac{\xi^{-2(s-1)}}{n}} \lesssim \sqrt{\frac{\rho}{n}}\xi^{\frac{d}{2}} + \frac{1}{n} + \xi^{-2(s-1)},$$

$$\text{(B.62)}$$

where $\Pi_\xi u := \sum_{\|z\|_\infty \leq \xi} u_z \phi_z(x)$ is the projection to the Fourier basis whose frequency is smaller than $\xi$ and $\Pi_{>\xi} u := \sum_{\|z\|_\infty \leq \xi} u_z \phi_z(x)$ is the projection to the Fourier basis whose frequency is larger than $\xi$. Then, the critical radius $r^*$ can be determined as follows:

$$\sqrt{\frac{r^*}{n}}\xi^{\frac{d}{2}} + \frac{1}{n} + \xi^{-2(s-1)} \simeq r^* \Rightarrow r^* \simeq \frac{\xi^d}{n} + \frac{1}{n} + \xi^{-2(s-1)}.$$

Moreover, by taking $\alpha = s$ and $\beta = 1$ in Lemma B.19 and applying strong convexity of the DRM objective function proved in Theorem B.1, we can upper bound the approximation error $\Delta \boldsymbol{E}_{\text{app}}$ as below:

$$\Delta \boldsymbol{E}_{\text{app}} \lesssim \xi^{-2(s-1)}.$$

Combining the two bounds above with Theorem B.9 yields that with high probability, we have:

$$\|\hat{u}_{DRM}^{Fourier} - u^*\|_{H^1}^2 \lesssim \Delta \boldsymbol{E}_{\text{app}} + r^* \lesssim \frac{\xi^d}{n} + \frac{1}{n} + \xi^{-2(s-1)}.$$

By equating two of the three terms above, we can solve for $\xi$ that yields the desired bound:

$$\frac{\xi^d}{n} \simeq \xi^{-2(s-1)} \Rightarrow \xi \simeq n^{\frac{1}{d+2s-2}}.$$

Plugging in the optimal $\xi$ gives us the final rate:

$$\|\hat{u}_{\text{DRM}}^{\text{Fourier}} - u^*\|_{H^1}^2 \lesssim \frac{\xi^d}{n} + \xi^{-2(s-1)} + \frac{1}{n} \lesssim n^{-\frac{2s-2}{d+2s-2}}.$$

$\square$

### B.4.2 PHYSICS INFORMED NEURAL NETWORK

**Theorem B.12** (Meta-theorem for Upper Bounds of Physics Informed Neural Network). *Let $u^* \in H^s(\Omega)$ denote the true solution to the PDE model with Dirichlet boundary condition:*

$$-\Delta u + Vu = f \text{ on } \Omega, \qquad\qquad\qquad\qquad \text{(B.63)}$$
$$u = 0 \text{ on } \partial\Omega,$$

*where $f \in L^2(\Omega)$ and $V \in L^\infty(\Omega)$ with $V - \frac{1}{2}\Delta V > C_{\min}, 0 < C_{\min} < V(x) \le V_{\max}$ and $-\Delta V(x) \le V_{\max}$. In Theorem B.2, it has been proved that $u^*$ can be obtained by minimizing the loss $\boldsymbol{E}(u)$:*

$$u^* = \underset{u \in H_0^1(\Omega)}{\arg\min} \boldsymbol{E}(u) := \underset{u \in H_0^1(\Omega)}{\arg\min} \left\{ \int_\Omega |\Delta u - Vu + f|^2 dx \right\}.$$

*For a fixed function space $\boldsymbol{F}(\Omega)$, consider the empirical loss induced by the Physics Informed Neural Network:*

$$\boldsymbol{E}_n(u) = \frac{1}{n} \sum_{j=1}^n \left[ |\Omega| \cdot \left( \Delta u(X_j) - V(X_j)u(X_j) + f(X_j) \right)^2 \right], \qquad \text{(B.64)}$$

*where $\{X_j\}_{j=1}^n$ are datapoints uniformly sampled from the domain $\Omega$. Then the Physics Informed Neural Network estimator associated with function space $\boldsymbol{F}(\Omega)$ is defined as the minimizer of $\boldsymbol{E}_n(u)$ over the function space $\boldsymbol{F}(\Omega)$:*

$$\hat{u}_{PINN} = \min_{u \in \boldsymbol{F}(\Omega)} \boldsymbol{E}_n(u).$$

*Moreover, we assume that there exists some constant $C > 0$ such that all function $u$ in the function space $\boldsymbol{F}(\Omega)$, the real solution $u^*$ and $f, V$ satisfy the following two conditions.*

- *The gradients and function value are uniformly bounded*

$$\max\left\{ \sup_{u \in \boldsymbol{F}(\Omega)} \|u\|_{L^\infty(\Omega)}, \sup_{u \in \boldsymbol{F}(\Omega)} \|\nabla u\|_{L^\infty(\Omega)}, \sup_{u \in \boldsymbol{F}(\Omega)} \|\Delta u\|_{L^\infty(\Omega)}, \right.$$
$$\left. \|u^*\|_{L^\infty(\Omega)}, \|\nabla u^*\|_{L^\infty(\Omega)}, \|\Delta u^*\|_{L^\infty(\Omega)}, V_{max}, \|f\|_{L^\infty(\Omega)} \right\} \le C. \qquad \text{(B.65)}$$

- *All the functions in the function space $\boldsymbol{F}(\Omega)$ satisfies the boundary condition*

$$u = 0 \text{ on } \partial\Omega.$$

*At the the same time, for any $\rho > 0$, we assume the Rademacher complexity of a localized function space $\boldsymbol{T}_\rho(\Omega) := \left\{ h := |\Omega| \cdot \left[ (\Delta u - Vu + f)^2 - (\Delta u^* - Vu^* + f)^2 \right] \,\Big|\, \|u - u^*\|_{H^2}^2 \le \rho \right\}$ can be upper bounded by a sub-root function $\phi = \phi(\rho) : [0, \infty) \to [0, \infty)$, i.e.*

$$\phi(4\rho) \le 2\phi(\rho) \text{ and } R_n(\boldsymbol{T}_\rho(\Omega)) \le \phi(\rho) \ (\forall \ \rho > 0). \qquad \text{(B.66)}$$

*For all constant $t > 0$. We denote $r^*$ to be the solution of the fix point equation of local Rademacher complexity $r = \phi(r)$. There exists two constants $C_p, C_q$ such that with probability $1 - C_p \exp(-C_q t)$, we have the following upper bound for the Physics Informed Neural Network Estimator*

$$\|\hat{u}_{PINN} - u^*\|_{H^2}^2 \lesssim \inf_{u_{\boldsymbol{F}} \in \boldsymbol{F}(\Omega)} \left( \boldsymbol{E}(u_{\boldsymbol{F}}) - \boldsymbol{E}(u^\star) \right) + \max\left\{ r^*, \frac{t}{n} \right\}.$$

*Proof.* To upper bound the excess risk $\Delta \boldsymbol{E}^{(n)}$, following(Xu, 2020; Lu et al., 2021b; Duan et al., 2021), we decompose the excess risk into approximation error and generalization error with probability $1 - e^{-C_q t}$, where $C_q > 0$ is some constant:

$$
\begin{aligned}
\Delta \boldsymbol{E}^{(n)}(\hat{u}_{\text{PINN}}) = \boldsymbol{E}(\hat{u}_{\text{PINN}}) - \boldsymbol{E}(u^\star) &= \big[\boldsymbol{E}(\hat{u}_{\text{PINN}}) - \boldsymbol{E}_n(\hat{u}_{\text{PINN}})\big] + \big[\boldsymbol{E}_n(\hat{u}_{\text{PINN}}) - \boldsymbol{E}_n(u_{\boldsymbol{F}})\big] \\
&\quad + \big[\boldsymbol{E}_n(u_{\boldsymbol{F}}) - \boldsymbol{E}(u_{\boldsymbol{F}})\big] + \big[\boldsymbol{E}(u_{\boldsymbol{F}}) - \boldsymbol{E}(u^\star)\big] \\
&\leq \big[\boldsymbol{E}(\hat{u}_{\text{PINN}}) - \boldsymbol{E}_n(\hat{u}_{\text{PINN}})\big] + \big[\boldsymbol{E}_n(u_{\boldsymbol{F}}) - \boldsymbol{E}(u_{\boldsymbol{F}})\big] + \big[\boldsymbol{E}(u_{\boldsymbol{F}}) - \boldsymbol{E}(u^\star)\big] \\
&\leq \big[\boldsymbol{E}(\hat{u}_{\text{PINN}}) - \boldsymbol{E}(u^*) + \boldsymbol{E}_n(u^*) - \boldsymbol{E}_n(\hat{u}_{\text{PINN}})\big] \\
&\quad + \frac{3}{2}\big[\boldsymbol{E}(u_{\boldsymbol{F}}) - \boldsymbol{E}(u^\star)\big] + \frac{t}{2n},
\end{aligned}
$$
(B.67)

where the expectation is on all sampled data. The inequality of the third line is because $\hat{u}_{\text{PINN}}$ is the minimizer of the empirical loss $\boldsymbol{E}_n$ in the solution set $\boldsymbol{F}(\Omega)$, so we have $\boldsymbol{E}_n(\hat{u}_{\text{PINN}}) \leq \boldsymbol{E}_n(u_{\boldsymbol{F}})$. The last inequality is based on the Bernstein inequality. The variance of $h = |\Omega| \cdot \big[(\Delta u - Vu + f)^2 - (\Delta u^* - Vu^* + f)^2\big]$ can be bounded by $\frac{\alpha'}{\alpha}\big[\boldsymbol{E}(u_{\boldsymbol{F}}) - \boldsymbol{E}(u^\star)\big]$ due to the strong convexity of the variation objective (B.69). According to the Bernstein inequality, there exists some constant $C_q > 0$, such that with probability $1 - e^{-C_q t}$ we have:

$$
\boldsymbol{E}_n(u_{\boldsymbol{F}}) - \boldsymbol{E}_n(u^*) - \boldsymbol{E}(u_{\boldsymbol{F}}) + \boldsymbol{E}(u^*) \leq \sqrt{\frac{t\big[\boldsymbol{E}(u_{\boldsymbol{F}}) - \boldsymbol{E}(u^\star)\big]}{n}} \leq \frac{1}{2}\big[\boldsymbol{E}(u_{\boldsymbol{F}}) - \boldsymbol{E}(u^\star)\big] + \frac{t}{2n}.
$$

Note that C.5 holds for all function lies in the function space $\boldsymbol{F}$. Thus, we can take $u_{\boldsymbol{F}} := \arg\min_{u_0 \in \boldsymbol{F}(\Omega)} \big(\boldsymbol{E}(u_0) - \boldsymbol{E}(u^\star)\big)$ and finally get

$$
\Delta \boldsymbol{E}^{(n)} \leq \underbrace{\boldsymbol{E}(\hat{u}_{\text{PINN}}) - \boldsymbol{E}(u^*) + \boldsymbol{E}_n(u^*) - \boldsymbol{E}_n(\hat{u}_{\text{PINN}})}_{\Delta \boldsymbol{E}_{\text{gen}}} + \frac{3}{2}\underbrace{\inf_{u_{\boldsymbol{F}} \in \boldsymbol{F}(\Omega)}\big(\boldsymbol{E}(u_{\boldsymbol{F}}) - \boldsymbol{E}(u^\star)\big)}_{\Delta \boldsymbol{E}_{\text{app}}} + \frac{t}{2n}.
$$

This inequality decompose the excess risk to the generalization error $\Delta \boldsymbol{E}_{\text{gen}} := \boldsymbol{E}(\hat{u}_{\text{PINN}}) - \boldsymbol{E}(u^*) + \boldsymbol{E}_n(u^*) - \boldsymbol{E}_n(\hat{u}_{\text{PINN}})$ and the approximation error $\Delta \boldsymbol{E}_{\text{app}} = \inf_{u_{\boldsymbol{F}} \in \boldsymbol{F}(\Omega)}\big(\boldsymbol{E}(u_{\boldsymbol{F}}) - \boldsymbol{E}(u^\star)\big)$.

From the lemmata proved in Section B.3, we already have an estimation of the approximation error's convergence rate. So now we'll focus on providing fast rate upper bounds of the generalization error for the two estimators using the localization techinque(Bartlett et al., 2005; Xu, 2020). To achieve the fast generalization bound, we focus on the following normalized empirical process

$$
\tilde{\boldsymbol{T}}_r(\Omega) := \Big\{\tilde{h}(x) := \frac{\mathbb{E}[h] - h(x)}{\mathbb{E}[h] + r} \mid h \in \boldsymbol{T}(\Omega)\Big\} \ (r > 0).
$$

First, we try to bound the expectation of the normalized empirical process. Applying the Symmetrization Lemma B.1, we can first bound the expectation as

$$
\sup_{\tilde{h} \in \tilde{\boldsymbol{T}}_r(\Omega)} \mathbb{E}_{x'}\Big[\frac{1}{n}\sum_{i=1}^n \tilde{h}(x_i')\Big] \leq \mathbb{E}_{x'}\Big[\sup_{h \in \boldsymbol{T}(\Omega)}\Big|\frac{1}{n}\sum_{i=1}^n \frac{h(x_i') - \mathbb{E}[h]}{\mathbb{E}[h] + r}\Big|\Big] \leq 2R_n(\hat{\boldsymbol{T}}_r(\Omega)).
$$

where the function class $\hat{\boldsymbol{S}}_r(\Omega)$ is defined as:

$$
\hat{\boldsymbol{T}}_r(\Omega) := \Big\{\hat{h}(x) := \frac{h(x)}{\mathbb{E}[h] + r} \mid h \in \boldsymbol{T}(\Omega)\Big\},
$$

where $\boldsymbol{T}(\Omega) = \Big\{h := |\Omega| \cdot \big[(\Delta u - Vu + f)^2 - (\Delta u^* - Vu^* + f)^2\big]\Big\}$. Then applying the Peeling Lemma B.4 to any function $h \in \boldsymbol{T}(\Omega)$ helps us upper bound the local Rademacher complexity $R_n(\hat{\boldsymbol{T}}_r(\Omega))$ with the function $\phi$ defined in equation B.66:

$$
R_n(\hat{\boldsymbol{T}}_r(\Omega)) = \mathbb{E}_\sigma\Big[\mathbb{E}_x\Big[\sup_{h \in \boldsymbol{T}(\Omega)} \frac{\frac{1}{n}\sum_{i=1}^n \sigma_i h(x_i)}{\mathbb{E}[h] + r}\Big]\Big] \leq \frac{4\phi(r)}{r}.
$$

Combining all inequalities derived above yields:

$$\sup_{\tilde{h} \in \tilde{T}_r(\Omega)} \mathbb{E}_{x'} \left[ \frac{1}{n} \sum_{i=1}^{n} \tilde{h}(x_i') \right] \leq 2R_n(\hat{\boldsymbol{T}}_r(\Omega)) \leq \frac{8\phi(r)}{r} \ (r > 0). \tag{B.68}$$

Secondly we'll apply the Talagrand concentration inequality, which requires us to verify the condition needed. We will first check that the expectation value $\mathbb{E}[h]$ is always non-negative for any $h \in \boldsymbol{S}(\Omega)$:

$$\mathbb{E}[h] = \frac{1}{|\Omega|} \int_\Omega |\Omega| \cdot (\Delta u - Vu + f)^2 dx - \frac{1}{|\Omega|} \int_\Omega |\Omega| \cdot (\Delta u^* - Vu^* + f)^2 dx$$
$$= \boldsymbol{E}(u) - \boldsymbol{E}(u^\star) \geq 0 \Rightarrow \mathbb{E}[h] \geq 0.$$

We will proceed to verify that any $\tilde{h} = \frac{\mathbb{E}[h]-h}{\mathbb{E}[h]+r} \in \tilde{\boldsymbol{T}}_r(\Omega)$ is of bounded inf-norm. We need to prove that any $h \in \boldsymbol{T}(\Omega)$ is of bounded inf-norm beforehand. Using boundedness condition listed in equation B.65 implies:

$$\|h\|_\infty = |\Omega| \cdot \|(\Delta u - Vu + f)^2 - (\Delta u^* - Vu^* + f)^2\|_\infty = |\Omega| \cdot \|(\Delta u - Vu + f)^2\|_\infty$$
$$\leq |\Omega| \cdot (\|\Delta u\|_\infty + V_{\max}\|u\|_\infty + \|f\|_\infty)^2 \leq |\Omega|(V_{\max} + 2)^2 C^2.$$

By taking $M := |\Omega|(V_{\max} + 2)^2 C^2$, we then have $\|h\|_\infty \leq M$ for all $h \in \boldsymbol{T}(\Omega)$. Note that the denominator can be lower bounded by $|\mathbb{E}[h] + r| \geq r > 0$. Combining these two inequalities help us upper bound the inf-norm $\|\tilde{h}\|_\infty = \sup_{x \in \Omega} |\tilde{h}(x)|$ as follows:

$$\|\tilde{h}\|_\infty = \frac{\|\mathbb{E}[h] - h\|_\infty}{|\mathbb{E}[h] + r|} \leq \frac{2\|h\|_\infty}{r} \leq \frac{2M}{r} =: \beta.$$

We will then check the normalized functions $\frac{\mathbb{E}[h]-h(x)}{\mathbb{E}[h]+r}$ in $\tilde{T}_r(\Omega)$ have bounded second moment, which is satisfied because of the regularity results of the PDE. We aim to show that there exist some constants $\alpha, \alpha' > 0$, such that for any $h \in \boldsymbol{T}(\Omega)$, the following inequality holds:

$$\alpha \mathbb{E}[h^2] \leq \|u - u^*\|_{H^2(\Omega)}^2 \leq \alpha' \mathbb{E}[h]. \tag{B.69}$$

The RHS of the inequality follows from strong convexity of the PINN objective function proved in Theorem B.2:

$$\mathbb{E}[h] = \boldsymbol{E}(u) - \boldsymbol{E}(u^*) \geq \min\{1, C_{\min}\}\|u - u^*\|_{H^2(\Omega)}^2.$$

The LHS of the inequality follows from boundedness condition listed in equation B.65 and the QM-AM inequality:

$$\mathbb{E}[h^2] = \int_\Omega \left[ (\Delta u - Vu + f)^2 - (\Delta u^* - Vu^* + f)^2 \right]^2 dx = \int_\Omega (\Delta u - Vu + f)^4 dx$$
$$\leq M^2 \int_\Omega (\Delta u - Vu - \Delta u^* + Vu^*)^2 dx \leq 2M^2 \int_\Omega [(\Delta u - \Delta u^*)^2 + V^2(u - u^*)^2] dx$$
$$\leq 2M^2 \max\{1, V_{\max}^2\}\|u - u^*\|_{H^2(\Omega)}^2$$

By picking $\alpha' = \frac{1}{\min\{1, C_{\min}\}}$ and $\alpha = \frac{1}{2M^2 \max\{1, V_{\max}^2\}}$, we have finished proving inequality B.69. Then we can can upper bound the expectation $\mathbb{E}[\tilde{h}^2]$ as:

$$\mathbb{E}[\tilde{h}^2] = \frac{\mathbb{E}[(h - \mathbb{E}[h])^2]}{|\mathbb{E}[h] + r|^2} = \frac{\mathbb{E}[h^2] - \mathbb{E}[h]^2}{|\mathbb{E}[h] + r|^2} \leq \frac{\mathbb{E}[h^2]}{|\mathbb{E}[h] + r|^2}.$$

Using the fact that $\mathbb{E}[h] \geq 0$ and inequality B.69, we can lower bound the denominator $|\mathbb{E}[h] + r|^2$ as follows:

$$|\mathbb{E}[h] + r|^2 \geq 2\mathbb{E}[h]r \geq \frac{2r\alpha}{\alpha'}\mathbb{E}[h^2].$$

Therefore, we can deduce that:

$$\mathbb{E}[\tilde{h}^2] \leq \frac{\mathbb{E}[h^2]}{|\mathbb{E}[h] + r|^2} \leq \frac{\mathbb{E}[h^2]}{\frac{2r\alpha}{\alpha'}\mathbb{E}[h^2]} = \frac{\alpha'}{2r\alpha} =: \sigma^2.$$

Hence, any function in the localized class $\tilde{T}_r(\Omega)$ is of bounded second moment.

It is easy to check that for any $\tilde{h} \in \tilde{T}_r(\Omega)$, we have

$$\mathbb{E}[\tilde{h}] = \frac{\mathbb{E}[h] - \mathbb{E}[h]}{\mathbb{E}[h] + r} = 0,$$

*i.e.* any function in the localized class $\tilde{S}_r(\Omega)$ is of zero mean.

Now we have verified that any function $\tilde{h} \in \tilde{S}_r(\Omega)$ satisfies all the required conditions. By taking $\mu$ to be the uniform distribution on the domain $\Omega$ and applying Talagrand's Concentration inequality given in Lemma B.3, we have:

$$\mathbb{P}_x\left[\sup_{\tilde{h} \in \tilde{T}_r(\Omega)} \frac{1}{n}\sum_{i=1}^{n}\tilde{h}(x_i) \geq 2\sup_{\tilde{h} \in \tilde{T}_r(\Omega)} \mathbb{E}_{x'}\left[\frac{1}{n}\sum_{i=1}^{n}\tilde{h}(x'_i)\right] + \sqrt{\frac{2t\sigma^2}{n}} + \frac{2t\beta}{n}\right] \leq e^{-t}.$$

By using the upper bound deduced above and plugging in the expressions of $\beta$ and $\sigma$, we can rewrite Talagrand's Concentration Inequality in the following way. With probability at least $1 - e^{-t}$, the inequality below holds:

$$\frac{1}{n}\sum_{i=1}^{n}\tilde{h}(x_i) \leq \sup_{\tilde{h} \in \tilde{S}_r(\Omega)} \frac{1}{n}\sum_{i=1}^{n}\tilde{h}(x_i) \leq 2\sup_{\tilde{h} \in \tilde{S}_r(\Omega)} \mathbb{E}_{x'}\left[\frac{1}{n}\sum_{i=1}^{n}\tilde{h}(x'_i)\right] + \sqrt{\frac{2t\sigma^2}{n}} + \frac{2t\beta}{n}$$

$$\leq \frac{16\phi(r)}{r} + \sqrt{\frac{t\alpha'}{n\alpha r}} + \frac{4Mt}{nr} =: \psi(r).$$

Let's pick the critical radius $r_0$ to be:

$$r_0 = \max\{2^{14}r^*, \frac{24Mt}{n}, \frac{36\alpha't}{\alpha n}\}. \tag{B.70}$$

Note that concavity of the function $\phi$ implies that $\phi(r) \leq r$ for any $r \geq r^*$. Combining this with the first inequality listed in B.66 yields:

$$\frac{16\phi(r)}{r} \leq \frac{2^{11}\phi(\frac{r_0}{2^{14}})}{2^{14}\frac{r_0}{2^{14}}} = \frac{1}{8} \times \frac{\phi(\frac{r_0}{2^{14}})}{\frac{r_0}{2^{14}}} \leq \frac{1}{8}.$$

On the other hand, applying equation B.70 yields:

$$\sqrt{\frac{\alpha't}{n\alpha r_0}} \leq \sqrt{\frac{\alpha't}{n\alpha}\frac{\alpha n}{36\alpha't}} = \frac{1}{6},$$

$$\frac{4Mt}{nr_0} \leq \frac{4Mt}{n} \times \frac{n}{24Mt} = \frac{1}{6}.$$

Summing the three inequalities above implies:

$$\psi(r_0) = \frac{16\phi(r_0)}{r_0} + \sqrt{\frac{t\alpha'}{n\alpha r_0}} + \frac{4Mt}{nr_0} \leq \frac{1}{8} + \frac{1}{6} + \frac{1}{6} < \frac{1}{2}.$$

By picking $r = r_0$, we can further deduce that for any function $u \in F(\Omega)$, the following inequality holds with probability $1 - e^{-t}$:

$$\frac{E(u) - E(u^*) - E_n(u) + E_n(u^*)}{E(u) - E(u^*) + r_0} = \frac{1}{n}\sum_{i=1}^{n}\tilde{h}(x_i) \leq \psi(r_0) < \frac{1}{2}.$$

Multiplying the denominator on both sides indicates:

$$\Delta E_{\text{gen}} = E(u) - E(u^*) - E_n(u) + E_n(u^*) \leq \frac{1}{2}\left[E(u) - E(u^*)\right] + \frac{1}{2}r_0 = \frac{1}{2}\Delta E^{(n)} + \frac{1}{2}r_0.$$

Substituting the upper bound above into the decomposition $\Delta \boldsymbol{E}^{(n)} \leq \Delta E_{\text{gen}} + \frac{3}{2}\Delta E_{\text{app}} + \frac{t}{2n}$ yields that with probability $1 - 2e^{-\min\{1,C_q\}t}$, we have:

$$\Delta \boldsymbol{E}^{(n)} \leq \Delta \boldsymbol{E}_{\text{gen}} + \frac{3}{2}\Delta \boldsymbol{E}_{\text{app}} + \frac{t}{2n} \leq \frac{1}{2}\Delta \boldsymbol{E}^{(n)} + \frac{1}{2}r_0 + \frac{3}{2}\Delta \boldsymbol{E}_{\text{app}} + \frac{t}{2n}.$$

Simplifying the inequality above yields that with probability $1 - 2e^{-\min\{1,C_q\}t}$, we have:

$$\Delta \boldsymbol{E}^{(n)} \leq r_0 + 3\Delta \boldsymbol{E}_{\text{app}} + \frac{t}{n} = 3 \inf_{u_{\boldsymbol{F}} \in \boldsymbol{F}(\Omega)} \left( \boldsymbol{E}(u_{\boldsymbol{F}}) - \boldsymbol{E}(u^\star) \right) + \max\{2^{14}r^*, 24M\frac{t}{n}, \frac{36\alpha'}{\alpha}\frac{t}{n}\} + \frac{t}{n}$$

$$\lesssim \inf_{u_{\boldsymbol{F}} \in \boldsymbol{F}(\Omega)} \left( \boldsymbol{E}(u_{\boldsymbol{F}}) - \boldsymbol{E}(u^\star) \right) + \max\left\{ r^*, \frac{t}{n} \right\}.$$

Moreover, using strong convexity of the PINN objective function proved in Theorem B.2 implies:

$$\Delta \boldsymbol{E}^{(n)} = \boldsymbol{E}(\hat{u}_{\text{PINN}}) - \boldsymbol{E}(u^*) \geq \min\{1, C_{\min}\}\|\hat{u}_{\text{PINN}} - u^*\|^2_{H^2(\Omega)}.$$

Combining the two bounds above yields that with probability $1 - 2e^{-\min\{1,C_q\}t}$, we have:

$$\|\hat{u}_{\text{PINN}} - u^*\|^2_{H^2(\Omega)} \lesssim \inf_{u_{\boldsymbol{F}} \in \boldsymbol{F}(\Omega)} \left( \boldsymbol{E}(u_{\boldsymbol{F}}) - \boldsymbol{E}(u^\star) \right) + \max\left\{ r^*, \frac{t}{n} \right\}.$$

$\square$

**Deep Neural Network Estimator.** For any $N \in \mathbb{Z}^+$, there exists some Deep Neural Network in $\Phi(L, W, S, B)$ with $L = O(1)$, $W = O(N)$, $S = O(N)$, $B = O(N)$, such that the approximation error $\Delta \boldsymbol{E}_{\text{app}} = O(N^{-\frac{2(s-2)}{d}})$ and generalization error $\Delta \boldsymbol{E}_{\text{gen}} = O(\frac{N \log N}{n})$. With optimal selection $N = n^{\frac{d}{d+2s-4}}$ to balance the bias and variance, we can achieve $n^{-\frac{2s-4}{d+2s-4}} \log n$ convergence rate for PINN estimator.

**Theorem B.13.** *(Final Upper Bound of PINN with Deep Neural Network Estimator) Under the assumptions in Theorem B.12, we consider the PINN objective with the sparse Deep Neural Network function space $\Phi(L, W, S, B)$, where the parameters $L = O(1)$, $W = O(n^{\frac{d}{d+2s-4}})$, $S = O(n^{\frac{d}{d+2s-4}})$ and $B = O(n^{\frac{d}{d+2s-4}})$. Then we have that the DNN estimator $\hat{u}_{PINN}^{DNN} = \min_{u \in \Phi(L,W,S,B)} \boldsymbol{E}_n^{PINN}(u)$ satisfies the following upper bound with high probability:*

$$\|\hat{u}_{PINN}^{DNN} - u^*\|^2_{H^2} \lesssim n^{-\frac{2s-4}{d+2s-4}} \log n.$$

*Proof.* On the one hand, by taking $s = 2$ and $p = 2$ in Theorem B.8 proved above, we have that there exists some Deep Neural Network $u_{\text{DNN}} \in \Phi(L, W, S, B)$ with $L = O(1), W = O(N), S = O(N), B = O(N)$, such that.

$$\|u_{\text{DNN}} - u^*\|^2_{H^2(\Omega)} \leq N^{-\frac{2s-4}{d}}\|u\|_{H^s(\Omega)}.$$

Applying strong convexity of the PINN objective function proved in Theorem B.2 yields the following upper bound on the approximation error of PINN:

$$\Delta \boldsymbol{E}_{\text{app}} \lesssim \|u_{\text{DNN}} - u^*\|^2_{H^2(\Omega)} \leq N^{-\frac{2s-4}{d}}.$$

On the other hand, from lemma B.18 proved above, we know that the function $\phi(\rho)$ that upper bounds the local Rademacher complexity of the Deep Neural Networks $u_{\text{DNN}}$ is of the same magnitude as $\sqrt{\frac{S3^L\rho}{n}} \log(BWn)$. By plugging in the magnitudes of $L, W, S, B$, we can determine the critical radius $r^*$:

$$\sqrt{\frac{r^*3^L S}{n}} \log(BWn) \simeq \sqrt{\frac{r^* N}{n}(2\log N + \log n)} \simeq r^* \Rightarrow r^* \simeq \frac{N(\log N + \log n)}{n}.$$

Combining the two bounds above with Theorem B.12 yields that with high probability, we have:

$$\|\hat{u}_{\text{PINN}}^{\text{DNN}} - u^*\|^2_{H^2} \lesssim \Delta \boldsymbol{E}_{\text{app}} + r^* \lesssim N^{-\frac{2(s-2)}{d}} + \frac{N(\log N + \log n)}{n}.$$

By equating the two terms above, we can solve for the optimal $N$ that yields the desired bound:

$$N^{-\frac{2(s-2)}{d}} \simeq \frac{N}{n} \Rightarrow N \simeq n^{\frac{d}{d+2s-4}}.$$

Plugging in the optimal $N$ gives us the magnitudes of the four parameters $L = O(1)$, $W = O(n^{\frac{d}{d+2s-4}})$, $S = O(n^{\frac{d}{d+2s-4}})$, $B = O(n^{\frac{d}{d+2s-4}})$, as well as the final rate:

$$\|\hat{u}_{\text{PINN}}^{\text{DNN}} - u^*\|_{H^2}^2 \lesssim N^{-\frac{2(s-2)}{d}} + \frac{N \log N}{n} \lesssim n^{-\frac{2(s-2)}{d+2(s-2)}} \log n.$$

$\square$

**Truncated Fourier Series Estimator.** For any $\xi \in \mathbb{Z}^+$, there exists some Truncated Fourier Series in $\boldsymbol{F}_\xi(\Omega)$ with approximation error $\Delta \boldsymbol{E}_{\text{app}} = O(\xi^{-2(s-2)})$ and generalization error $\Delta \boldsymbol{E}_{\text{gen}} = O(\frac{\xi^d}{n})$. With optimal selection $\xi = n^{\frac{1}{d+2s-4}}$ to balance the bias and variance, we can achieve $n^{-\frac{2s-4}{d+2s-4}}$ convergence rate for the Fourier estimator.

**Theorem B.14.** *(Final Upper Bound of PINN with Truncated Fourier Series Estimator) Under the assumptions in Theorem B.12, we consider the PINN objective with the Truncated Fourier Series function space $\boldsymbol{F}_\xi(\Omega)$, where the parameter $\xi = \Theta(n^{\frac{1}{d+2s-4}})$. Then we have that the Fourier estimator $\hat{u}_{PINN}^{Fourier} = \min_{u \in \boldsymbol{F}_\xi(\Omega)} \boldsymbol{E}_n^{PINN}(u)$ satisfies the following upper bound with high probability:*

$$\|\hat{u}_{PINN}^{Fourier} - u^*\|_{H^2}^2 \lesssim n^{-\frac{2s-4}{d+2s-4}}$$

*Proof.* Let's firstly derive the function $\phi(\rho)$ that upper bounds the local Rademacher complexity of $\boldsymbol{T}_\rho(\Omega) := \left\{ h := |\Omega| \cdot \left[ (\Delta u - Vu + f)^2 - (\Delta u^* - Vu^* + f)^2 \right] \;\middle|\; u \in \boldsymbol{J}_{\rho,\xi}(\Omega) \right\}$, where $\boldsymbol{J}_{\rho,\xi}(\Omega) = \left\{ v \in F_\xi(\Omega) \;\middle|\; \|v\|_{H^2(\Omega)}^2 \leq \rho \right\}$ denotes the localized Truncated Fourier Series space. From Talagrand Contraction Lemma B.2, Lemma B.6 and Lemma B.8 proved above, we have

$$
\begin{aligned}
R_n(\boldsymbol{T}_\rho(\Omega)) &\lesssim R_n \left( \left\{ u - u^* : u \in \boldsymbol{J}_{\rho,\xi}(\Omega), \|u - u^*\|_{H^2(\Omega)} \leq \sqrt{\rho} \right\} \right) \\
&\quad + R_n \left( \left\{ \Delta u - \Delta u^* : u \in \boldsymbol{J}_{\rho,\xi}(\Omega), \|u - u^*\|_{H^2(\Omega)} \leq \sqrt{\rho} \right\} \right) \\
&\lesssim \mathbb{E}_X \left[ \mathbb{E}_\sigma \left[ \sup_{u \in \boldsymbol{F}_{\rho,\xi}(\Omega)} \frac{1}{n} \sum_{i=1}^n \sigma_i (u(X_i) - \Pi_\xi u^*(X_i)) \middle| \|u - \Pi_\xi u^*\|_{H^1(\Omega)}^2 \leq \rho \right] \right] \\
&\quad + \mathbb{E}_X \left[ \mathbb{E}_\sigma \left[ \sup_{u \in \boldsymbol{F}_{\rho,\xi}(\Omega)} \frac{1}{n} \sum_{i=1}^n \sigma_i (\Delta u(X_i) - \Delta \Pi_\xi u^*(X_i)) \middle| \|u - \Pi_\xi u^*\|_{H^1(\Omega)}^2 \leq \rho \right] \right] \\
&\quad + \mathbb{E}_X \left[ \mathbb{E}_\sigma \left[ \frac{1}{n} \sum_{i=1}^n \sigma_i \Delta \Pi_{>\xi} u^*(X_i) \right] \right] + \mathbb{E}_X \left[ \mathbb{E}_\sigma \left[ \frac{1}{n} \sum_{i=1}^n \sigma_i \Pi_{>\xi} u^*(X_i) \right] \right] \\
&\lesssim \mathbb{E}_X \left[ \mathbb{E}_\sigma \left[ \sup_{v \in \boldsymbol{F}_{\rho,\xi}(\Omega)} \frac{1}{n} \sum_{i=1}^n \sigma_i v(X_i) \middle| \|v\|_{H^2(\Omega)}^2 \leq \rho \right] \right] \\
&\quad + \mathbb{E}_X \left[ \mathbb{E}_\sigma \left[ \sup_{v \in \boldsymbol{F}_{\rho,\xi}(\Omega)} \frac{1}{n} \sum_{i=1}^n \sigma_i \Delta v(X_i) \middle| \|v\|_{H^2(\Omega)}^2 \leq \rho \right] \right] \\
&\quad + \mathbb{E}_X \left[ \mathbb{E}_\sigma \left[ \frac{1}{n} \sum_{i=1}^n \sigma_i \Delta \Pi_{>\xi} u^*(X_i) \right] \right] + \mathbb{E}_X \left[ \mathbb{E}_\sigma \left[ \frac{1}{n} \sum_{i=1}^n \sigma_i \Pi_{>\xi} u^*(X_i) \right] \right] \\
&\lesssim \sqrt{\frac{\rho}{n}} \xi^{\frac{d}{2}} + \sqrt{\frac{\|\Pi_{>\xi} u^*\|_{H^2}^2}{n}} \lesssim \sqrt{\frac{\rho}{n}} \xi^{\frac{d}{2}} + \sqrt{\frac{\xi^{-2(s-2)}}{n}} \lesssim \sqrt{\frac{\rho}{n}} \xi^{\frac{d}{2}} + \frac{1}{n} + \xi^{-2(s-2)},
\end{aligned}
$$

(B.71)

where $\Pi_\xi u := \sum_{\|z\|_\infty \leq \xi} u_z \phi_z(x)$ is the projection to the Fourier basis whose frequency is smaller than $\xi$ and $\Pi_{>\xi} u := \sum_{\|z\|_\infty \leq \xi} u_z \phi_z(x)$ is the projection to the Fourier basis whose frequency is

larger than $\xi$. Then, the critical radius $r^*$ can be determined as follows:

$$\sqrt{\frac{r^*}{n}}\xi^{\frac{d}{2}} + \frac{1}{n} + \xi^{-2(s-2)} \simeq r^* \Rightarrow r^* \simeq \frac{\xi^d}{n} + \frac{1}{n} + \xi^{-2(s-2)},$$

Moreover, by taking $\alpha = s$ and $\beta = 1$ in Lemma B.19 and applying strong convexity of the DRM objective function proved in Theorem B.2, we can upper bound the approximation error $\Delta \boldsymbol{E}_{\text{app}}$ as below:

$$\Delta \boldsymbol{E}_{\text{app}} \lesssim \xi^{-2(s-2)}.$$

Combining the two bounds above with Theorem B.12 yields that with high probability, we have:

$$\|\hat{u}_{\text{PINN}}^{\text{Fourier}} - u^*\|_{H^2}^2 \lesssim \Delta \boldsymbol{E}_{\text{app}} + r^* \lesssim \frac{\xi^d}{n} + \frac{1}{n} + \xi^{-2(s-2)}.$$

By equating two of the three terms above, we can solve for $\xi$ that yields the desired bound:

$$\frac{\xi^d}{n} \simeq \xi^{-2(s-2)} \Rightarrow \xi \simeq n^{\frac{1}{d+2s-4}}.$$

Plugging in the optimal $\xi$ gives us the final rate:

$$\|\hat{u}_{\text{PINN}}^{\text{Fourier}} - u^*\|_{H^2}^2 \lesssim \frac{\xi^d}{n} + \xi^{-2(s-2)} + \frac{1}{n} \lesssim n^{-\frac{2s-4}{d+2s-4}}.$$

$\square$

## C  PROOF OF MODIFIED DRM

In this section, we provide the proof of the modified deep Ritz method here. We first provide a similar meta-theorem as we did for DRM.

**Theorem C.1** (Meta-theorem for Upper Bounds of Modified Deep Ritz Method). *Let $u^* \in H^s(\Omega)$ denote the true solution to the PDE model with Dirichlet boundary condition:*

$$\begin{aligned} -\Delta u + Vu &= f \text{ on } \Omega, \\ u &= 0 \text{ on } \partial\Omega, \end{aligned} \tag{C.1}$$

*where $f \in L^2(\Omega)$ and $V \in L^\infty(\Omega)$ with $0 < V_{\min} \le V(x) \le V_{\max} > 0$. In Theorem B.1, it has been proved that $u^*$ can be obtained by minimizing the loss $\boldsymbol{E}(u)$:*

$$u^* = \underset{u \in H_0^1(\Omega)}{\arg\min} \boldsymbol{E}(u) := \underset{u \in H_0^1(\Omega)}{\arg\min} \left\{ \frac{1}{2} \int_\Omega \left[ \|\nabla u\|^2 + V|u|^2 \right] dx - \int_\Omega fu dx \right\}.$$

*For a fixed function space $\boldsymbol{F}(\Omega)$, consider the empirical loss induced by the Modified Deep Ritz Method ($N \ge n$):*

$$\boldsymbol{E}_{N,n}(u) = \frac{1}{N} \sum_{i=1}^N \left[ |\Omega| \cdot \frac{1}{2} \|\nabla u(X_i')\|^2 \right] + \frac{1}{n} \sum_{j=1}^n \left[ |\Omega| \cdot \left( \frac{1}{2} V(X_j)|u(X_j)|^2 - f(X_j)u(X_j) \right) \right], \tag{C.2}$$

*where $\{X_i'\}_{i=1}^N$ and $\{X_j\}_{j=1}^n$ are datapoints uniformly and independently sampled from the domain $\Omega$. Then the Modified Deep Ritz estimator associated with function space $\boldsymbol{F}(\Omega)$ is defined as the minimizer of $\boldsymbol{E}_{N,n}(u)$ over the function space $\boldsymbol{F}(\Omega)$:*

$$\hat{u}_{MDRM} = \min_{u \in \boldsymbol{F}(\Omega)} \boldsymbol{E}_{N,n}(u).$$

*Moreover, we assume that there exists some constant $C > 0$ such that all function $u$ in the function space $\boldsymbol{F}(\Omega)$, the real solution $u^*$ and $f, V$ satisfy the following two conditions.*

- *The gradients and function value are uniformly bounded*

$$\max\left\{ \sup_{u \in \boldsymbol{F}(\Omega)} \|u\|_{L^\infty(\Omega)}, \sup_{u \in \boldsymbol{F}(\Omega)} \|\nabla u\|_{L^\infty(\Omega)}, \|u^*\|_{L^\infty(\Omega)}, \|\nabla u^*\|_{L^\infty(\Omega)}, V_{max}, \|f\|_{L^\infty(\Omega)} \right\} \le C. \tag{C.3}$$

- *All the functions in the function space $\boldsymbol{F}(\Omega)$ satisfy the boundary condition*

$$u = 0 \text{ on } \partial\Omega.$$

*At the the same time, for any $\rho > 0$, we assume the Rademacher complexity of the following vector-valued function space*

$$\boldsymbol{S}_\rho(\Omega) := \left\{ (h_1, h_2) \big| h_1 := |\Omega| \cdot \left[ \frac{1}{2}\Big( \|\nabla u\|^2 - \|\nabla u^*\|^2 \Big) \right], \right.$$
$$\left. h_2 := |\Omega| \cdot \left[ \frac{1}{2} V(|u|^2 - |u^*|^2) - f(u - u^*) \right], \|u - u^*\|_{H^1}^2 \le \rho \right\}.$$

*can be upper bounded by a sub-root function $\phi = \phi(\rho) : [0, \infty) \to [0, \infty)$, i.e.*

$$\phi(4\rho) \le 2\phi(\rho) \text{ and } R_{N,n}(\boldsymbol{S}_\rho(\Omega)) \le \phi(\rho) \ (\forall \ \rho > 0), \tag{C.4}$$

*where $R_{N,n}(\boldsymbol{S}) := R_N(\{h_1|(h_1, h_2) \in \boldsymbol{S}\}) + R_n(\{h_2|(h_1, h_2) \in \boldsymbol{S}\})$. For all constant $t > 0$. We denote $r^*$ to be the solution of the fix point equation of local Rademacher complexity $r = \phi(r)$. There exists two constants $C_p, C_q$ such that with probability $1 - C_p \exp(-C_q t)$, we have the following upper bound for the Modified Deep Ritz Estimator*

$$\|\hat{u}_{MDRM} - u^*\|_{H^1}^2 \lesssim \inf_{u_{\boldsymbol{F}} \in \boldsymbol{F}(\Omega)} \Big( \boldsymbol{E}(u_{\boldsymbol{F}}) - \boldsymbol{E}(u^\star) \Big) + \max \left\{ r^*, \frac{t}{n} \right\}.$$

*Proof.* To upper bound the excess risk $\Delta \boldsymbol{E}^{(N,n)} := \boldsymbol{E}(\hat{u}_{MDRM}) - \boldsymbol{E}(u^*)$, following(Xu, 2020; Lu et al., 2021b; Duan et al., 2021), we decompose the excess risk into approximation error and generalization error with probability $1 - e^{-t}$:

$$
\begin{aligned}
\Delta \boldsymbol{E}^{(N,n)} = \big[ \boldsymbol{E}(\hat{u}_{MDRM}) - \boldsymbol{E}(u^\star) \big] = &\big[ \boldsymbol{E}(\hat{u}_{MDRM}) - \boldsymbol{E}_{N,n}(\hat{u}_{MDRM}) \big] + \big[ \boldsymbol{E}_{N,n}(\hat{u}_{MDRM}) - \boldsymbol{E}_{N,n}(u_{\boldsymbol{F}}) \big] \\
&+ \big[ \boldsymbol{E}_{N,n}(u_{\boldsymbol{F}}) - \boldsymbol{E}(u_{\boldsymbol{F}}) \big] + \big[ \boldsymbol{E}(u_{\boldsymbol{F}}) - \boldsymbol{E}(u^\star) \big] \\
\le &\big[ \boldsymbol{E}(\hat{u}_{MDRM}) - \boldsymbol{E}_{N,n}(\hat{u}_{MDRM}) \big] + \big[ \boldsymbol{E}_{N,n}(u_{\boldsymbol{F}}) - \boldsymbol{E}(u_{\boldsymbol{F}}) \big] + \big[ \boldsymbol{E}(u_{\boldsymbol{F}}) - \boldsymbol{E}(u^\star) \big] \\
\le &\big[ \boldsymbol{E}(\hat{u}_{MDRM}) - \boldsymbol{E}(u^*) + \boldsymbol{E}_{N,n}(u^*) - \boldsymbol{E}_{N,n}(\hat{u}_{MDRM})] \big] \\
&+ 2\big[ \boldsymbol{E}(u_{\boldsymbol{F}}) - \boldsymbol{E}(u^\star) \big] + \frac{4t}{\min\{N, n\}},
\end{aligned}
\tag{C.5}
$$

where the expectation is on all sampled data. The inequality of the third line is because $\hat{u}_{MDRM}$ is the minimizer of the empirical loss $\boldsymbol{E}_n$ in the solution set $\boldsymbol{F}(\Omega)$, so we have $\boldsymbol{E}_{N,n}(\hat{u}_{MDRM}) \le \boldsymbol{E}_{N,n}(u_{\boldsymbol{F}})$. The last inequality is based on the Bernstein inequality. For any $u_{\boldsymbol{F}} \in \boldsymbol{F}(\Omega)$, we use $h_{\boldsymbol{F},1}, h_{\boldsymbol{F},2}$ to denote the following two functions:

$$
\begin{aligned}
h_{\boldsymbol{F},1} &:= \frac{1}{2}\Big( \|\nabla u_{\boldsymbol{F}}\|^2 - \|\nabla u^*\|^2 \Big), \\
h_{\boldsymbol{F},2} &:= \frac{1}{2} V(|u_{\boldsymbol{F}}|^2 - |u^*|^2) - f(u_{\boldsymbol{F}} - u^*).
\end{aligned}
$$

Applying Bernstein's inequality twice to $h_{\boldsymbol{F},1}$ and $h_{\boldsymbol{F},2}$ implies that there exists some constant $C_q$, such that with probability $1 - 2e^{-C_q t}$, the following two inequalities hold simultaneously:

$$
\boldsymbol{E}_N(h_{\boldsymbol{F},1}) - \boldsymbol{E}(h_{\boldsymbol{F},1}) \le \sqrt{\frac{t\frac{\alpha}{\alpha'} \boldsymbol{E}[h_{\boldsymbol{F},1}^2]}{N}},
$$
$$
\boldsymbol{E}_n(h_{\boldsymbol{F},2}) - \boldsymbol{E}(h_{\boldsymbol{F},2}) \le \sqrt{\frac{t\frac{\alpha}{\alpha'} \boldsymbol{E}[h_{\boldsymbol{F},2}^2]}{n}}.
$$

Note that the variance sum $\boldsymbol{E}[h_{\boldsymbol{F},1}^2] + \boldsymbol{E}[h_{\boldsymbol{F},2}^2]$ can be upper bounded by $\frac{\alpha'}{\alpha}\big[ \boldsymbol{E}(u_{\boldsymbol{F}}) - \boldsymbol{E}(u^\star) \big]$ due to the strong convexity of the variation objective (C.9). Adding the two inequalities above implies

with probability $1 - 2e^{-C_q t}$ we have:

$$\boldsymbol{E}_{N,n}(u_{\boldsymbol{F}}) - \boldsymbol{E}_{N,n}(u^*) - \boldsymbol{E}(u_{\boldsymbol{F}}) + \boldsymbol{E}(u^*) = \boldsymbol{E}_N(h_{\boldsymbol{F},1}) - \boldsymbol{E}(h_{\boldsymbol{F},1}) + \boldsymbol{E}_n(h_{\boldsymbol{F},2}) - \boldsymbol{E}(h_{\boldsymbol{F},2})$$

$$\leq \sqrt{\frac{t\frac{\alpha}{\alpha'}\boldsymbol{E}[h_{\boldsymbol{F},1}^2]}{N}} + \sqrt{\frac{t\frac{\alpha}{\alpha'}\boldsymbol{E}[h_{\boldsymbol{F},2}^2]}{n}}$$

$$\leq \sqrt{\frac{2t\frac{\alpha}{\alpha'}\Big(\boldsymbol{E}[h_{\boldsymbol{F},1}^2] + \boldsymbol{E}[h_{\boldsymbol{F},1}^2]\Big)}{\min\{N,n\}}}$$

$$\leq \sqrt{\frac{2t\big[\boldsymbol{E}(u_{\boldsymbol{F}}) - \boldsymbol{E}(u^\star)\big]}{\min\{N,n\}}} \leq \big[\boldsymbol{E}(u_{\boldsymbol{F}}) - \boldsymbol{E}(u^\star)\big] + \frac{4t}{\min\{N,n\}}.$$

Note that C.5 holds for all function lies in the function space $\boldsymbol{F}$. Thus, we can take $u_{\boldsymbol{F}} := \arg\min_{u_{\boldsymbol{F}}\in\boldsymbol{F}(\Omega)}\Big(\boldsymbol{E}(u_{\boldsymbol{F}}) - \boldsymbol{E}(u^\star)\Big)$ and finally get:

$$\Delta\boldsymbol{E}^{(N,n)} \leq \underbrace{\boldsymbol{E}(\hat{u}_{\mathrm{MDRM}}) - \boldsymbol{E}(u^*) + \boldsymbol{E}_{N,n}(u^*) - \boldsymbol{E}_{N,n}(\hat{u}_{\mathrm{MDRM}})}_{\Delta\boldsymbol{E}_{\mathrm{gen}}} + 2\underbrace{\inf_{u_{\boldsymbol{F}}\in\boldsymbol{F}(\Omega)}\Big(\boldsymbol{E}(u_{\boldsymbol{F}}) - \boldsymbol{E}(u^\star)\Big)}_{\Delta\boldsymbol{E}_{\mathrm{app}}} + \frac{4t}{n}.$$

This inequality decomposes the excess risk to the generalization error $\Delta\boldsymbol{E}_{\mathrm{gen}} := \boldsymbol{E}(\hat{u}_{\mathrm{MDRM}}) - \boldsymbol{E}(u^*) + \boldsymbol{E}_{N,n}(u^*) - \boldsymbol{E}_{N,n}(\hat{u}_{\mathrm{MDRM}})$ and the approximation error $\Delta\boldsymbol{E}_{\mathrm{app}} = \inf_{u_{\boldsymbol{F}}\in\boldsymbol{F}(\Omega)}\Big(\boldsymbol{E}(u_{\boldsymbol{F}}) - \boldsymbol{E}(u^\star)\Big)$. From the lemmata proved in Section B.3, we already have an estimation of the approximation error's convergence rate. So now we'll focus on providing fast rate upper bounds of the generalization error for the two estimators using the localization techinque(Bartlett et al., 2005; Xu, 2020). To achieve the fast generalization bound, we focus on the following two normalized empirical processes:

$$\tilde{\boldsymbol{S}}_{r,1}(\Omega) := \big\{\tilde{h}_1(x) := \frac{\mathbb{E}[h_1] - h_1(x)}{\mathbb{E}[h_1] + \mathbb{E}[h_2] + r} \mid (h_1, h_2) \in \boldsymbol{S}(\Omega)\big\} \ (r > 0),$$

$$\tilde{\boldsymbol{S}}_{r,2}(\Omega) := \big\{\tilde{h}_2(x) := \frac{\mathbb{E}[h_2] - h_2(x)}{\mathbb{E}[h_1] + \mathbb{E}[h_2] + r} \mid (h_1, h_2) \in \boldsymbol{S}(\Omega)\big\} \ (r > 0).$$

where the space $\boldsymbol{S}(\Omega)$ is defined as:

$$\boldsymbol{S}(\Omega) := \Big\{(h_1, h_2)\big|h_1 := |\Omega| \cdot \Big[\frac{1}{2}\Big(\|\nabla u\|^2 - \|\nabla u^*\|^2\Big)\Big],$$
$$h_2 := |\Omega| \cdot \Big[\frac{1}{2}V(|u|^2 - |u^*|^2) - f(u - u^*)\Big], u \in \boldsymbol{F}(\Omega)\Big\}.$$

First, we try to bound the expectation of the two normalized empirical processes. Applying the Symmetrization Lemma B.1, we can first bound the two expectations as:

$$\sup_{\tilde{h}_1\in\tilde{S}_{r,1}(\Omega)} \mathbb{E}_{y'}\Big[\frac{1}{N}\sum_{i=1}^N \tilde{h}_1(y_i')\Big] \leq \mathbb{E}_{y'}\Big[\sup_{h_1\in S_1(\Omega)}\Big|\frac{1}{N}\sum_{i=1}^N \frac{h_1(y_i') - \mathbb{E}[h_1]}{\mathbb{E}[h_1] + \mathbb{E}[h_2] + r}\Big|\Big] \leq 2R_N(\hat{\boldsymbol{S}}_{r,1}(\Omega)),$$

$$\sup_{\tilde{h}_2\in\tilde{S}_{r,2}(\Omega)} \mathbb{E}_{y}\Big[\frac{1}{n}\sum_{j=1}^n \tilde{h}_2(y_j)\Big] \leq \mathbb{E}_{y}\Big[\sup_{h_2\in S_2(\Omega)}\Big|\frac{1}{n}\sum_{i=1}^n \frac{h_2(y_j) - \mathbb{E}[h_2]}{\mathbb{E}[h_1] + \mathbb{E}[h_2] + r}\Big|\Big] \leq 2R_n(\hat{\boldsymbol{S}}_{r,2}(\Omega)).$$

where the function classes $\hat{\boldsymbol{S}}_{r,k}(\Omega)$ $(1 \leq k \leq 2)$ are defined as:

$$\hat{\boldsymbol{S}}_{r,1}(\Omega) := \big\{\hat{h}_1(x) := \frac{h_1(x)}{\mathbb{E}[h_1] + \mathbb{E}[h_2] + r} \mid (h_1, h_2) \in \boldsymbol{S}(\Omega)\big\},$$

$$\hat{\boldsymbol{S}}_{r,2}(\Omega) := \big\{\hat{h}_2(x) := \frac{h_2(x)}{\mathbb{E}[h_1] + \mathbb{E}[h_2] + r} \mid (h_1, h_2) \in \boldsymbol{S}(\Omega)\big\}.$$

Applying the modified Peeling Lemma B.5 to any function $h = (h_1, h_2) \in \mathbf{S}(\Omega)$ helps us upper bound the sum of the two local Rademacher complexities $R_N(\hat{\mathbf{S}}_{r,1}(\Omega)) + R_n(\hat{\mathbf{S}}_{r,2}(\Omega))$ with the function $\phi$ defined in equation C.4:

$$R_N(\hat{\mathbf{S}}_{r,1}(\Omega)) + R_n(\hat{\mathbf{S}}_{r,2}(\Omega)) = \mathbb{E}_\sigma \left[ \mathbb{E}_y \left[ \sup_{h \in \mathbf{S}(\Omega)} \frac{\frac{1}{N} \sum_{i=1}^N \sigma_i h_1(y_i)}{\mathbb{E}[h_1] + \mathbb{E}[h_2] + r} \right] \right] + \mathbb{E}_\tau \left[ \mathbb{E}_{y'} \left[ \sup_{h \in \mathbf{S}(\Omega)} \frac{\frac{1}{n} \sum_{j=1}^n \tau_j h_2(y'_j)}{\mathbb{E}[h_1] + \mathbb{E}[h_2] + r} \right] \right]$$

$$= \mathbb{E}_\sigma \left[ \mathbb{E}_{y,y'} \left[ \sup_{h \in \mathbf{S}(\Omega)} \frac{\frac{1}{N} \sum_{i=1}^N \sigma_i h_1(y_i)}{\mathbb{E}[h_1] + \mathbb{E}[h_2] + r} + \sup_{h \in \mathbf{S}(\Omega)} \frac{\frac{1}{n} \sum_{j=1}^n \tau_j h_2(y'_j)}{\mathbb{E}[h_1] + \mathbb{E}[h_2] + r} \right] \right]$$

$$= R_{N,n}(\hat{\mathbf{S}}_r(\Omega)) \le \frac{4\phi(r)}{r}.$$

Combining all inequalities derived above yields:

$$\sup_{\tilde{h}_1 \in \tilde{S}_{r,1}(\Omega)} \mathbb{E}_{y'} \left[ \frac{1}{N} \sum_{i=1}^N \tilde{h}_1(y'_i) \right] + \sup_{\tilde{h}_2 \in \tilde{S}_{r,2}(\Omega)} \mathbb{E}_y \left[ \frac{1}{n} \sum_{j=1}^n \tilde{h}_2(y_j) \right] \tag{C.6}$$

$$\le 2R_N(\hat{\mathbf{S}}_{r,1}(\Omega)) + 2R_n(\hat{\mathbf{S}}_{r,2}(\Omega)) = 2R_{N,n}(\hat{\mathbf{S}}_r(\Omega)) \le \frac{8\phi(r)}{r} \ (r > 0).$$

Secondly we'll apply the Talagrand concentration inequality to the two function classes $\tilde{S}_{r,1}(\Omega)$ and $\tilde{S}_{r,2}(\Omega)$, which requires us to verify the conditions needed. We will first check that the expectation sum $\mathbb{E}[h_1] + \mathbb{E}[h_2]$ is always non-negative for any $(h_1, h_2) \in \mathbf{S}(\Omega)$:

$$\mathbb{E}[h_1] + \mathbb{E}[h_2] = \frac{1}{|\Omega|} \int_\Omega |\Omega| \cdot \left( \frac{1}{2} \|\nabla u(x)\|^2 + \frac{1}{2} V(x)|u(x)|^2 - f(x)u(x) \right) dx$$

$$- \frac{1}{|\Omega|} \int_\Omega |\Omega| \cdot \left( \frac{1}{2} \|\nabla u^\star(x)\|^2 + \frac{1}{2} V(x)|u^\star(x)|^2 - f(x)u^\star(x) \right) dx$$

$$= \mathbf{E}(u) - \mathbf{E}(u^\star) \ge 0 \Rightarrow \mathbb{E}[h_1] + \mathbb{E}[h_2] \ge 0.$$

Next, We will verify that $\tilde{S}_{r,1}(\Omega)$ satisfies all three requirements. At first, we will show that any $\tilde{h}_1 = \frac{\mathbb{E}[h_1] - h_1}{\mathbb{E}[h_1] + \mathbb{E}[h_2] + r} \in \tilde{\mathbf{S}}_{r,1}(\Omega)$ is of bounded inf-norm. We need to prove that any $h_1 \in \mathbf{S}_1(\Omega)$ is of bounded inf-norm beforehand. Using boundedness condition listed in equation C.3 implies:

$$\|h_1\|_\infty = \| \frac{1}{2} \left( \|\nabla u\|^2 - \|\nabla u^*\|^2 \right) \|_\infty \le \frac{1}{2} \left( \|\nabla u\|_\infty^2 + \|\nabla u^*\|_\infty^2 \right) \le C^2.$$

By taking $M_1 := C^2$, we then have $\|h_1\|_\infty \le M_1$ for all $h_1 \in \mathbf{S}_1(\Omega)$. Note that the denominator of $\tilde{h}_1$ can be lower bounded by $|\mathbb{E}[h_1] + \mathbb{E}[h_2] + r| \ge r > 0$. Combining these two inequalities help us upper bound the inf-norm $\|\tilde{h}_1\|_\infty = \sup_{x \in \Omega} |\tilde{h}_1(x)|$ as follows:

$$\|\tilde{h}_1\|_\infty = \frac{\|\mathbb{E}[h_1] - h_1\|_\infty}{|\mathbb{E}[h_1] + \mathbb{E}[h_2] + r|} \le \frac{2\|h_1\|_\infty}{r} \le \frac{2M_1}{r} =: \beta_1.$$

Also, it is easy to check that for any $\tilde{h}_1 \in \tilde{\mathbf{S}}_{r,1}(\Omega)$, we have

$$\mathbb{E}[\tilde{h}_1] = \frac{\mathbb{E}[h_1] - \mathbb{E}[h_1]}{\mathbb{E}[h_1] + \mathbb{E}[h_2] + r} = 0,$$

*i.e.* any function in the localized class $\tilde{\mathbf{S}}_{r,1}(\Omega)$ is of zero mean.

Moreover, we take $\sigma_1^2 = \sup_{\tilde{h}_1 \in \tilde{\mathbf{S}}_{r,1}(\Omega)} \mathbb{E}[\tilde{h}_1^2]$ to be the upper bound on the second moment of functions in $\tilde{\mathbf{S}}_{r,1}(\Omega)$. Now we have verified that any function $\tilde{h}_1 \in \tilde{\mathbf{S}}_{r,1}(\Omega)$ satisfies all the required conditions. By taking $\mu$ to be the uniform distribution on the domain $\Omega$ and applying Talagrand's Concentration inequality given in Lemma B.3, we have:

$$\mathbb{P}_x \left[ \sup_{\tilde{h}_1 \in \tilde{\mathbf{S}}_{r,1}(\Omega)} \frac{1}{N} \sum_{i=1}^N \tilde{h}_1(x_i) \ge 2 \sup_{\tilde{h}_1 \in \tilde{\mathbf{S}}_{r,1}(\Omega)} \mathbb{E}_y \left[ \frac{1}{N} \sum_{i=1}^N \tilde{h}_1(y_i) \right] + \sqrt{\frac{2t\sigma_1^2}{N}} + \frac{2t\beta_1}{N} \right] \le e^{-t}. \tag{C.7}$$

Moreover, We will verify that $\tilde{S}_{r,2}(\Omega)$ also satisfies all three requirements. At first, we will show that any $\tilde{h}_2 = \frac{\mathbb{E}[h_2] - h_2}{\mathbb{E}[h_1] + \mathbb{E}[h_2] + r} \in \tilde{S}_{r,2}(\Omega)$ is of bounded inf-norm. We need to prove that any $h_2 \in S_2(\Omega)$ is of bounded inf-norm beforehand. Using boundedness condition listed in equation C.3 implies:

$$
\begin{aligned}
\|h_2\|_\infty &= \|\frac{1}{2}V(|u|^2 - |u^*|^2) - f(u - u^*)\|_\infty \\
&\le \frac{1}{2}V_{\max}\left(\|u\|_\infty^2 + \|u^*\|_\infty^2\right) + \|f\|_\infty\left(\|u\|_\infty + \|u^*\|_\infty\right) \\
&\le \frac{1}{2}V_{\max} \times 2C^2 + 2C^2 = (V_{\max} + 2)C^2.
\end{aligned}
$$

By taking $M_2 := (V_{\max} + 2)C^2$, we then have $\|h_2\|_\infty \le M_2$ for all $h_2 \in S_2(\Omega)$. Note that the denominator of $\tilde{h}_2$ can be lower bounded by $|\mathbb{E}[h_1] + \mathbb{E}[h_2] + r| \ge r > 0$. Combining these two inequalities help us upper bound the inf-norm $\|\tilde{h}_2\|_\infty = \sup_{x \in \Omega} |\tilde{h}_2(x)|$ as follows:

$$
\|\tilde{h}_2\|_\infty = \frac{\|\mathbb{E}[h_2] - h_2\|_\infty}{|\mathbb{E}[h_1] + \mathbb{E}[h_2] + r|} \le \frac{2\|h_2\|_\infty}{r} \le \frac{2M_2}{r} =: \beta_2.
$$

Also, it is easy to check that for any $\tilde{h}_2 \in \tilde{S}_{r,2}(\Omega)$, we have

$$
\mathbb{E}[\tilde{h}_2] = \frac{\mathbb{E}[h_2] - \mathbb{E}[h_2]}{\mathbb{E}[h_1] + \mathbb{E}[h_2] + r} = 0,
$$

*i.e.* any function in the localized class $\tilde{S}_{r,2}(\Omega)$ is of zero mean.

Moreover, we take $\sigma_2^2 = \sup_{\tilde{h}_2 \in \tilde{S}_{r,2}(\Omega)} \mathbb{E}[\tilde{h}_2^2]$ to be the upper bound on the second moment of functions in $\tilde{S}_{r,2}(\Omega)$. Now we have verified that any function $\tilde{h}_2 \in \tilde{S}_{r,2}(\Omega)$ satisfies all the required conditions. By taking $\mu$ to be the uniform distribution on the domain $\Omega$ and applying Talagrand's Concentration inequality given in Lemma B.3, we have:

$$
\mathbb{P}_{x'}\left[\sup_{\tilde{h}_2 \in \tilde{S}_{r,2}(\Omega)} \frac{1}{n}\sum_{j=1}^{n}\tilde{h}_2(x_j') \ge 2\sup_{\tilde{h}_2 \in \tilde{S}_{r,2}(\Omega)}\mathbb{E}_{y'}\left[\frac{1}{n}\sum_{j=1}^{n}\tilde{h}_2(y_j')\right] + \sqrt{\frac{2t\sigma_2^2}{n}} + \frac{2t\beta_2}{n}\right] \le e^{-t}.
$$
(C.8)

By applying a union bound to the two inequalities derived in C.7 and C.8, we can derive that with probability at least $1 - 2e^{-t}$, the inequality below holds:

$$
\begin{aligned}
\frac{1}{N}\sum_{i=1}^{N}\tilde{h}_1(x_i') + \frac{1}{n}\sum_{j=1}^{n}\tilde{h}(x_j) &\le \sup_{\tilde{h}_1 \in \tilde{S}_{r,1}(\Omega)}\frac{1}{N}\sum_{i=1}^{N}\tilde{h}_1(x_i) + \sup_{\tilde{h}_2 \in \tilde{S}_{r,2}(\Omega)}\frac{1}{n}\sum_{j=1}^{n}\tilde{h}_2(x_j') \\
&\le 2\sup_{\tilde{h}_1 \in \tilde{S}_{r,1}(\Omega)}\mathbb{E}_y\left[\frac{1}{N}\sum_{i=1}^{N}\tilde{h}_1(y_i)\right] + \sqrt{\frac{2t\sigma_1^2}{N}} + \frac{2t\beta_1}{N} \\
&\quad + 2\sup_{\tilde{h}_2 \in \tilde{S}_{r,2}(\Omega)}\mathbb{E}_{y'}\left[\frac{1}{n}\sum_{j=1}^{n}\tilde{h}_2(y_j')\right] + \sqrt{\frac{2t\sigma_2^2}{n}} + \frac{2t\beta_2}{n} \\
&\le \frac{16\phi(r)}{r} + \sqrt{\frac{2t}{n}}(\sigma_1 + \sigma_2) + \frac{2t(\beta_1 + \beta_2)}{n}.
\end{aligned}
$$

By the definition of $\beta_1$ and $\beta_2$, we have that the term $\frac{2t(\beta_1 + \beta_2)}{n}$ can be upper bounded by:

$$
\frac{2t(\beta_1 + \beta_2)}{n} = \frac{4t(M_1 + M_2)}{nr} \le \frac{4(V_{\max} + 3)C^2 t}{nr}.
$$

Now we will derive some upper bound on the sum $\sigma_1 + \sigma_2$. By definition we have that:

$$
\begin{aligned}
(\sigma_1 + \sigma_2)^2 &\le 2(\sigma_1^2 + \sigma_2^2) = 2\left[\sup_{\tilde{h}_1 \in \tilde{S}_{r,1}(\Omega)}\mathbb{E}[\tilde{h}_1^2] + \sup_{\tilde{h}_2 \in \tilde{S}_{r,2}(\Omega)}\mathbb{E}[\tilde{h}_2^2]\right] \\
&= 2\left[\sup_{h \in S(\Omega)}\frac{\mathbb{E}[h_1^2] - \mathbb{E}[h_1]^2}{|\mathbb{E}[h_1] + \mathbb{E}[h_2] + r|^2} + \sup_{h \in S(\Omega)}\frac{\mathbb{E}[h_2^2] - \mathbb{E}[h_2]^2}{|\mathbb{E}[h_1] + \mathbb{E}[h_2] + r|^2}\right] \\
&\le 4\sup_{h \in S(\Omega)}\frac{\mathbb{E}[h_1^2] + \mathbb{E}[h_2^2]}{|\mathbb{E}[h_1] + \mathbb{E}[h_2] + r|^2}.
\end{aligned}
$$

Now it suffices to derive an upper bound of $\frac{\mathbb{E}[h_1^2]+\mathbb{E}[h_2^2]}{|\mathbb{E}[h_1]+\mathbb{E}[h_2]+r|^2}$ for any $h \in \boldsymbol{S}(\Omega)$. The existence of such an upper bound is guaranteed because of the regularity results of the PDE. We aim to show that there exist some constants $\alpha, \alpha' > 0$, such that for any $h \in \boldsymbol{S}(\Omega)$, the following inequality holds:

$$\alpha(\mathbb{E}[h_1^2] + \mathbb{E}[h_2^2]) \leq \|u - u^*\|_{H^1(\Omega)}^2 \leq \alpha'(\mathbb{E}[h_1] + \mathbb{E}[h_2]). \tag{C.9}$$

The RHS of the inequality follows from strong convexity of the DRM objective function proved in Theorem B.1:

$$\mathbb{E}[h_1] + \mathbb{E}[h_2] = \boldsymbol{E}(u) - \boldsymbol{E}(u^*) \geq \frac{\min\{1, V_{\min}\}}{4}\|u - u^*\|_{H^1(\Omega)}^2.$$

The LHS of the inequality follows from boundedness condition listed in equation C.3 and the QM-AM inequality:

$$\begin{aligned}
\mathbb{E}[h_1^2] + \mathbb{E}[h_2^2] &= \int_\Omega \frac{1}{4}\Big(\|\nabla u\|^2 - \|\nabla u^*\|^2\Big)^2 dx + \int_\Omega \left[\frac{1}{2}V(|u|^2 - |u^*|^2) - f(u - u^*)\right]^2 dx \\
&\leq \frac{1}{4}\int_\Omega \Big(\|\nabla u\|^2 - \|\nabla u^*\|^2\Big)^2 dx + \frac{1}{2}\int_\Omega V^2(|u|^2 - |u^*|^2)^2 dx + 2\int_\Omega f^2(u - u^*)^2 dx \\
&\leq \frac{1}{4}\int_\Omega \Big|\|\nabla u\| - \|\nabla u^*\|\Big|^2 (\|\nabla u\| + \|\nabla u^*\|)^2 dx + \frac{1}{2}V_{\max}^2 \int_\Omega \Big||u| - |u^*|\Big|^2 (|u| + |u^*|)^2 dx \\
&\quad + 2C^2 \int_\Omega (u - u^*)^2 dx \leq C^2 \int_\Omega \|\nabla u - \nabla u^*\|^2 dx + 2C^2(1 + V_{\max}^2)\int_\Omega |u - u^*|^2 dx \\
&\leq 2C^2(1 + V_{\max}^2)\|u - u^*\|_{H^1(\Omega)}^2.
\end{aligned}$$

By picking $\alpha' = \frac{4}{\min\{1, V_{\min}\}}$ and $\alpha = \frac{1}{2C^2(1 + V_{\max}^2)}$, we have finished proving inequality C.9. Then we can can upper bound the term $\frac{\mathbb{E}[h_1^2]+\mathbb{E}[h_2^2]}{|\mathbb{E}[h_1]+\mathbb{E}[h_2]+r|^2}$ as:

$$\frac{\mathbb{E}[h_1^2] + \mathbb{E}[h_2^2]}{|\mathbb{E}[h_1] + \mathbb{E}[h_2] + r|^2} \leq \frac{\frac{\alpha'}{\alpha}\Big(\mathbb{E}[h_1] + \mathbb{E}[h_2]\Big)}{2r\Big(\mathbb{E}[h_1] + \mathbb{E}[h_2]\Big)} \leq \frac{\alpha'}{2\alpha r}.$$

Combining the bounds derived above helps us upper bound the term $\sqrt{\frac{2t}{n}}(\sigma_1 + \sigma_2)$ as below:

$$\sqrt{\frac{2t}{n}}(\sigma_1 + \sigma_2) \leq \sqrt{\frac{8t}{n}}\sqrt{\sup_{h \in \boldsymbol{S}(\Omega)}\frac{\mathbb{E}[h_1^2] + \mathbb{E}[h_2^2]}{|\mathbb{E}[h_1] + \mathbb{E}[h_2] + r|^2}} \leq \sqrt{\frac{4\alpha't}{n\alpha r}}.$$

Thus, using the two upper bounds on $\sqrt{\frac{2t}{n}}(\sigma_1 + \sigma_2)$ and $\frac{2t(\beta_1+\beta_2)}{n}$, we have

$$\begin{aligned}
\frac{1}{N}\sum_{i=1}^N \tilde{h}_1(x_i') + \frac{1}{n}\sum_{j=1}^n \tilde{h}(x_j) &\leq \frac{16\phi(r)}{r} + \sqrt{\frac{2t}{n}}(\sigma_1 + \sigma_2) + \frac{2t(\beta_1 + \beta_2)}{n} \\
&\leq \frac{16\phi(r)}{r} + \sqrt{\frac{4\alpha't}{n\alpha r}} + \frac{4(V_{\max} + 3)C^2 t}{nr} = \psi(r).
\end{aligned}$$

Let's pick the critical radius $r_0$ to be:

$$r_0 = \max\{2^{14}r^*, \frac{24Mt}{n}, \frac{144\alpha't}{\alpha n}\}. \tag{C.10}$$

Note that concavity of the function $\phi$ implies that $\phi(r) \leq r$ for any $r \geq r^*$. Combining this with the first inequality listed in C.4 yields:

$$\frac{16\phi(r_0)}{r_0} \leq \frac{2^{11}\phi(\frac{r_0}{2^{14}})}{2^{14}\frac{r_0}{2^{14}}} = \frac{1}{8} \times \frac{\phi(\frac{r_0}{2^{14}})}{\frac{r_0}{2^{14}}} \leq \frac{1}{8}.$$

On the other hand, applying equation C.10 yields:

$$\sqrt{\frac{4\alpha't}{n\alpha r_0}} \le \sqrt{\frac{4\alpha't}{n\alpha} \frac{\alpha n}{144\alpha't}} = \frac{1}{6},$$

$$\frac{4(V_{\max}+3)C^2 t}{nr_0} \le \frac{4(V_{\max}+3)C^2 t}{n} \times \frac{n}{24(V_{\max}+3)C^2 t} = \frac{1}{6}.$$

Summing the three inequalities above implies:

$$\psi(r_0) = \frac{16\phi(r_0)}{r_0} + \sqrt{\frac{4\alpha't}{n\alpha r_0}} + \frac{4(V_{\max}+3)C^2 t}{nr_0} \le \frac{1}{8} + \frac{1}{6} + \frac{1}{6} < \frac{1}{2}.$$

By picking $r = r_0$, we can further deduce that for any function $u \in \boldsymbol{F}(\Omega)$, the following inequality holds with probability $1 - 2e^{-t}$:

$$\frac{\boldsymbol{E}(u) - \boldsymbol{E}(u^*) + \boldsymbol{E}_{N,n}(u^*) - \boldsymbol{E}_{N,n}(u)}{\boldsymbol{E}(u) - \boldsymbol{E}(u^*) + r_0} = \frac{1}{n} \sum_{i=1}^{n} \tilde{h}(x_i) \le \psi(r_0) < \frac{1}{2}.$$

Multiplying the denominator on both sides indicates:

$$\Delta \boldsymbol{E}_{\text{gen}} = \boldsymbol{E}(u) - \boldsymbol{E}(u^*) + \boldsymbol{E}_{N,n}(u^*) - \boldsymbol{E}_{N,n}(u) \le \frac{1}{2}\Big[\boldsymbol{E}(u) - \boldsymbol{E}(u^*)\Big] + \frac{1}{2}r_0 = \frac{1}{2}\Delta \boldsymbol{E}^{(n)} + \frac{1}{2}r_0.$$

Substituting the upper bound above into the decomposition $\Delta \boldsymbol{E}^{(n)} \le \Delta \boldsymbol{E}_{\text{gen}} + 2\Delta \boldsymbol{E}_{\text{app}} + \frac{4t}{n}$ yields that with probability $1 - 4e^{-\min\{1, C_q\}t}$, we have:

$$\Delta \boldsymbol{E}^{(n)} \le \Delta \boldsymbol{E}_{\text{gen}} + \frac{3}{2}\Delta \boldsymbol{E}_{\text{app}} + \frac{t}{2n} \le \frac{1}{2}\Delta \boldsymbol{E}^{(n)} + \frac{1}{2}r_0 + 2\Delta \boldsymbol{E}_{\text{app}} + \frac{4t}{n}.$$

Simplifying the inequality above yields that with probability $1 - 4e^{-\min\{1, C_q\}t}$, we have:

$$\Delta \boldsymbol{E}^{(n)} \le r_0 + 4\Delta \boldsymbol{E}_{\text{app}} + \frac{8t}{n} = 3 \inf_{u_{\boldsymbol{F}} \in \boldsymbol{F}(\Omega)} \Big(\boldsymbol{E}(u_{\boldsymbol{F}}) - \boldsymbol{E}(u^\star)\Big) + \max\{2^{14}r^*, 24M\frac{t}{n}, \frac{36\alpha'}{\alpha}\frac{t}{n}\} + \frac{8t}{n}$$

$$\lesssim \inf_{u_{\boldsymbol{F}} \in \boldsymbol{F}(\Omega)} \Big(\boldsymbol{E}(u_{\boldsymbol{F}}) - \boldsymbol{E}(u^\star)\Big) + \max\Big\{r^*, \frac{t}{n}\Big\}.$$

Moreover, using strong convexity of the DRM objective function proved in Theorem B.1 implies:

$$\Delta \boldsymbol{E}^{(n)} = \boldsymbol{E}(\hat{u}_{\text{MDRM}}) - \boldsymbol{E}(u^*) \gtrsim \|\hat{u}_{\text{MDRM}} - u^*\|_{H^1(\Omega)}^2.$$

Combining the two bounds above yields that with probability $1 - 4e^{-\min\{1, C_q\}t}$, we have:

$$\|\hat{u}_{\text{MDRM}} - u^*\|_{H^1(\Omega)}^2 \lesssim \inf_{u_{\boldsymbol{F}} \in \boldsymbol{F}(\Omega)} \Big(\boldsymbol{E}(u_{\boldsymbol{F}}) - \boldsymbol{E}(u^\star)\Big) + \max\Big\{r^*, \frac{t}{n}\Big\}.$$

$\square$

**Truncated Fourier Series Estimator.** Next we aim to show that the truncated Fourier series estimator can achieve the min-max optimal rate using the MDRM objective function. For any $\xi \in \mathbb{Z}^+$ satisfying $\xi^2 < \frac{N}{n}$, there exists some Truncated Fourier Series in $\boldsymbol{F}_\xi(\Omega)$ with approximation error $\Delta \boldsymbol{E}_{\text{app}} = O(\xi^{-2(s-1)})$ and generalization error $\Delta \boldsymbol{E}_{\text{gen}} = O(\frac{\xi^{d-2}}{n})$. With optimal selection $\xi = n^{\frac{1}{d+2s-4}}$ to balance the bias and variance, we can achieve $n^{-\frac{2s-2}{d+2s-4}}$ convergence rate for the Fourier estimator.

**Theorem C.2.** *(Final Upper Bound of MDRM with Truncated Fourier Series Estimator) Under the assumptions in Theorem C.1, we consider the Modified Deep Ritz objective with the Truncated Fourier Series function space $\boldsymbol{F}_\xi(\Omega)$, where the parameter $\xi = \Theta(n^{\frac{1}{d+2s-2}})$. By assuming $\frac{\xi^d}{N} < \frac{\xi^{d-2}}{n}$, we have that the Fourier estimator $\hat{u}_{MDRM}^{Fourier} = \min_{u \in \boldsymbol{F}_\xi(\Omega)} \boldsymbol{E}_{N,n}^{MDRM}(u)$ satisfies the following upper bound with high probability:*

$$\|\hat{u}_{MDRM}^{Fourier} - u^*\|_{H^1}^2 \lesssim n^{-\frac{2s-2}{d+2s-4}}$$

*Proof.* Following the same proof as shown for the DRM upper bound in Theorem B.11, we firstly need to determine the critical radius $r^*$:

$$\sqrt{\frac{r^*}{n}}\xi^{\frac{d-2}{2}} + \sqrt{\frac{r^*}{N}}\xi^{\frac{d}{2}} + \frac{1}{n} + \frac{1}{N} + \xi^{-2(s-1)} \simeq r^*.$$

For we have assumed $\frac{\xi^d}{N} < \frac{\xi^{d-2}}{n}$, the solution of the fixed point equation is $r^* \simeq \frac{\xi^{d-2}}{n} + \frac{1}{n} + \xi^{-2(s-1)}$. On the other hand, by taking $\alpha = s$ and $\beta = 1$ in Lemma B.19 and applying strong convexity of the DRM objective function proved in Theorem B.1, we can upper bound the approximation error $\Delta\boldsymbol{E}_{\text{app}}$ as below:

$$\Delta\boldsymbol{E}_{\text{app}} \lesssim \xi^{-2(s-1)}.$$

Combining the two bounds above with Theorem C.1 yields that with high probability, we have:

$$\|\hat{u}_{\text{MDRM}}^{\text{Fourier}} - u^*\|_{H^1}^2 \lesssim \Delta\boldsymbol{E}_{\text{app}} + r^* \lesssim \frac{\xi^{d-2}}{n} + \frac{1}{n} + \xi^{-2(s-1)}.$$

By equating the two of the three terms above, we can solve for $\xi$ that yields the desired bound:

$$\frac{\xi^{d-2}}{n} \simeq \xi^{-2(s-1)} \Rightarrow \xi \simeq n^{\frac{1}{d+2s-4}}.$$

Plugging in the expression of $\xi$ gives the final upper bound:

$$\|\hat{u}_{\text{MDRM}}^{\text{Fourier}} - u^*\|_{H^1}^2 \lesssim \frac{\xi^{d-2}}{n} + \xi^{-2(s-2)} + \frac{1}{n} \lesssim n^{-\frac{2s-2}{d+2s-4}}.$$

$\square$

# D PROOF OF THE LOWER BOUNDS

## D.1 PRELIMINARIES ON TOOLS FOR LOWER BOUNDS

In this section, we list the standard tools we use to establish the lower bound. The main tool we use is the Fano's inequailty and the Varshamov-Gilber Lemma.

**Lemma D.1** (Fano's methods). *Assume that $V$ is a unifrom random variable over set $\mathcal{V}$, then for any markov chain $V \to X \to \hat{V}$, we always have:*

$$\mathcal{P}(\hat{V} \neq V) \geq 1 - \frac{I(V;X) + \log 2}{\log(|\mathcal{V}|)}.$$

**Lemma D.2** (Varshamov-Gillbert Lemma,(Tsybakov, 2008) Theorem 2.9). *Let $D \geq 8$. There exists a subset $\mathcal{V} = \{\tau^{(0)}, \cdots, \tau^{(2^{D/8})}\}$ of $D-$dimensional hypercube $\mathcal{H}^D = \{0,1\}^D$ such that $\tau^{(0)} = (0, 0, \cdots, 0)$ and the $\ell_1$ distance between every two elements is larger than $\frac{D}{8}$:*

$$\sum_{l=1}^{D} \|\tau^{(j)} - \tau^{(k)}\|_{\ell_1} \geq \frac{D}{8}, \text{for all } 0 \leq j, k \leq 2^{D/8}.$$

## D.2 PROOF OF LOWER BOUND

In this section, we provide the proof of the lower bound for learning a PDE. Our proof uses standard Fano method to establish minimax lower bound but finally leads to a non-standard convergence rate. We state standard results for Fano methods in Appendix D.1. Following is the proof our main lower bound.

**Theorem D.1** (Lower bound). *We denote $u^*(f)$ to be the solution of the PDE 2.1 and we can access randomly sampled data $\{X_i, Y_i\}_{i=1,\cdots,n}$ as described in Section 2.2.*

**DRM Lower Bound.** *For all estimators $\psi : (\mathbb{R}^d)^{\otimes n} \times \mathbb{R}^{\otimes n} \to H^s(\Omega)$, we have*

$$\inf_{\psi} \sup_{u^* \in H^s(\Omega)} \mathbb{E}\|\psi(\{X_i, f_i\}_{i=1,\cdots,n}) - u^*(f)\|_{H^1}^2 \gtrsim n^{-\frac{2s-2}{d+2s-4}}. \tag{D.1}$$

**PINN Lower Bound.** *For all estimators $\psi : \left(\mathbb{R}^d\right)^{\otimes n} \times \mathbb{R}^{\otimes n} \to H^s(\Omega)$, we have*

$$\inf_{\psi} \sup_{u^* \in H^s(\Omega)} \mathbb{E}\|\psi(\{X_i, f_i\}_{i=1,\cdots,n}) - u^*(f)\|_{H^2}^2 \gtrsim n^{-\frac{2s-4}{d+2s-4}}. \tag{D.2}$$

*Proof.* We construct the following bump function to construct the multiple hypothesis test used for proving the lower bound. Consider a simple $C^\infty$ bump function supported on $[0,1]^d$

$$g(x) = \prod_{i=1}^d \xi(x_i), x = (x_1, \cdots, x_d),$$

where $\xi : \mathbb{R} \to \mathbb{R}$ is a non-zero funtion in $C^\infty(\mathbb{R})$ with support contained in $[0,1]$ and satisfies $\xi(x) \neq 0, \frac{d}{dx}\xi(x) \neq 0$. Then $\nabla g(x) \neq 0$ and the support of function $g$ is $[0,1]^d$.

Next, we take $m = [n^{\frac{1}{2s-4+d}}]$ and consider a regular gird $x^{(j)}, j \in [m]^d$. According to the Varshamov-Gilbert lemma, there exist $2^{m^d/8}$ $(0,1)$-sequences $\tau^{(1)}, \cdots, \tau^{(2^{m^d/8})} \in \{0,1\}^{m^d}$ such that $\|\tau^{(k)} - \tau^{(k')}\|^2 \geq \frac{m^d}{8}$ for all $0 < k \neq k' \leq 2^{m^d/8}$. Then we construct the multiple hypothesis as

$$u_k(x) = \sum_{j \in [m]^d} \tau_j^{(k)} \frac{\omega}{m^{s+\frac{d}{2}}} g(m(x - x^{(j)})), k = 1, 2, \cdots, 2^{m^d/8},$$

where $\omega$ is a constant to be determined later. It is easy to find out that $u_k \in C^s$.

Then we reduce solving the PDE to a multiple hypothesis testing problem, which considers all mappings from $n$ sampled data to the constructed hypothesis $\Psi : \left(\mathbb{R}^d\right)^{\otimes n} \times \mathbb{R}^{\otimes n} \to \mathcal{V} := \{u_i | i = 1, 2, \cdots, 2^{m^d/8}\}$. Then we apply the local Fano method and check that we can obtain a constant lower bound of $\mathcal{P}(\hat{V} \neq V)$ for any estimator $\hat{V}$. From the local Fano method, we know that

$$I(V; X) \leq \frac{1}{|\mathcal{V}|^2} \sum_z \sum_{v \neq v'} D_{KL}(P_v || P_v'),$$

where $P_k$ denotes the joint distribution of the sampled data $(x, y)$. In specific, $x$ follows a uniform distribution on $[0,1]^d$ and $y = f(X) + \epsilon$, where $\epsilon$ is independently sampled from a standard Gaussian distribution $N(0,1)$. Then we have

$$KL(P_k || P_{k'}) = \mathbb{E}\log(\frac{dP_k}{dP_{k'}}) = \|\Delta u_k + V u_k\|_{L_2}^2 \leq \frac{C\omega}{m^{2s-4}}.$$

Using Fano inequality, if we select $m \propto [n^{\frac{1}{2s-4+d}}]$ then we have the following lower bound when $\omega$ is taken to be sufficiently large:

$$\mathcal{P}(\hat{V} \neq V) \geq 1 - \frac{I(V; X) + \log 2}{\log(|\mathcal{V}|)} \geq 1 - \frac{\frac{8C\omega}{m^{2s-4}}}{m^d \log 2} \geq 1/2.$$

At the same time, we can estimate the separation of the hypotheses in two different norms:

- Deep Ritz Method:

$$\int_{[0,1]^d} \|\nabla u_k - \nabla u_{k'}\|^2 dx = \frac{\kappa^2}{m^{2s-2+d}} \sum_{j \in [m]^d} \|\tau_j^{(k)} - \tau_j^{(k')}\|_1 \int_{\mathbb{R}^d} \|\nabla g(x)\|^2 dx \gtrsim \frac{1}{m^{2s-2}}.$$

- Physic Informed Neural Network:

$$\int_{[0,1]^d} \|\Delta u_k - \Delta u_{k'}\|^2 dx = \frac{\kappa^2}{m^{2s-4+d}} \sum_{j \in [m]^d} \|\tau_j^{(k)} - \tau_j^{(k')}\|_1 \int_{\mathbb{R}^d} \Delta g(x)^2 dx \gtrsim \frac{1}{m^{2s-4}}.$$

Plugging in $m \propto [n^{\frac{1}{2s-4+d}}]$, we know that with constant probability we have

$$\inf_{\psi} \sup_{u^*\in H^s(\Omega)} \mathbb{E}\|\psi(\{X_i, Y_i\}_{i=1,\cdots,n}) - u^*(f)\|_{H^1}^2 \gtrsim n^{-\frac{2s-2}{d+2s-4}}, \tag{D.3}$$

$$\inf_{\psi} \sup_{u^*\in H^s(\Omega)} \mathbb{E}\|\psi(\{X_i, Y_i\}_{i=1,\cdots,n}) - u^*(f)\|_{H^2}^2 \gtrsim n^{-\frac{2s-4}{d+2s-4}}. \tag{D.4}$$

$\square$

# E  INTUITION BEHIND THE SUB-OPTIMALITY OF THE UNMODIFIED DEEP RITZ METHODS

In this section, we aim to discuss the intuition behind the sub-optimality of the unmodified DRM via using the truncation Fourier basis. To simplify the notation, in this section we consider the following simplest Poisson equation $\Delta u = f$ on the hypercube with zero Dirichlet boundary condition. To illustrate the necessity of the modification we made, we consider the difference between the following two estimators

- **Estimator 1.** We use the truncated Fourier basis estimator to learn the right hand side function $f$ and then we invert the PDE exactly to get the estimated $u$.

- **Estimator 2.** We plug in a parametrization of the truncated fourier basis into the empirical DRM objective

We would like to point out that *estimator 1 isn't build for computational consideration*. Instead, we use it to consider the statistical limit of our sampled data. We first show that the estimator 1 can achieve the minimax optimal estimation error.

**Error Of Estimator 1**  Firstly, we show that if one wants to learn the function $u$ in $H_0^1$ norm, one need to learn the right hand side function $f$ in $H_0^{-1}$ norm. The $H_0^{-1}$ norm is defined as the dual norm of the $H^1$ norm, *i.e.* $\|u\|_{H_0^{-1}} = \max_{\|v\|_{H_0^1}\leq 1} \langle u, v\rangle$. Once we assume we have an estimate $\hat{f}$ of $f$ in $H_0^{-1}$, we can have an estimate of $u$ via $\hat{u} := (\Delta)^{-1}\hat{f}$, whose distance to $u$ in the $H^1$ norm satisfies:

$$\|\nabla u - \nabla\hat{u}\|_{H_0^1} = \max_{\|v\|_{H_0^1}\leq 1} \langle \nabla u - \nabla\hat{u}, \nabla v\rangle$$

$$= \max_{\|v\|_{H_0^1}\leq 1} \langle \Delta u - \Delta\hat{u}, v\rangle$$

$$= \max_{\|v\|_{H_0^1}\leq 1} \langle f - \hat{f}, v\rangle = \|f - \hat{f}\|_{H_{-1}}.$$

Estimator 1 using the truncated Fourier estimator to estimate the right hand side function $f$. Suppose we can access a random sample of observed data as $\{x_i, f(x_i)\}_{i=1}^n$, then the Fourier coefficient $f_z := \langle u, \phi_z\rangle$ can be estimated as $\hat{f}_z := \frac{1}{n}\sum_{i=1}^n f(x_i)\phi_z(x_i)$. To bound the estimation error of $\hat{f} := \sum_{\|z\|_\infty\leq Z} \hat{f}_z\phi_z$ in $H_0^{-1}$, we first apply the bias-variance decomposition:

$$\mathbb{E}\|\hat{f} - f\|_{H_0^{-1}}^2 \leq \|\mathbb{E}\hat{f} - \hat{f}\|_{H_0^{-1}}^2 + \mathbb{E}\|f - \mathbb{E}\hat{f}\|_{H_0^{-1}}^2$$

We first bound the bias term $\|\mathbb{E}\hat{f} - f\|_{H^{-1}}^2$. Given $\mathbb{E}\hat{f} = \sum_{\|z\|_\infty\leq Z} f_z\phi_z$, we have that for a truncation set $Z$ of the from $\mathcal{Z} := \{z\in\mathbb{N}^d | \|z\|_\infty \leq Z\}$, the bias term can be controlled by:

$$\|\sum_{\|z\|_\infty>Z} f_z\phi_z\|_{H^{-1}}^2 \leq C \sum_{\|z\|_\infty>Z} f_z^2 z^{-2} \leq \|z\|^{-2(s-1)}\|f\|_{H_{\alpha-2}}^2$$

Next we estimate the variance of the estimator by decomposing the variance into the following sum:

$$\mathbb{E}\|f - \hat{f}\|^2_{H_{-1}} \leq \mathbb{E} \sum_{\|z\|_\infty \leq Z} (\hat{f}_z - f_z)^2 \|\phi_z\|^2_{H_{-1}} \leq \sum_{\|z\|_\infty \leq Z} |z|^{-1} \mathrm{Var}(\hat{f}_z).$$

Finally we achieve a $Z^{-2(s-1)} + \frac{Z^{d-2}}{n}$ upper bound for estimator 1. With optimal selection of $Z$, we can achieve the min-max optimal convergence rate $n^{-\frac{2s-2}{d+2s-4}}$.

**Difference Between Estimator 1 and Estimator 2**    Next we aim to understand the Deep Ritz Method objective function via plugging in a truncated Fourier series estimator. We consider an estimator of the form $u = \sum \hat{u}_z \phi_z(x)$, which lies in the space of truncated Fourier series. Then the empirical DRM objective function can be expressed as

$$\frac{1}{2n} \sum_{i=1}^n \left( \sum_z \hat{u}_z \nabla \phi_z(x_i) \right)^2 + \sum_z \hat{u}_z \phi_z(x_i) f(x_i). \tag{E.1}$$

We observe that (E.1) is a quadratic formula with respect to the Fourier coefficients $\boldsymbol{u} := (u_z)_{\|z\|_\infty \leq Z}$. Thus, we can rewrite it as the following matrix form

$$\min \frac{1}{2} \boldsymbol{u}^\top \hat{A} \boldsymbol{u} + \boldsymbol{u}^\top \hat{f}, \text{ where } \hat{A} = \left( \frac{1}{n} \sum_{i=1}^n \nabla \phi_i(x_i) \nabla \phi_j(x_i) \right)_{\|i\|_\infty \leq Z, \|j\|_\infty \leq Z}. \tag{E.2}$$

Based on the matrix formulation E.2, we can compare the solution given by the two estimators

- **Estimator 1:** The Fourier coefficients of the solution of Estimator 1 are
$$\hat{\boldsymbol{u}}_1 = \mathrm{diag}\left(\|z\|^2\right)^{-1}_{\|z\|_\infty \leq Z} \hat{f}. \tag{E.3}$$

- **Estimator 2:** The Fourier coefficients of the solution of Estimator 2 are
$$\hat{\boldsymbol{u}}_2 = \hat{A}^{-1} \hat{f}. \tag{E.4}$$

Note that $\mathbb{E}\hat{A} = \left(\|z\|^2\right)_{\|z\|_\infty \leq Z}$. Thus, we can further introduce another variance from the sampling of $A$. By directly estimating $\hat{\boldsymbol{u}}_1 - \hat{\boldsymbol{u}}_2$, we will show that this term will be larger than the final convergence rate. Notice that

$$\|\hat{\boldsymbol{u}}_1 - \hat{\boldsymbol{u}}_2\|^2_{H^1} = f^\top \left( (\mathbb{E}\hat{A})^{-1} - \hat{A}^{-1} \right)^\top \mathrm{diag}\left(\|z\|^2\right)_{\|z\|_\infty \leq Z} \left( (\mathbb{E}\hat{A})^{-1} - \hat{A}^{-1} \right) f \tag{E.5}$$

Next we aim to bound $\left( (\mathbb{E}\hat{A})^{-1} - \hat{A}^{-1} \right)$. We first use the Matrix Bernstein Inequality([Tropp, 2015](#)) to bound the $H^1$ distance between $\hat{\boldsymbol{u}}_1$ and $\hat{\boldsymbol{u}}_2$. According to the Matrix Bernstein Inequality, we have that with probability $1 - e^{-t}$, the following inequality holds

$$\left\| \left( \mathbb{E}\hat{A} \right) - \hat{A} \right\|_{\boldsymbol{H}} \leq \sqrt{\frac{Z^d}{n}} + \frac{t}{n}, \tag{E.6}$$

where $\| \cdot \|_{\boldsymbol{H}}$ is the matrix operator norm respect to the vector $\| \cdot \|_{\boldsymbol{H}}$ defined as $\|z\|^2_{\boldsymbol{H}} = z^\top \mathrm{diag}\left(\|z\|^2\right)^{-1}_{\|z\|_\infty \leq Z} z$. Note that

$$\left( I + (\mathbb{E}\hat{A})^{-1} \left( \hat{A} - (\mathbb{E}\hat{A}) \right) \right) \left( (\mathbb{E}\hat{A})^{-1} - \hat{A}^{-1} \right) = (\mathbb{E}\hat{A})^{-1} \left( \hat{A} - (\mathbb{E}\hat{A}) \right) (\mathbb{E}\hat{A})^{-1} \tag{E.7}$$

When $n$ is large enough, we know that $\frac{1}{2}I \preccurlyeq I + (\mathbb{E}\hat{A})^{-1} \left( \hat{A} - (\mathbb{E}\hat{A}) \right) \preccurlyeq I$ with high probability. Thus the term $\|\hat{u}_1 - \hat{u}_2\|^2_{H^1}$ is at the scale of $\left\| \left( \mathbb{E}\hat{A} \right) - \hat{A} \right\|^2_{\boldsymbol{H}} \approx \frac{Z^d}{n}$, which is of the same magnitude as what we get from the empirical process approach in our main proof. It is also larger than $\frac{Z^{d-2}}{n}$, which is the magnitude of the variance term for $\hat{u}_1$. Therefore, here we conjecture that the our bound for DRM itself is tight and leads to the sub-optimal convergence rate.

