# OpenReview forum: "Machine Learning For Elliptic PDEs: Fast Rate Generalization Bound, Neural Scaling Law and Minimax Optimality"
_ICLR.cc/2022/Conference — ICLR 2022 Poster_

### Official Review · Reviewer_Er9e · 2021-10-28

**Correctness:** 3
**Technical Novelty And Significance:** 3
**Empirical Novelty And Significance:** 3
**Recommendation:** 6
**Confidence:** 3

**Main Review:**

Strength:
1. The authors derived faster upper bound for PINN and DRM than existing results.

2. The authors proposed a modified DRM. The modified method together with truncated Fourier basis method.

3. The proof is very detailed and clear.


Weakness:
1. Compared with the network architecture in (Jiao et. al., 2021a) and (Duan et. al., 2021), the architecture used in this paper is a sparse network, which makes it less practical.

2. Another draw back is that the authors assumes the gradient of any function in the network class is bounded by a constant. This assumption is difficult to satisfy since the gradient depends on the network width and depth. In the theorems, the depth depends on the number of samples.

3. The organization of the paper needs to be improved. I believe the main theorems are Theorem A10, A11, A13 and A14. However, the authors only stated informal versions of these theorems in the body part. Even in these theorem mentioned above, the conditions are unclear. For example, in Theorem A9, the authors require the network class has bounded gradient. But this assumption is not mentioned in A10 and A11, which I believe is proved based on A9. The assumption in Proposition 2.1 is not mentioned in A10 and A11, either.

4. For numerical experiments, the authors only presented the errors versus the number of training data. The network architecture and how it is trained are not mentioned. What is the computational cost?

Comments:
1. On page 3, the authors said the variational problem considered in this paper and that considered in (Hutter & Rigollet, 2019; Manole et al, 2021) are different and leads to technical difference. Could the authors comment on the difference?

2. On page 3, the authors said the proof in this paper can be extended to nonlinear ones. Could the authors briefly explain how to extend it?

3. Page 15: At the end of the proof of Theorem A.1, the right-hand side of the last inequality has a factor max(1,V_{min}). The inequality does not hold when V<1. But I think this can be fixed.

Typo:
page 3: '... can be extend to ...'-> '... can be extended to ...'

**Summary Of The Paper:**

This paper studies the the statistical error of the Deep Ritz Method and Physics-Informed Neural Networks using neural networks and truncated Fourier basis in solving PDEs. The static Schrodinger equation is used as a prototype PDE. With appropriate assumptions, the authors established upper and lower bounds of the error for both methods. The upper bound derived in this paper improves existing results with a faster rate. The authors also proved that the upper bound of PINN is nearly optimal. Some numerical experiments are conducted to verify the results.

**Summary Of The Review:**

In general, this paper is a good paper and provides solid theoretical results on solving PDEs by deep neural networks. But the writing and organization need to be improved.

---

> ### Author Response · Authors · 2021-11-16
> **Author Response**
>
> Thanks for your insightful comments and careful correction. The detailed response to each point is as follows:
>
>
> **Compared with the network architecture in (Jiao et. al., 2021a) and (Duan et. al., 2021)**
>
> I guess (Jiao et. al., 2021a) and (Duan et. al., 2021) also need the sparsity assumption, for the sparsity (i.e. the number of weights) is used to control the Rademacher complexity. Fix me if I’m wrong.
>
> **Regarding the organization of the paper**
>
> Sorry for the confusion. We put all the assumptions we need in Theorem B.12 (in the revision version, in the original version it’s A.12) for PINN, Theorem B.9 (in the revision version, in the original version it’s A.9) for DRM, and Theorem C.1 (in the revision version, in the original version it’s B.1)  for MDRM. (Theorems A10, A11, A13, and A14 are considered as examples/corollaries of the meta theorems.) We’ve put the same assumption as theorem xxx in our revision.
>
> In our revision, we also add a proof sketch section in the appendix for readers to understand our work easier.
>
> **Regarding the experiments**
>
> We follow the same setting as [1]. We tried to put all the experimental details in the main text then it’ll exceed the 9-page limit. Thus we just mention the same network and the same training epochs as [1] in the main text. We can put a detailed version in our appendix if the reviewer considers it to be important.
>
> [1] Chen J, Du R, Wu K. A comparison study of deep Galerkin method and deep Ritz method for elliptic problems with different boundary conditions[J]. arXiv preprint arXiv:2005.04554, 2020.
>
> **Regarding (Hutter \&  Rigollet, 2019; Manole et al, 2021)**
>
>
> (Hutter \& Rigollet, 2019; Manole et al, 2021) deals with the Kantorovich dual, which doesn’t have a gradient in the variational form. In our paper, we need further to consider the complexity of the gradient term in the variational form and it leads to the modified deep ritz formulation. Actually the variance of Monte Carlo sampling the gradient term is quite large and we have a discussion of it in appendix E.
>
>
> **Extension to nonlinear problems**
>
> Our proof only needs regularity results shown in appendix B1 (in the modified version). Once you have this kind of regularity results (like the p-laplacian equation), then you can reproduce the decomposition in our paper. (but be careful, some of the problems may lead to suboptimal rates)
>
> **About $\max(1,V_{min})$**
>
> We apologize for the typo, it should be $\min (1,V_{min})$. We also fixed other typos we found recently, thanks to the reviewer for helping improve our manuscript.

---

### Official Review · Reviewer_16v3 · 2021-11-01

**Correctness:** 4
**Technical Novelty And Significance:** 4
**Empirical Novelty And Significance:** 2
**Recommendation:** 8
**Confidence:** 3

**Main Review:**

A number of recent works (many but not all cited by the authors) analyze ML-inspired numerical methods for PDE, with one
outstanding question being to what extent these methods evade the curse of dimension.  The present work uses far more sophisticated techniques from statistical learning theory, in particular a "localization technique for Rademacher complexity sums", and gets stronger results.  While not leading to dramatically new conclusions, these are apparently the first tight bounds for these problems.

**Summary Of The Paper:**

This paper carries out a variety of studies, both theoretical and numerical, on numerical solution of a Schrodinger equation using deep learning inspired methods.  The main results are upper and lower bounds on a power law scaling for sample complexity, function of dimension and regularity, which are tight for one of the methods.  Another method (Deep Ritz) has a proposed improvement.

**Summary Of The Review:**

Valuable technical advances on an important and popular topic.

---

> ### Author Response · Authors · 2021-11-16
> **Author Response**
>
> We thank the reviewer for the generous comments and score. We still also want to discuss some of the surprising findings as our contribution.
>
> **Regarding the new conclusions**
>
> Yes, we agree our technique is standard, but actually, we still consider we have some interesting new conclusions.  For PDE, one needs H1 localized Rademacher complexity which is smaller than L2 localized one for Fourier but not the case for **sparse** neural network. The consequence is that **sparse** neural networks are a good space for learning functions but are not a good space for solving PDEs. This is the first lesson we learned from this project.
>
> The second lesson we learned is sampling the gradient is actually more expensive which is discussed in appendix E. Based on this observation, we proposed Modified DRM which achieves the best mini-max optimal rate.

---

> > ### Comment · Reviewer_16v3 · 2021-11-27
> > **thanks**
> >
> > Thanks for the response.  I see, these are interesting points.

---

### Official Review · Reviewer_7DAR · 2021-11-02

**Correctness:** 4
**Technical Novelty And Significance:** 3
**Empirical Novelty And Significance:** 1
**Recommendation:** 6
**Confidence:** 3

**Main Review:**

The paper studies approximation power of DL-based PDE solvers, mainly the Deep Ritz Methods (DRM) and Physics-informed neural neural network method (PINN). A key observations is proposition 2.1/2.2, in which generalization/approximation error is upper/lower bounded tightly by certain energy functional $ E = E(u, |\nabla u|, \Delta u, ...)$. As such, the problems themself are reduced to the classical machine learning setting:  **generalization bound = complexity bound + approximation bound**. The paper proceeds (in the appendix) to control these two using local Rademacher complexity argument (where assumptions of uniform bounded are needed), B-splines, covering numbers... (The appendix is not very well organized!)

A couple comments of the experimental sections.

(1) Sec 6.3 seems not directly related to the paper (PDE solver). They are quite universally observed in DL.
(2) Could you provide a link for the code/ colab?

Crucial assumptions for Theorem 4.2, 4.2 etc have been mentioned in the main text, though it is brought up in discussion section.

[Strength] The paper provides several bounds (rigorous proof) for the approximation error for deep-learning-based PDE solvers, some of these bounds are tight.


[Weakness]
(1) The assumptions (e.g. eq (A.43), the functions, derivatives are UNIFORMLY bounded) are quite strong as acknowledged by the authors, which rule out many interesting cases and remove higher order terms in the proof.
(2) The paper is about approximation / representation power of neural networks. Optimization and thus optimization-related-generalization are not covered by the paper.




Minor:
(1) The appendix is long. It is not easy to find the proof of each theorem in the appendix. Please provide a pointer to the proof of each theorem.
(2) Given the length of the appendix ~ 50 pages, it should have a content page, a brief introduction to walk the readers through its main structure, what is the main theme of each section, etc.

**Summary Of The Paper:**

Applying deep learning (DL) to solving PDEs numerically has been a very exciting research directions. The current paper studies certain statistics properties of approximating linear Elliptic PDEs using neural networks (and truncated Fourier series). Under certain (quite strong) assumptions on the function class, the authors proved sharper (in some cases, tight)  bounds for the approximation errors.

**Summary Of The Review:**

DL-based PDE solvers are important and exciting direction of deep learning. The current paper provides several insights about the statistical properties of DL-based solver. Although there are several limitation (mentioned above), I think the contribution is significant enough for a ICLR publication and many researchers in the ICLR community will find it insightful (at least for me)

---

> ### Author Response · Authors · 2021-11-16
> **Author Response**
>
> Thanks for your insightful comments and careful correction.  We must apologize for the inconvenience to understand our proof and we made a new revision of the paper appendix A2.  The detailed response to each point is as follows:
>
> **Regarding the boundedness assumption**
>
> Yes, we admit this. This is because of the drawback of using localization Rademacher complexity ( for we used a Talagrand inequality). The recent work of [1] proposes a way to do the localization without the boundedness assumption (by avoiding using the Talagrand inequality) may help to address this problem.
>
> [1] Xu Y, Zeevi A. Towards Optimal Problem Dependent Generalization Error Bounds in Statistical Learning Theory[J]. arXiv preprint arXiv:2011.06186, 2020.
>
> **Regarding the optimization error**
>
> Yes, this is always the difficulty when we use a neural network. But in the neural tangent kernel regime, one can actually say something [2].
>
>
> [2] Nitanda A, Suzuki T. Optimal rates for averaged stochastic gradient descent under neural tangent kernel regime. ICLR 2020.
>
> **About Our Revision**
>
> For we provide a lower bound and upper bound for three methods, after we put all the main results (even not the formal version) in the main text, we’ve already exceeded the page limit. We put an organization of the appendix and proof sketch in the new appendix. Sorry again for the confusion. We hope to hear from you about your opinion about our newly added proof sketch. In case it's still not clear, we would appreciate your feedback to improve it.
>
> **Section 6.3**
>
> In this section 6.3, we aim to show when the approximation error is extremely small, then we’ll only have the generalization error which leads to overcoming the curse of dimensionality results. Another view is that if the s in our rate $\frac{2s-2}{d+2s-4}$ is very large (much larger than $d$), then the rate will go back to the parametric rate.
> The code version is still nasty, I’ll try to clean the code for section 6.3 as fast as I can and upload it as an appendix.  (colab link will break the anonymous? We can provide a colab version after the decision process.)
>
> **About Our Revision**
>
> For we provide a lower bound and upper bound for three methods,  after we put all the main results (even not the formal version) in the main text, we’ve already exceeded the page limit. We put an organization of the appendix and proof sketch in the new appendix. Sorry again for the confusion.

---

### Official Review · Reviewer_RB8r · 2021-11-02

**Correctness:** 3
**Technical Novelty And Significance:** 3
**Empirical Novelty And Significance:** 2
**Recommendation:** 8
**Confidence:** 4

**Main Review:**

The paper is well motivated and understanding the generalization and optimality rates for NNs for PDEs is an important problem. Moreover, the authors provide a lower bound which is important.

However, I would like to point out that currently the way the paper is presented is very hard to follow. Most of the discussion and proof sketches are left for the appendix, and the theorems that are presented in the main paper are informally stated. Furthermore, there is not proof sketch provided with the theorems that would intuit how the result was proved. This makes it hard to verify the results.

For example, like the previous work (Duan et al., 2022 and Jiao et al., 2021) the authors use a spline based construction to prove the neural network approximability. However the authors point out that they get better bounds due to the strong convexity of DRM and PINN loss. However, I am not sure where exactly is it helping when compared to the proof sketch provided in the previous paper? An explanation of this in the main paper is lacking.

Regarding the lower bound, I am quite confused about what authors mean by "The lower bound shows a non-standard exponent different from non-parametric estimation of a function."

**Summary Of The Paper:**

This paper establishes statistical lower bounds and upper bounds for (a modified) Deep Ritz method and PINNs based learning of solutions of PDEs when the estimators belong either to a class of sparse neural networks or lie in truncated fourier basis. They utilize the fact that the objective in DRM and PINNS is strongly convex, and use it to get a faster generalization bound O(1/n) instead of O(1/\sqrt{n}). Given that the upper bound for the initial non-modified version of DRM does not match the lower bound, they introduced a modified deep ritz method, where the number of samples to estimate the gradient squared is greater than (the ration is provided in the statement) rest of the objective. This enables them to achieve minimax optimality for DRM as well.

Through their experiments they verify that the number of training samples n and the test error follows a power low with \alpha = 1/d as indicated by the derived rates.

**Summary Of The Review:**

I think the results of the paper are relevant however, I think that the exposition needs to be improved for the main paper, as currently it is hard to follow.

---

> ### Author Response · Authors · 2021-11-16
> **Author Response**
>
> We thank the reviewer for the comments and interest in our work. We apologize for the inconvenience to understand our proof and we made a new version of Appendix A2 trying to clarify further. We hope this can help the reviewer evaluate our paper.
>
> The detailed response to each point is as follows:
>
> **Regarding the proof technique**
>
> Sorry for the lack of proof sketch in our paper. We only put a simple version of the proof sketch at the beginning of section 4. We now provide a detailed version in Appendix A2 in our revision.
>
>
> **Why Strongly Convex leads to Faster Rate**
>
> Our result is in analogous to the “Fast rates of convergence for learning problems” [1] in machine learning theory/empirical process. For a complete overview of this topic, we’d encourage one to read [2].
> A simple intuition of this problem can be explained by the following example.
>
>
> Let us consider learning the mean of the distribution p by minimizing l2 loss (population loss $mean=\arg\min_\theta E_{x} (\theta-x)^2$ and empirical loss $\hat\theta=\arg\min_\theta\frac{1}{n}\sum_{i=1}^n(\theta-x_i)^2=\frac{1}{n}\sum_{i=1}^n x_i$ ).  From chernoff inequailty, we know that $\hat\theta-\theta$ is $O(1/\sqrt{n})$. Thus we know the concerntration of the loss function/excess risk will have $ E_{x} (\hat \theta-x)^2- E_{x} (mean-x)^2=\|\theta-mean\|^2=O(1/n)$ . The fast rate is from the square like property (precisely Bernstein condition) of the loss function. (The analysis for mean estimation is easy due to the closed form solution, which is not the case for general problems. But it's an Intuition why the rate should be $O(1/n)$.)
>
>
> To prove the $O(1/n)$ convergence of the loss, we use localization [1,2]. Localization technique will bound the generalization error as the solution of the fixed point equation $R_n(\{f|Loss(f)-Loss(f^\ast)\le r\})=r$. If we can bound $R_n(\{f|Loss(f)-Loss(f^\ast)\le r\})=O(\sqrt{\frac{r}{n}})$ (the dependency of $r$ will rely on the strongly convex/square like condition), then the solution of the fixed point equation is $O(1/n)$. (the solution of $\sqrt{\frac{r}{n}}=r$ is $r=1/n$)
>
> The difference here is the metric used to define the localized set. In the previous papers, the metric used is the $\ell_2$ distance, while the $H_1$/$H_2$ norm used in our paper leads to tighter bound for the Rademacher complexity.
>
> We’ve put a detailed discussion about this part in appendix A2. Hope this can help the reviewer to understand the proof and change the evaluation of our paper.
>
> [1] https://web.stanford.edu/class/cs229t/2017/Lectures/fast-rates.pdf
>
> [2] Xu Y, Zeevi A. Towards Optimal Problem Dependent Generalization Error Bounds in Statistical Learning Theory. Neurips 2020.
>
>
> **Regarding the non-standard exponent**
>
> For estimating a function in $H_s$ the convergence rate is $n^{-2s/d+2s}$ but our bound looks more like  $n^{-(2s-2)/d+2s-4}$ for the deep ritz method. The $2s-2$ in the numerator is different from the $2s-4$ in the denominator is the part we consider as non-standard. The $2s-2$ in the lower bound is because of the regularity loss from the PDE problem. The $2s-2$ in the upper bound comes from the $H_1$ version (but not the $\ell_2$ version widely used) localized Rademacher complexity. We’ve put this in the main text in the revision.
>
> **About Our Revision**
>
> For we provide a lower bound and upper bound for three methods,  after we put all of the main results in the main text, we’ve already exceeded the page limit. We put an organization of the appendix and proof sketch in the new appendix. Sorry again for the confusion.
>
> We must apologize for the confusion led by the presentation and try to add a proof sketch in the appendix. Hope this can address the reviewer’s concern and increase the evaluation of our paper accordingly. We hope to hear from you about your opinion about our newly added proof sketch. In case it's still not clear, we would appreciate your feedback to improve it.

---

### Decision · Program_Chairs · 2022-01-20

**Decision:**

Accept (Poster)

**Comment:**

This paper proposes a new theory for modified DRM and PINN for solving elliptical PDEs, and delivers valuable advances on important topics.